# Evaluation of random forests for short-term daily streamflow forecasting in rainfall and snowmelt-driven watersheds

Leo T. Pham[1], Lifeng Luo[2], and Andrew O. Finley[1,2]

[1]Department of Forestry, Michigan State University, East Lansing, Michigan, USA
[2]Department of Geography, Environment, and Spatial Sciences, Michigan State University, East Lansing, Michigan, USA

**Correspondence:** Leo Pham (phamleo@msu.edu)

**Abstract.** In the past decades, data-driven machine learning (ML) models have emerged as promising tools for short-term streamflow forecasting. Among other qualities, the popularity of ML models for such applications is due to their relative ease in implementation, less strict distributional assumption, and competitive computational and predictive performance. Despite the encouraging results, most applications of ML for streamflow forecasting have been limited to watersheds where rainfall is the major source of runoff. In this study, we evaluate the potential of random forests (RF), a popular ML method, to make streamflow forecast at 1-day lead time at 86 watersheds in the Pacific Northwest. These watersheds cover diverse climatic conditions and physiographic settings and exhibit varied contributions of rainfall and snowmelt to their streamflow. Watersheds are classified into three hydrologic regimes: rainfall-dominated, transient, and snowmelt-dominated, based on the timing of center-of-annual flow volume. RF performance is benchmarked against naïve and multiple linear regression (MLR) models and evaluated using four criteria: coefficient of determination, root mean squared error, mean absolute error, and Kling-Gupta efficiency (KGE). Model evaluation scores suggest that the RF performs better in snowmelt-driven watersheds compared to rainfall-driven watersheds. The largest improvement in forecasts, compared to benchmark models, are found among rainfall-driven watersheds. RF performance deteriorates with increase in catchment slope and soil sandiness. We note disagreement between two popular measures of RF variable importance and recommend jointly considering these measures with the physical processes under study. These and other results presented provide new insights for effective application of RF-based streamflow forecasting.

## 1 Introduction

Nearly all aspects of water resource management, risk assessment, and early-warning systems for floods rely on accurate streamflow forecast. Yet streamflow forecasting remains a challenging task due to the dynamic nature of runoff in response to spatial and temporal variability in rainfall and catchment characteristics. Therefore, development of skillful and robust streamflow models is an active area of study in hydrology and related engineering disciplines.

While physical models remain a common and powerful tool for predicting streamflow, ML models are gaining popularity due to some of their unique qualities and potential advantages. Compared with the often labor-intensive and computationally expensive task of parameterizing in physical model (Tolson and Shoemaker, 2007; Boyle et al., 2000), ML models are data-

25 driven and can identify patterns in the input-output relationship without explicit knowledge of the physical processes and onerous computational demand. To make up for their limited ability to provide interpretation of the underlying mechanisms, ML models often require less calibration data than physical models, have demonstrated high accuracy in their predictive performance, are computationally efficient, and can be used in real-time forecasting (Adamowski, 2008; Mosavi et al., 2018). ML models are particularly useful when accurate prediction is the central inferential goal (Dibike and Solomatine, 2001),

whereas conceptual rainfall-runoff model can provide a better understanding of hydrologic phenomena and catchment yields and responses (Sitterson et al., 2018). Artificial neural networks (ANN), neuro-fuzzy (a combination of ANNs and fuzzy logic), support vector machine (SVM), and decision trees (DT) are reported to be among the most popular and effective for both short-term and long-term flood forecast (Mosavi et al., 2018). For example, Dawson et al. (2006) provided flood risk estimation at ungauged sites using ANN at catchments across United Kingdom. Rasouli et al. (2012) predicted streamflow at

lead times of 1-7 days with local observations and climate indices using three ML methods: Bayesian neural network (BNN), SVM, and Gaussian process (GP). They found BNN outperformed multiple linear regression (MLR) as well as the other two ML models. Their study also found models trained using climate indices yielded improved longer lead time forecasts (e.g., 5–7 days). Tongal and Booij (2018) forecasted daily streamflow in four rivers in the United States with SVR, ANN, and RF coupled with a baseflow separation method (i.e., separating the two different components of streamflow into baseflow and

surface flow). Obringer and Nateghi (2018) compared eight parametric, semi-parametric, and non-parametric ML algorithms to forecast urban reservoir levels in Atlanta, Georgia. Their results showed RF yielded the most accurate forecasts.

Despite the promising results reported in existing literature, most ML streamflow forecast applications are limited to watersheds where rainfall is the major contributor. In many settings, particularly non-arid mountainous regions in Western USA, a combination of rainfall and spring snowmelt can drive streamflow (Johnstone, 2011; Knowles et al., 2007). The amount of snow

accumulation and its contribution to discharge also vary among the watersheds (Knowles et al., 2006). Both watershed-scale hydrologic and statistical models have been used to assess the current and future stream hydrology and associated flood risks (Salathé Jr et al., 2014; Wenger et al., 2010; Tohver et al., 2014; Pagano et al., 2009). Safeeq et al. (2014) simulated streamflows in 217 watersheds at annual and seasonal time scales using the Variable Infiltration Capacity (VIC) model at 1/16° and 1/20° spatial resolutions. The study found that the model was able to capture the hydrologic behavior of the studied watersheds with

a reasonable accuracy. Yet the authors recommend careful site-specific model calibration, using not only streamflow but also snow water equivalent (SWE) data, would be expected to improve model performance and reduce model bias. Pagano et al. (2009) applied Z-score regression to daily SWE from Snow Telemetry (SNOTEL) stations and year-to-date precipitation data to predict seasonal streamflow volume in unregulated streams in Western US. The authors reported the skill of these forecasts is comparable to the official published outlooks. A natural question is whether ML models can produce comparable performance

in these watersheds where streamflow contributions come from a mixture of snowmelt and rainfall, as well as where snowmelt dominates sources. Considering the prominent role of snowpack in water management and contribution of rapid snowmelt in flood events, such question is worth exploring. To this end, we evaluate the potential of RF in making short-term streamflow forecast at 1-day lead time across 86 watersheds in the Pacific Northwest Hydrologic Region (Fig. 1). The U.S. Geological Survey (2020) defines this region as hydrologic region 17 or HUC 17. HUC-17 consists of sub-basins and watersheds of the

Columbia River that span varying hydrologic regimes. The selected watersheds have long-term record of unregulated streamflow and different streamflow contributions of rainfall and snowmelt. Drainage basin factors such as topography, vegetation, and soil can affect the response time and mechanisms of runoff (Dingman, 2015). Few studies attempted to account for or report these effects on models' performance. Without such consideration, it is difficult to determine if a data-driven model can be generalized to watersheds not included in the given study. Therefore, our objectives are (1) to examine and compare the performance of RF in a number of watersheds across hydrologic regimes and (2) to explore the role of catchment characteristics in model performance that are overlooked in previous studies.

In practice, RF can be trained to forecast streamflow at various timescales, depending on the input variables provided. Rasouli et al. (2012) forecasted streamflow at 1-7 day lead times using three ML models and data from combinations of climate indices and local meteo-hydrologic observations. The authors concluded that models with local observations as predictors were generally best at shorter lead times while models with local observations plus climate indices were best at longer lead times of 5–7 days. Also, the skillfulness of all three models decreased with increasing lead times. In our study, we focused on 1-day lead time forecasting and therefore did not include long-term climate information. At longer lead times, changes in weather conditions would likely exert much greater control on runoff and the performance of the model.

We select RF to forecast streamflow for two reasons. First, RF has been referenced to deliver high performance in short-term streamflow forecasts (Mosavi et al., 2018; Papacharalampous and Tyralis, 2018; Li et al., 2019; Shortridge et al., 2016), making it a good candidate for our study. Second, RF allows for some level of interpretability. This is delivered through two measures of predictive contribution of variables: mean decrease in accuracy (MDA) and mean decrease in node impuritiy (MDI). These two measures have been widely used as means for variable selection in classification and regression studies in bioinformatics (Chen and Ishwaran, 2012), remote sensing classification (Pal, 2005), and flood hazard risk assessment (Wang et al., 2015). The interpretability of a ML model, however, can be a controversial subject and remains an active area of study (Ribeiro et al., 2016; Carvalho et al., 2019). Both model-agnostic, such as permutation-based feature importance (Breiman, 2001), and model-specific, such as gini-based for RF (Breiman et al., 1984) and gradient-based for ANNs (Shrikumar et al., 2017), interpretation methods can provide useful insights into how the ML models make their predictions. While the referred interpretability does not directly translate to interpretation of the physical processes, it can provide insight into relationships among predictors and streamflow response.

The remainder of the paper is arranged as follows. Section 2 provides a brief introduction to RF, relevant parameters (which can also be referred to as "hyper-parameters" in the ML literature), and selected evaluation criteria. Section 3 describes the study area, datasets, and predictor selection. Results and discussion are given in Sect. 4 along with limitations and recommendation for future research. A summary and indication of future work are is provided in Sect. 5.

**Figure 1.** (a) Elevation (m) shading map showing the Pacific Northwest Hydrologic Unit, 86 selected stream gauges (triangles), and their drainage area (cyan delineation lines), and SNOTEL stations (brown squares). Examples of annual hydrographs of (b) rainfall-dominated, (c) transient, and (d) snowmelt-dominated watersheds. Figures (b-d) are based on 2009-2018 daily flow data at three sites 12043300 (48.2° N, 124.4° W), 12048000 (48° N, 123.1° W), and 10396000 (42.7° N, 118.9° W) respectively.

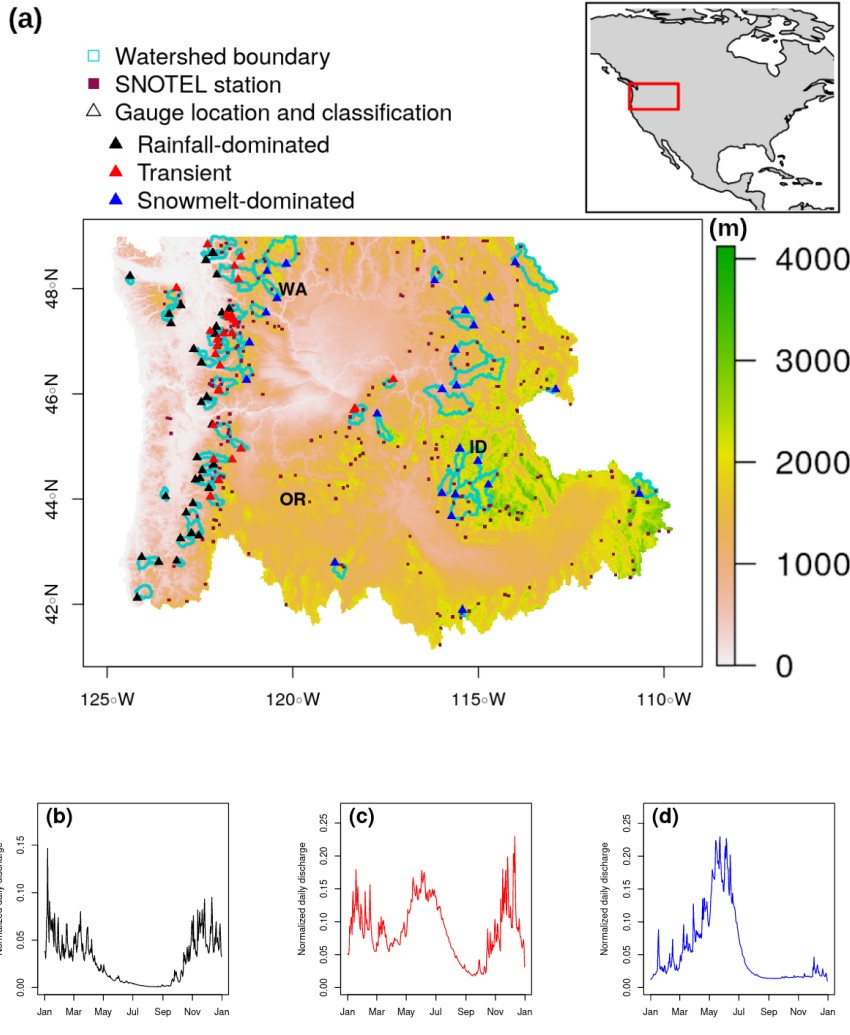

**Figure 2.** Structure of a RF and relevant parameters

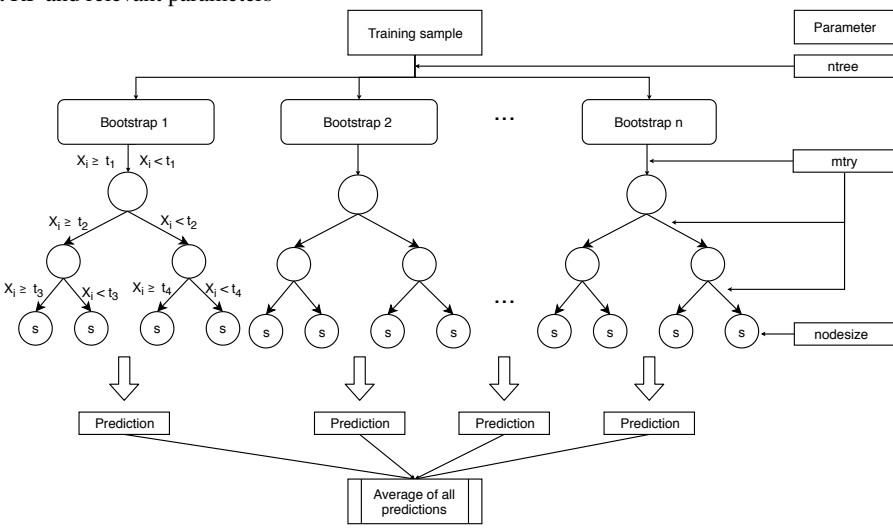

## 2 Methodology

### 2.1 Random forests

Proposed by Breiman (2001), RF is a supervised, non-parametric algorithm within the decision tree family that comprises an ensemble of decorrelated trees to yield prediction for classification and regression tasks. Non-parametric methods such as RF do not assume any particular family for the distribution of the data (Altman and Bland, 1999). Since a single decision tree can produce high variance and is prone to noise (James et al., 2013), RF addresses this limitation by generating multiple trees where each tree is built on a bootstrapped sample of the training data (Fig. 2, Algorithm 1). Each time a binary split is made in a tree (also known as split node), a random subset of predictors (without replacement) from the full set of predictor variables is considered (Fig. 2). One predictor from these candidates is used to make the split where the expected sum variances of the response variable in the two resulting nodes is minimized (Algorithm 1, Step 3). The randomization process in generating the subset of the features prevents one or more particularly strong predictor from getting repeatedly chosen at each split, resulting in highly correlated trees (Breiman, 2001). After all the trees are grown, the forests make prediction on a new data point by having all trees run through the predictors. In the end, the forests cast a majority vote on a label class for classification task or produce a value for regression task by averaging all predictions. Breiman (2001) provided full details on RF and its merit. The `randomForest` package in R developed by Liaw et al. (2002) was used for model training and validation in our study. The step-by-step of building a regression RF follows:

---
**Algorithm 1** Building a regression RF
---
**Step 1:** $n$ bootstrap samples are drawn from training set, each has the same size as the training sample. This is also known as `ntree` or number of trees in the forest.

**Step 2:** At each binary node split, a subset of `mtry` predictors, $X_i$, is randomly selected from $p$ predictor space, $\Omega_p$, that results in $X_i \in \Omega_p$ for $\{i \in 1,..., \texttt{mtry}\}$, $\texttt{mtry} < p$.

**Step 3:** The single best combination of predictor $X_i$ among $X$ predictor variables and threshold $t$ is selected to split the observations, $y_j$, into binary regions $R_1 = \{\, y_j | X_i < t \,\}$ and $R_2 = \{\, y_j | X_i \geq t \,\}$ that minimize:

$$\sum_{j:y_j \in R_1} (y_j - \hat{y}_{R_1})^2 + \sum_{j:y_j \in R_2} (y_j - \hat{y}_{R_2})^2 \tag{1}$$

where $\hat{y}_{R_1}$ is the mean of observations in $R_1$ and $\hat{y}_{R_2}$ is the mean of observations in $R_2$.

**Step 4:** Repeat step 2-3 until all terminal region contains less than `nodesize` observations.

---

Due to sampling with replacement, some observations may not be selected during the bootstrap. These are referred to as out-of-bag or OOB and used to estimate the error of the tree on unseen data. It has been estimated that approximately 37% of samples constitute OOB data (Huang and Boutros, 2016). An average OOB error is calculated for each subsequently added tree to provide an estimate of the performance gain. The OOB error can be particularly sensitive to the number of random predictors used at each split `mtry` and number of trees `ntree` (Huang and Boutros, 2016). Generally, the predictive performance improves (or OOB error decreases) as `ntree` increases. However, recent research has shown that depending on the dataset, there is a limit for number of trees where additional growing does not improve performance (Oshiro et al., 2012). It has been advised that `mtry` is set to no larger than 1/3 of total number of predictors for optimal regression prediction (Liaw et al., 2002), which is also the default value in `randomForest` function in R and widely adopted in literature. Nevertheless, Huang and Boutros (2016) found that this value is dataset-dependent and could be tuned to improve the performance of RF. Bernard et al. (2009) argued that the number of relevant predictors highly influences optimal `mtry` value. In this study, we select the optimal `mtry` using an exhaustive search strategy, in which all possible values of `mtry` are considered, using R package `Caret` (Kuhn et al., 2008). While all considered parameters might have an effect on the performance of RF, we chose to focus on two parameters, `ntree` and `mtry`, for a number of reasons. The main reason is that these two parameters were originally introduced by Breiman (2001) in the development of RF algorithm. Second, `ntree` in a forest is a parameter that is tunable but not optimized and should be set sufficiently high (Oshiro et al., 2012; Probst et al., 2019) for RF to achieve good performance. It has been theoretically proven that more trees are always better (Probst et al., 2019). In other words, optimal ntree value can go to infinity. The reduction in error, however, becomes negligible after a sufficiently large number of trees. Furthermore, empirical results provided in previous works suggest that `mtry` is the most influential out of parameters in RF (Bernard et al., 2009; Van Rijn and Hutter, 2018; Probst et al., 2019). Figure 2 illustrates the step-by-step operating principle of growing RF and the relevant parameters.

## 2.2 Variable importance in random forests

In addition to assessing a model's overall predictive ability, there is also interest in understanding the contribution of each predictor variable to model performance. There are two built-in measures for assessing variable importance in RF: mean decrease in accuracy (MDA) and mean decrease in node impurity (MDI). Both were developed by Breiman (Breiman et al., 1984; Breiman, 2001). After all trees are grown, OOB data during training is used to compute the first measure. At each tree, the mean squared error (MSE) between predicted and observed is calculated. Then the values of each of the $p$ predictors are randomly permuted with other predictor variables held constant. The difference between the previous and new MSE is averaged over all trees. This is considered the predictor variable's MDA (Liaw et al., 2002) and values are reported in percent difference in MSE. The procedure is repeated for each predictor variable. Given that there is a strong association between a predictor and response variable, breaking such bond would potentially result in large error in the prediction (i.e., large MDA). MDA value can be negative where a predictor has no predictive power and adds noise to the model. Strobl et al. (2007), however, expressed caution that permutation-based measures such as MDA could show a bias towards correlated predictor variables by overestimating their importance, particularly in high-dimensional data sets.

The second method, MDI, measures the average error reduction each time a predictor is selected to make a split during training. It is based on the principle that a binary split only occurs when residual errors (or impurity) of two descendent nodes are less than that of their parent node. The MDI of a predictor is the sum of all gains across all trees divided by the number of trees. Because the scale of MDI depends on values of response variable, raw MDI provides little interpretation. Following Wang et al. (2015), we computed relative MDI for each variable, which in our case is calculated by dividing each predictor variable's MDI by the sum of MDI from all predictors at each watershed. When scaled by 100, this relative MDI is a percentage and can be interpreted as the relative contribution of each predictor to the total reduction in node impurities. In the case where a predictor makes no contribution during the splitting, the relative MDI would be effectively zero. For both measures, the larger the value, the more important the predictor.

## 2.3 Benchmark models

We benchmark the performance of RF during the validation period against multiple linear regression (MLR) and simple naïve models using the calculated Pearson correlation coefficient ($r$) between forecasted and observed values for each model. In naïve model, we assume "minimal-information" scenario and the best estimate of the streamflow from the next day is the observed value from current day (Gupta et al., 1999). Its $r$, in this case, is the 1-day autocorrelation coefficient in the time series and measures of the strength of persistence. We train and verify MLR model using same data sets and predictors supplied to RF model.

## 2.4 Performance evaluation criteria

There exist different model performance criteria and each provides unique insights on the correspondence between forecasted and observed streamflow values. While $r$ and its square, namely coefficient of determination ($R^2$), are often used, Legates

and McCabe Jr (1999) discussed the limitation of these two measures where they were reported to be especially oversensitive to extreme values or outliers. The authors recommended that absolute error measures (i.e., root mean squared error or mean absolute error) and goodness-of-fit measure, such as the Nash-Sutcliffe efficiency (NSE), could provide more reliable and conservative assessment of the models. Kling-Gupta efficiency (KGE) is a relatively new metric that was developed based on a decomposition of NSE (Gupta et al., 2009). This goodness-of-fit measure is gaining popularity as a benchmark metric for hydrologic models by addressing several shortcomings diagnosed with NSE. For these reasons, we selected the following four criteria to evaluate RF performance: $R^2$, RMSE, MAE, and KGE. These criteria cover various aspects of model's performance and also provide intuitive interpretation as explained in the remainder of this section.

$R^2$ can be interpreted as the proportion of the variance in the observed values that can be explained by the model. Values are in the range between 0 and 1 where 1 indicates the model is able to explain all variation in the observed dataset.

$$R^2 = \left( \frac{\sum\limits_{i=1}^{N} (\hat{y}_i - \overline{\hat{y}})(y_i - \overline{y})}{\sqrt{\sum\limits_{i=1}^{N} (\hat{y}_i - \overline{\hat{y}})^2} \sqrt{\sum\limits_{i=1}^{N} (y_i - \overline{y})^2}} \right)^2 \tag{2}$$

where N is total number of the observations during the validation period, $\hat{y}_i$ and $y_i$ are the forecasted and observed values at day $i$ respectively with

$$\overline{y_i} = \frac{1}{N} \sum\limits_{i=1}^{N} y_i \quad \text{and} \quad \overline{\hat{y}_i} = \frac{1}{N} \sum\limits_{i=1}^{N} \hat{y}_i \tag{3}$$

MAE provides an average magnitude of the errors in the model's predictions without considering the direction (underestimation or overestimation).

$$MAE = \frac{\sum\limits_{i=1}^{N} |\hat{y}_i - y_i|}{N} \tag{4}$$

RMSE is the standard deviation of the residuals between the predictions and observations. It is more sensitive to larger error due to the squared operation. Both MAE and RMSE scores range between 0 and $\infty$ where a score of 0 indicates a perfect match between predicted and observed data. The standardization in streamflow measurements (described in Sect. 3) allows comparison of MAE and RMSE across gauges.

$$RMSE = \sqrt{\frac{\sum\limits_{i=1}^{N} (\hat{y}_i - y_i)^2}{N}} \tag{5}$$

KGE metric ranges between $-\infty$ and 1. While there currently is not a definitive KGE scale, Knoben et al. (2019) showed KGE values in the range between $-0.41$ and 1 indicate the model improves upon the mean flow benchmark, which assumes the predicted streamflow values equal to the mean of all observations. KGE value of 1 suggests the model can perfectly reproduce observations. KGE is calculated as follows:

$$KGE = 1 - \sqrt{(r-1)^2 + (\alpha-1)^2 + (\beta-1)^2} \tag{6}$$

where $r$ is the Pearson correlation coefficient, $\alpha$ is a measure of relative variability in the forecasted and observed values, and $\beta$ represents the bias:

$$\alpha = \frac{\sigma_{\hat{y}}}{\sigma_y} \quad \text{and} \quad \beta = \frac{\mu_{\hat{y}}}{\mu_y} \tag{7}$$

where $\sigma_{\hat{y}}$ is the standard deviation in observations, $\sigma_y$ is the standard deviation in forecasted values, $\mu_{\hat{y}}$ is the forecasted mean, and $\mu_y$ is observation mean.

In hydrological forecast, one might be interested in the ability of the model to capture more extreme events rather than the overall performance. This is particularly relevant in flood risk assessment and flood forecasting where floods are associated with discharge exceeding a high percentile (typically $\geq 90^{\text{th}}$)(Cayan et al., 1999). The definition of "extreme" depends on the objective of the study. Here, we adopt the peak-over-threshold method. For the validation period, we calculated the $90^{\text{th}}$, $95^{\text{th}}$, and $99^{\text{th}}$ percentile streamflow values at each watershed. These are considered thresholds. If an observed daily streamflow exceeded this threshold, it would be considered an extreme event. We measure the ability of RF to capture these events using two additional criteria: probability of detection (POD) and false alarm rate (FAR). The calculation followed as in (Karran et al., 2013).

$$POD = \frac{P(\hat{y}_i > \omega | y_i > \omega)}{P(y_i > \omega)} \tag{8}$$

and

$$FA = \frac{P(\hat{y}_i > \omega | y_i < \omega)}{P(y_i < \omega)} \tag{9}$$

where $\omega$ is a specified threshold.

## 3  Study Area and data

### 3.1  Watersheds in the Pacific Northwest Hydrologic Region

In this study, we focus on watersheds in the Pacific Northwest Hydrologic Region (Fig. 1). This region covers an area of 836,517 km$^2$ and encompasses all of Washington, six other states, and British Columbia, Canada. For the purpose of maintaining consistency in monitoring protocol and data, we only consider watersheds on the US territory. The Columbia River and its tributaries make up the majority of the drainage area, traveling more than 2000 km with an extensive network of more than 100 hydroelectric dams and reservoirs have been built along these river channels. Hydropower in the Columbia River Basin

supplies approximately 70 percent of Pacific Northwest energy (Payne et al., 2004). Flood control is also an important aspect of reservoir operation in this region.

The north-south running Cascade Mountain Range divides the region into eastern and western parts and strongly influence the regional climate. The windward (west) side of the mountain receives an ample amount of winter precipitation compared to the leeward (east) side. When temperature falls near freezing point, precipitation comes in the form of snow and provides

water storage for dry summer months. Summers tend to be cool and comparatively dry. East of the Cascades, summer rainfall result from rapidly built thunderstorm and convective events that can produce flash floods (Mass, 2015). For this region, proximity to the ocean creates a more moderate climate with a narrower seasonal temperature range compared to the inland areas, particularly in the winter. Spatial trends and variations in annual mean temperature, total precipitation, drainage area, and elevation of the watersheds are shown in Fig. 3.

**Figure 3.** Gauge locations with color gradient indicating variations in (a) drainage area (km$^2$), watershed mean elevation (m), (c) annual precipitation (cm), and (d) annual mean temperature (°C).

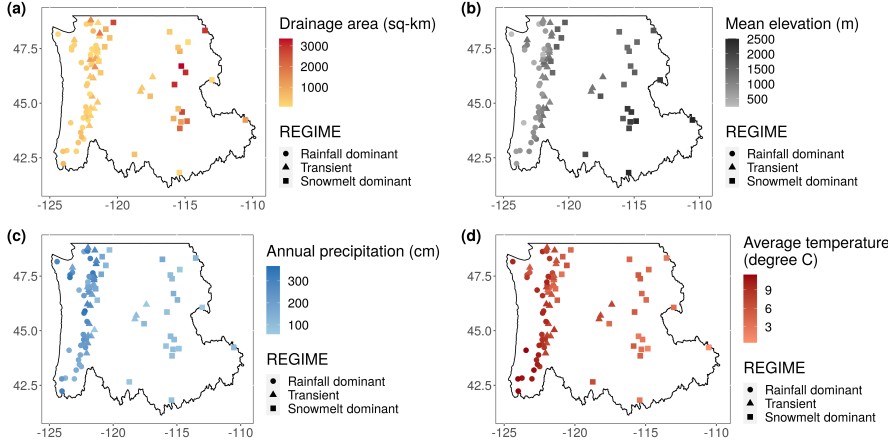

**Table 1.** Number of USGS gauges used in the study for each flow regime, mean watershed elevation, drainage area, annual precipitation, and annual mean temperature ranges.

| Hydrologic regime | Number of gauges | Mean watershed elevation (m) | Drainage area ($km^2$) | Mean annual precipitation (cm) | Mean annual temperature (°C) |
|---|---|---|---|---|---|
| Rainfall-dominated | 33 | 239 - 1207 | 58 - 703 | 122.0 - 367.0 | 5.4 - 11.5 |
| Transient | 28 | 813 - 1477 | 58 - 1855 | 63.2 - 314.0 | 4.16 - 8.42 |
| Snowmelt-dominated | 25 | 1349 - 2509 | 51 - 3355 | 58.0 - 177.0 | 0.4 - 6.62 |

## 3.2 Data

### 3.2.1 Streamflow

Our analysis uses streamflow data available through the USGS National Water Information System (NWIS) (https://waterdata. usgs.gov/nwis/sw). From NWIS, we selected daily streamflow time series for gauges using the following criteria: 1) continuous operation during the 10-year period between 2009 and 2018, 2) have less than 10 percent of missing data, and 3) positioned in watersheds with "natural" flow that is minimally interrupted by anthropogenic intervention. The third criterion was met using the GAGES-II: Geospatial Attributes of gauges for Evaluating Streamflow dataset (Falcone, 2011) classification to identify watersheds with least-disturbed hydrologic condition and represented natural flow. We performed additional screening by computing correlation coefficient between the respective gauge and mean basin streamflow and removed those with a correlation of less than 0.5. We also excluded small creeks with drainage area less than 50 $km^2$. In total, 86 watersheds were selected (Fig. 1).

Following methodology proposed in (Wenger et al., 2010), the watersheds were further grouped into three classes of hydrologic regimes based on the timing of center-of-annual flow, which is defined as the date at which half of the total annual flow volume is exceeded. The annual flow calculations follow a water-year calendar that begins October 1[st] and ends September 30[th]. These three hydrologic regimes include: "early" streams with flow time < 150 (27 February), "late" streams with flow time > 200 (18 April), and "intermediate" streams with flow time between 150 and 200. These hydrologic regimes correspond to rainfall-dominated, snowmelt-dominated, and transient or transitional (mixture of rain and snowmelt) hydrographs, respectively. While this particular classification and its variants have been used in various studies related to water resources in this region (Mantua et al., 2009; Elsner et al., 2010; Vano et al., 2015), we adopted this partition in our study for two reasons. First, as Regonda et al. (2005) pointed out, the classification provides a summary of information about type and timing of precipitation, timing of snowmelt, and the contribution of these hydro-climatic variables to streamflow. This helps us assess model performance in consideration of sources of runoff. Second, the classification provides a basis to generalize the results to other watersheds that are not part of the study.

On average, records at these watersheds have less than 3 percent missing data during the 2009–2018 period. The drainage area of the watersheds range between 51 $km^2$ and 3355 $km^2$, and the mean elevation range from 239 m and 2509 m, estimated from 30-m resolution digital elevation model (Table 1).

### 3.2.2 Precipitation

Daily precipitation observations were obtained from the AN81d PRISM dataset (Di Luzio et al., 2008). This gridded dataset has a resolution of 4 km, covers the entire continental US from January 1981 to present, and is continuously updated every 6 months. Best estimate gridded value is derived by using all the available data from numbers of station networks ingested by the PRISM Climate Group. A combination of climatologically aided interpolation (CAI) and radar interpolation were used in developing PRISM dataset. In our study, watershed daily precipitation time series were constructed by computing the arithmetic mean for precipitation values of all grid points that fall within the given watershed.

### 3.2.3 Snow water equivalent and temperatures

SWE is defined as the depth of water that would be obtained if a column of snow were completely melted (Pan et al., 2003). Daily SWE data were retrieved from 201 SNOTEL stations in HUC 17. These stations are part of the network of over 800 sites located in remote, high-elevation mountain watersheds in the western U.S. The elevation of these stations are in the range of 128 m and 3142 m. At SNOTEL sites, SWE is measured by a snow pillow—a pressure sensitive pad that weighs the snowpack and records the reading via a pressure transducer. As the temperature shift is the primary trigger for snowmelt, daily maximum temperature (TMAX) and minimum temperature (TMIN) from SNOTEL sensors were also retrieved and included as predictors for streamflow. The obtained data reflected the last measurement recorded for the respective day at each site. We only supplied the last measurement from SNOTEL stations because not all predictors have sub-daily values. The dataset is mostly complete, with 99.6 %, 99.6 %, and 99.9 % of the observations available for three variables TMAX, TMIN, and SWE respectively. Because of the sparse coverage of SNOTEL sites, daily average values were calculated at USGS basin level (6-digit Hydrological Unit), similar to the currently reported snow observations from National Water and Climate Center (www.wcc.nrcs.usda.gov/snow/snow_map.html), and subsequently applied to the watersheds located in that basin. There is a total of 15 basins, each contains a number of SNOTEL stations in the range between 6 and 30 (Table S2 in the Supplement). It is noted the *in situ* data from these of stations cannot capture the spatial variability of snow accumulation and computing an area-averaged snowpack value from observations remains a challenging task (Mote et al., 2018). The SNOTEL averages therefore represent first-order estimates of snow coverage and temperature conditions.

### 3.2.4 Predictor selection

Future daily mean streamflow ($Q_{t+1}$) is the response variable in our study. We attempt to explain the variability in $Q_{t+1}$ using eight relevant predictors from the three datasets (Table 2). Selection of predictors is based on thorough review of the literature from previous studies and our understanding of the hydrology of this region. Specifically, precipitation ($P_t$) is intuitively a driver of streamflow. $SWE_t$ provides storage information on the amount of accumulated snow available for runoff and is influenced by changes in temperature ($TMAX_t$ and $TMIN_t$). Given that there is high temporal correlation in daily temperatures, TMIN and TMAX data can provide useful signal to our streamflow forecast. Previous day streamflow ($Q_t$) is particularly important due to high degree of persistence that exist in the time series. A hydrological year consists of 73 pentads where each

**Table 2.** List of predictors.

| No. | Predictors | Index | Unit | Source |
|---|---|---|---|---|
| 1 | Streamflow at day $t$ | $Q_t$ | m$^3$ s$^{-1}$ | USGS |
| 2 | Precipitation | $P_t$ | mm | PRISM |
| 3 | Sum of 3-day precipitation ($P_t + P_{t-1} + P_{t-2}$) | $P3_t$ | mm | Derived from PRISM |
| 4 | Snow water equivalent | $SWE_t$ | mm | SNOTEL |
| 5 | Maximum temperature | $TMAX_t$ | °C | SNOTEL |
| 6 | Minimum temperature | $TMIN_t$ | °C | SNOTEL |
| 7 | Snowmelt ($SW_t$ - $SW_{t-1}$) | $SD_t$ | mm | Derived from SNOTEL |
| 8 | Pentad | $PEN_t$ | - | - |

comprises of five consecutive days and observation for each day is indexed with a pentad value between 1 and 73. Data pre-processing showed moderate to strong non-linear temporal correlation between daily streamflow and the pentad at each gauge. We also derived two variables: sum of 3-day precipitation ($P3_t$) and snowmelt ($SD_t$) from available data. Inclusion of 3-day precipitation was to account for large winter storms that can last for several days, which often result in surges in streamflow. $SD_t$ was calculated as the difference between SWE at day $t$ and $t-1$. A positive value of $SD_t$ indicates snow accumulation and negative value indicates melt.

Soil moisture is also a relevant variable in streamflow modeling as it controls the partition between infiltration and runoff of precipitation (Aubert et al., 2003). However, soil moisture data is often limited and incomplete, especially at daily interval and therefore not included in this study. The data were divided into two sets: training consisting of seven years 2009–2015 and a validation set of three years 2016–2018. We standardized training and validation data at each gauge using min-max scaling. First, we computed the min and max values from training data sets for each of the predictor and response variables at each watershed. These min and max values were then used to standardize both training and validation data sets. The training data, which were used to compute min-max values for standardization, therefore have values between 0 and 1. A flowchart representing the input-output model using RF is shown in Fig. 4.

## 4 Results and discussion

### 4.1 Parameter tuning

As we mentioned in Sect. 2, error rate in RF can be sensitive to two parameters: the number of trees `ntree` and number of randomly selected predictors available for splitting at each node `mtry`. We tested RF on training data sets of 30 randomly chosen watersheds and observed that the reduction in out-of-bag MAE error is negligible after 2000 trees. We then set `ntree=2000` for all 86 watersheds. `mtry`, on the other hand, was tuned empirically using a combination of exhaustive search approach and cross-validation.

The goal of tuning is to select the `mtry` parameter value that would optimize the performance of the model. The candidates were evaluated based on their OOB mean absolute error (MAE). At each watershed, eight possible candidate values of `mtry`

**Figure 4.** Flowchart showing the input-output model using RF

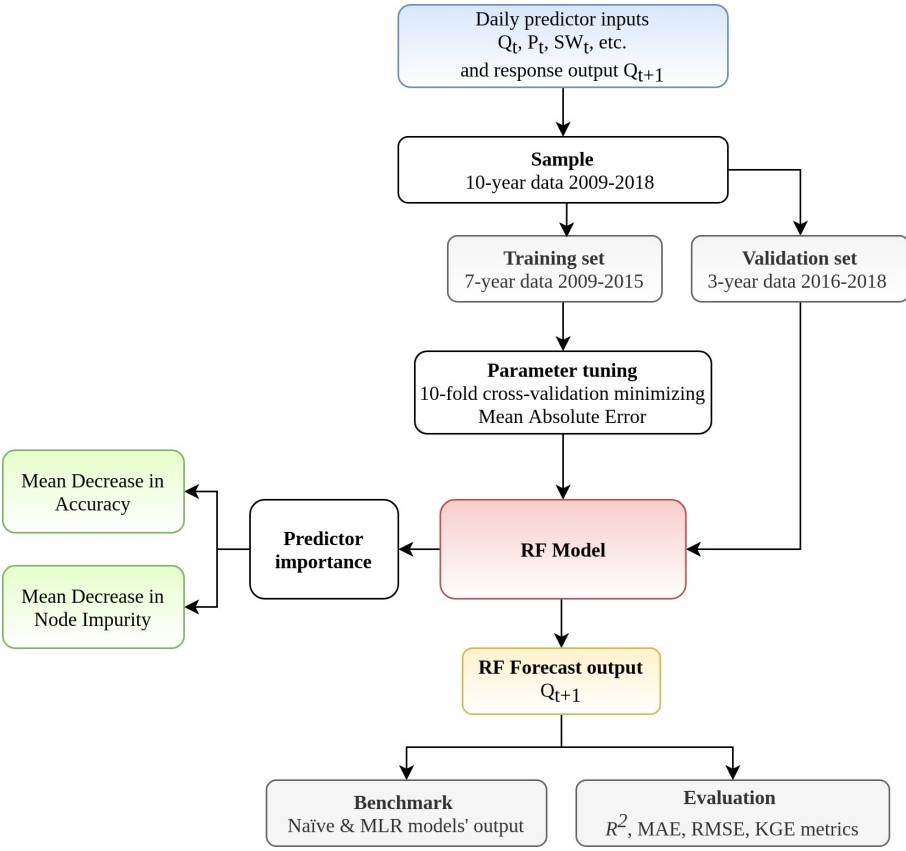

**Figure 5.** Out-of-bag mean absolute error plotted against `mtry` during optimal parameter search at Carbon River Watershed (USGS site 12094000).

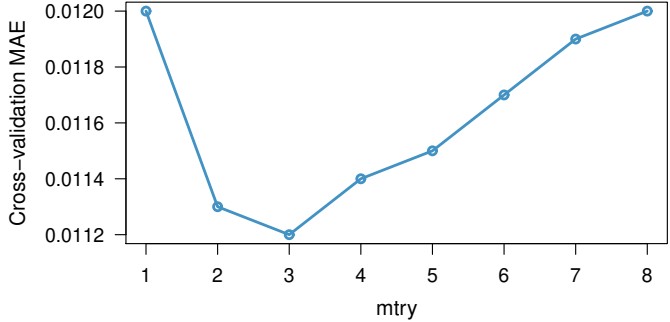

**Table 3.** The optimized parameter `mtry` using exhaustive-search strategy (`mtry` = {1, 2, 6, 7, 8} were considered but not found as the optimal value at any gauge).

| `mtry` | Number of gauges | Median MAE |
|:---:|:---:|:---:|
| 3 | 29 | 0.0127 |
| 4 | 44 | 0.0116 |
| 5 | 13 | 0.0079 |

**Figure 6.** Boxplots for Pearson correlation coefficient between forecasted and observed values for three models: RF, naïve, and MLR across three flow regimes. Two-sample Wilcoxon rank-sum significance tests are performed and p-value (in black) are included for each pair of models.

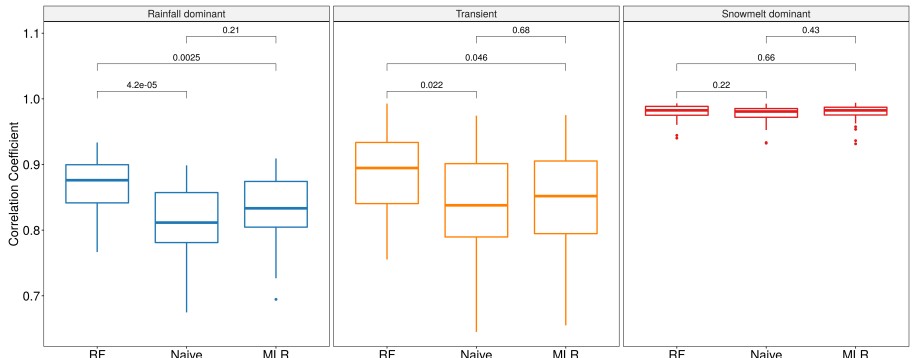

(1-8) were analyzed by 3 repetitions of 10-fold cross validation from the train data set. Averaging the MAE of repetitions of the cross-validation procedure can provide more reliable results as the variance of the estimation is reduced (Seibold et al., 2018). To illustrate, in Fig. 5, lowest cross-validation MAE is obtained at `mtry` = 3 at Carbon River Watershed (USGS Site 12094000). The results of tuning for all gauges (Table 3) show that the optimal `mtry` values are {3, 4, 5} with median MAE of 0.0127, 0.0116, and 0.0079 respectively. The optimal `mtry` at each gauge was then used in both training and validating the model. Because the number of predictors in our study is relatively small, computation burden of the exhaustive search was manageable. As the number of candidate grows, a random search strategy (Probst et al., 2019), in which values are drawn randomly from a specified space, can be more computationally efficient.

## 4.2 Benchmark RF against MLR and naïve models

Figure 6 shows the distributions of Pearson correlation coefficient ($r$) between forecasted and observed values obtained from the three models: RF, naïve, and MLR. Non-parametric, two-sample Wilcoxon rank-sum significance tests (Wilcoxon et al., 1970), which are used to assess whether the values obtained between two separate groups are systematically different from one another, suggest that the pair-wise differences in $r$ values between RF and the other two models are statistically significant ($p < 0.05$) in two flow regimes. RF is observed to outperform both naïve and MLR models in rainfall-driven and transient watersheds. Among snowmelt-driven watersheds, the three models yield similar correlation coefficients ($p > 0.05$). In Fig.

**Figure 7.** Pairwise scatter plots of Pearson correlation coefficient between forecasted and observed values for (a) RF vs. naïve model, (b) RF vs MLR, and (c) MLR vs. naïve model. Each dot represents a watershed (n=86).

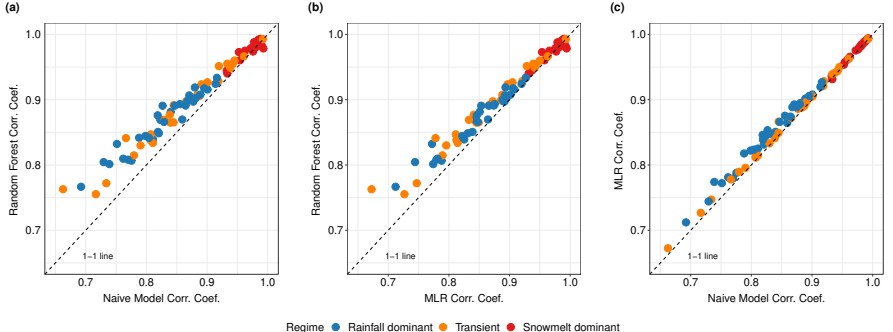

7a, we observe most points lie on the left of the 1-to-1 line, suggesting that RF outperforms naïve model at most individual watersheds in rainfall-driven and transient regimes. We also discern that large improvement, defined as the positive difference in $r$ values between RF and naïve model, tends to occur with lower persistence (lower $r$ values from the naïve model). This suggests that application of RF would be most benefiting at watersheds where next-day streamflow is less dependent on the condition of the current day. Among snowmelt-driven watersheds, the data points lie on the 1-to-1 line, indicating that the three models show marginal difference in $r$ values. As Mittermaier (2008) pointed out, the choice of reference can affect the perceived performance of the forecast system. Our pair-wise comparisons highlight the fact that evaluating data-driven models should be performed in consideration of the autocorrelation structure in the data (Hwang et al., 2012). Without accounting for persistence, it would be inadequate to conclude that RF gives better performance in snowmelt-driven watersheds. Nevertheless, we observe RF outperformed MLR in all rainfall-dominated and transitional watersheds and 19 out of 25 snowmelt-dominated watersheds. The median $r$ values for RF in the three groups are (0.88, 0.89, 0.98) compared to (0.85, 0.87, 0.98) for MLR. This may reflect RF's better ability to capture non-linear relationship between streamflow and other variables.

## 4.3 Evaluation of RF overall performance

We next evaluated the overall performance of RF across three flow regimes using four criteria: $R^2$, KGE, MAE, and RMSE (Table 4, Fig. 8). Here, we observe a similar trend in $R^2$, KGE, MAE, and RMSE scores compared to $r$-value trend in Fig. 6, where RF performs better in snowmelt-dominated than in rainfall-dominated (higher $R^2$ and KGE, lower MAE and RMSE). Snowmelt-dominated watersheds have the smallest range of $R^2$ values across the three groups. This may suggest that there is less variability in flow behaviors at individual gauges in this group and is consistent with the observed data where the hydrographs of snowmelt-driven watersheds tend to be less flashy compared to rainfall-driven watersheds. Not surprisingly, transitional group has the largest spread in $R^2$ values as watersheds in this group share characteristics from the other two groups.

Because RMSE is more sensitive to larger errors compared to MAE, the difference between the two scores represents the extent in which outliers are present in error values (Legates and McCabe Jr, 1999). In rainfall-driven and transient groups, the

**Figure 8.** Streamflow daily forecast scores computed over the validation period for RF model in four metrics: R-squared, KGE, MAE, and RMSE.

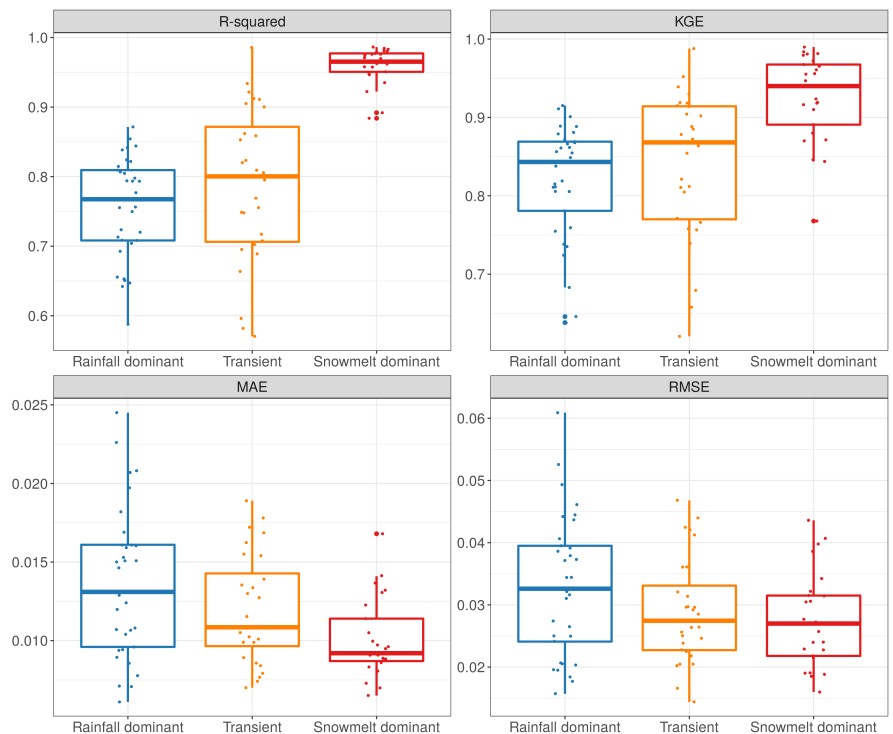

**Table 4.** Descriptive statistics of the four criteria used to evaluate the overall performance of RF: $R^2$, KGE, MAE, and RMSE.

| Metric | Flow regime | Min | Q1 | Median | Q3 | Max |
|---|---|---|---|---|---|---|
| $R^2$ | Rainfall-dominated | 0.59 | 0.71 | 0.77 | 0.81 | 0.87 |
| | Transient | 0.57 | 0.71 | 0.80 | 0.87 | 0.99 |
| | Snowmelt-dominated | 0.88 | 0.95 | 0.97 | 0.98 | 0.99 |
| KGE | Rainfall-dominated | 0.64 | 0.78 | 0.84 | 0.87 | 0.92 |
| | Transient | 0.62 | 0.77 | 0.86 | 0.91 | 0.99 |
| | Snowmelt-dominated | 0.77 | 0.89 | 0.94 | 0.97 | 0.99 |
| MAE | Rainfall-dominated | 0.0061 | 0.0096 | 0.0131 | 0.0161 | 0.0245 |
| | Transient | 0.0070 | 0.0097 | 0.0109 | 0.0143 | 0.0189 |
| | Snowmelt-dominated | 0.0065 | 0.0087 | 0.0092 | 0.0114 | 0.0168 |
| RMSE | Rainfall-dominated | 0.0157 | 0.0241 | 0.0326 | 0.0395 | 0.0609 |
| | Transient | 0.0144 | 0.0227 | 0.0275 | 0.0331 | 0.0468 |
| | Snowmelt-dominated | 0.0160 | 0.0218 | 0.0270 | 0.0315 | 0.0436 |

shape of the boxplot distributions remain fairly consistent between the two error scores, suggesting that distribution of large errors is similar to that of mean errors in these watersheds (Fig. 8). The MAE scores are heavily skewed towards 0 while RMSE scores are more evenly spread among snowmelt-driven watersheds. In snowmelt-driven watersheds, we observe a noticeably wider interquartile range (difference between first quartile and third quartile) in RMSE plot compared to MAE plot. This indicates that RF can still be susceptible to underestimation or overestimation in watersheds where the mean error is relatively low.

In Table 4, KGE scores are reported in a range of 0.64–0.99 for all watersheds. The median values for each flow regime are 0.84, 0.87, and 0.94. As observed mean flow is used in the calculation of KGE, Knoben et al. (2019) suggested that a KGE score greater than -0.41 indicates a hydrologic model improves upon the forecast with mean flow, independent of the basin. Therefore, RF can be seen to give satisfactory performance at all watersheds in our study. Our results are comparable to findings in (Tongal and Booij, 2018) where authors compare the performance of RF, SVM, and ANN to simulate daily discharge with baseflow separation at four rivers in California and Washington. Although authors did not classify these basins, it can be inferred that three of the rivers were rainfall-driven and one was snowmelt-driven. RF model in their study produced KGE scores of 0.41, 0.81, and 0.92 for the rainfall-driven water basins (without baseflow separation). However, our KGE scores for snowmelt-fed watersheds (with a median of 0.94) are higher compared to the reported 0.55 in their study.

## 4.4 RF performance on extreme streamflows

We also examine the model's capacity to forecast extreme events because of their potential high impact and associated flood risks in this region. Ability of RF to correctly detect extreme flows exceeding $90^{th}$, $95^{th}$, and $99^{th}$ percentile thresholds (defined as the POD) for each watershed are plotted against the FAR in Fig. 9. A threshold point falling below the no-skill line indicates the model yields higher FAR than POD and is considered to have no predictive power for that threshold. RF becomes expectedly less skilful in its forecasts with increase in magnitude of the events. The model tends to perform better among snowmelt-dominated watersheds (higher POD, lower FAR) compared to those in transient and rainfall-driven groups. At the $95^{th}$ threshold, RF can forecast correctly at least 50 percent of the extreme events (POD > 0.5) at most watersheds. At the $99^{th}$ threshold, the difference in RF's ability to forecast extreme streamflow among the three flow regimes becomes less obvious. In snowmelt-driven watersheds, 8 out of 25 have POD > 0.5, 9 have POD between 0.01 and 0.5, and 8 have a POD of 0. While few studies have examined complex diurnal hydrologic responses in high-elevation catchments (Graham et al., 2013), our particular result suggests large surges in streamflow sustained by spring and early summer snowmelt can be difficult to predict, even at 1-day lead time, and is an ongoing research subject (Ralph et al., 2014; Cho and Jacobs, 2020). In our study, we observe high POD is accompanied by low FAR for the same threshold. This may suggest that RF is skillful in its forecasts of extreme events.

## 4.5 Analysis of variable importance

Variable importance is a useful feature in both understanding the underlying process of current model and generating insights for selection of variable in future studies (Louppe et al., 2013). RF quantifies variable importance through two measures: MDA

**Figure 9.** The probability of detection (POD) plotted against the false alarm rate (FAR) for three extreme thresholds: $90^{th}$, $95^{th}$, and $99^{th}$ percentiles. Thin black line connects values from the same watershed. (Vertical axis) Number of times RF *correctly* forecasted events that exceeded the threshold divided by the total number of exceedance. (Horizontal axis) Number of times RF *incorrectly* forecasted events that exceeded the threshold divided by the total number of non-exceedance. It is noted that the scales of the horizontal and vertical axes are not 1-to-1 in the plotted partial receiver operating characteristic (ROC) curve.

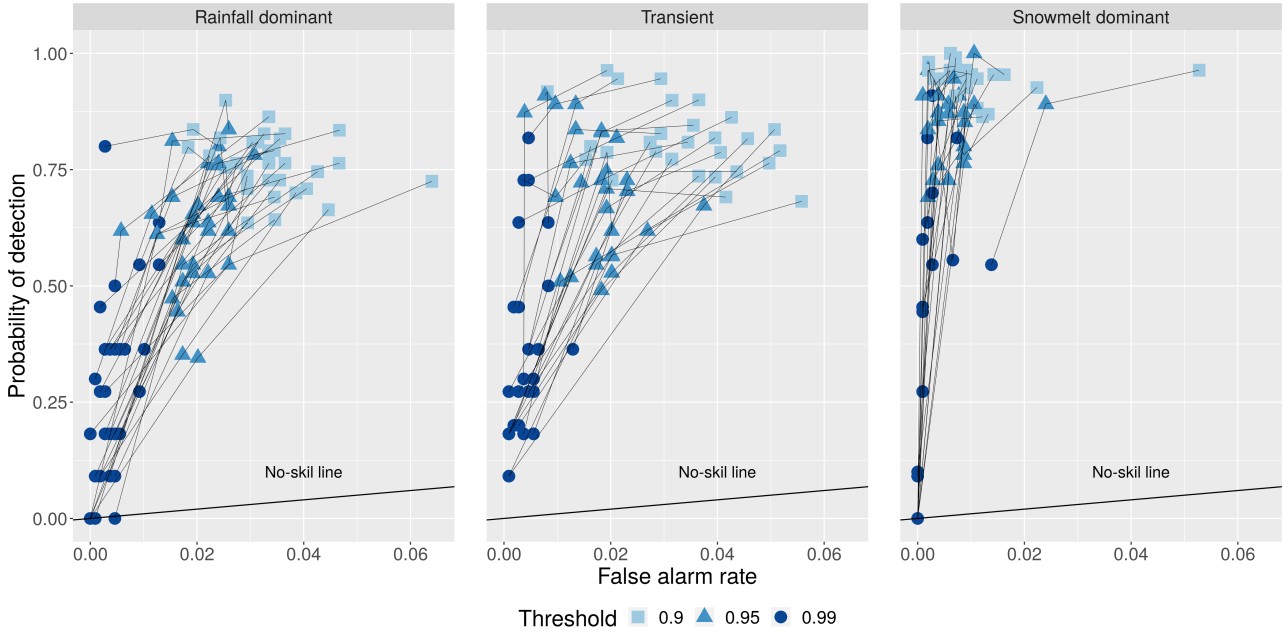

**Figure 10.** Barplots show importance of predictor variables using (a-c) MDA and (d-f) MDI criteria. Length of the blue bars indicates the median value across the watersheds for each flow regime and the thin black bar represents the full range of the values.

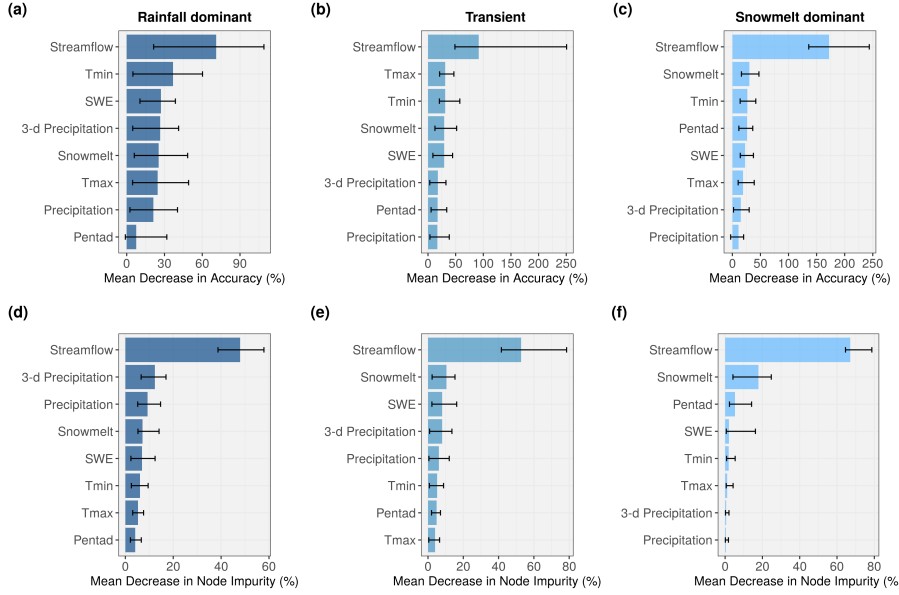

and MDI (Fig. 10). In both measures, the higher value indicates variable contributes more to the model accuracy. Intuitively, streamflow from previous day is shown to be the most importance variable due to persistence. This is reflected across three flow regimes and two measures. We also observe the sum of 3-day precipitation tends to have more predictive power than than 1-day precipitation. Maximum temperature and minimum temperature share similar contribution where minimum temperature tends to receive slightly higher scores. Among snowmelt-dominated watersheds (Fig. 10c and 10f), we anticipate snow indices ($SD_t$ and $SWE_t$) contribute more in the prediction than precipitation and this is also reflected. Surprisingly, pentad comes third and fourth in MDI and MDA respectively. This supports the long-term snowpack memory of daily streamflow (Zheng et al., 2018) and can be useful in real-time prediction. Precipitation does not seem to have significant contribution to the model's accuracy among the snowmelt-dominated watersheds. Although PRISM precipitation data includes both rainfall and snowfall, it is likely that the majority of fallen precipitation in these high-altitude watersheds is stored as snow on the surface and does not immediately contribute to runoff. Li et al. (2017) estimated that 37 % of the precipitation falls as snow in western US, yet snowmelt is responsible for 70 % of the total runoff in mountainous areas. It is still very surprising to observe such low contribution of precipitation variable to RF model accuracy. Nevertheless, we observe general agreement between the two measures in ranking of the variables in snowmelt-driven group.

In transient and rainfall-dominated groups, there is noticeable disagreement between the two criteria. Precipitation ($P_t$) and 3-day precipitation ($P3_t$) tend to rank lower in MDA measure (Fig. 10a and 10b) compared to MDI (Fig. 10d and 10e). Specifically, in rainfall-dominated group, 3-day precipitation and precipitation are placed 2[nd] and 3[rd] based on median MDI compared to 4[th] and 7[th] in MDA. Maximum and minimum temperatures, on the other hand, tend to be more important in MDA calculation compared to in MDI. In Shortridge et al. (2016), RF model was used to predict streamflow at five rain-fed rivers in Ethiopia. Similarly calculated MDA in that study suggested precipitation was less important (7.71 %) than temperature (12.74 %). Linear model in the same study, however, considered the coefficient for precipitation to be significant ($p << 0.01$) while temperature coefficient was not ($p = 0.08$). In Obringer and Nateghi (2018), the authors predicted daily reservoir levels in three reservoirs in Indiana, Texas, and Atlanta using RF and other ML techniques. Precipitation was reported as the least important variable and ranked behind dew point temperature and humidity. Inspecting the probability density functions of our predictors, we suspect that for variables that are heavily skewed and zero-inflated (e.g., precipitation), permutation-based MDA may underestimate their importance compared to those that are more normally distributed such as maximum and minimum temperatures. In our precipitation data (both training and validation), at least 30 percent of the daily observations are zeros across the watersheds. There is a high likelihood that the day with zero precipitation ends up with the same value during the shuffling process, thus potentially affecting the randomness created to compute MDA. While we did not perform additional simulation to further confirm whether MDA and MDI measures are sensitive to highly-skewed and zero-inflated variables, this can be a topic of future research. Strobl et al. (2007), however, showed RF variable importance measures can be unreliable in situations where predictor variables vary in their scale of measurement. It is noted that the scale of measurement does not only refer to the numeric range but also the nature of the data (e.g., ordinal vs. continuous). Among our 8 predictors in our study, pentad is considered an ordinal variable. Also, the scales of measurement of precipitation and temperature variables are slightly different. Precipitation is a flux variable and comprises discrete and continuous components in that if it does not rain the

amount of rainfall is discrete whereas if it rains the amount is continuous. Temperature is a state variable and always continuous. Temperature predictors receiving higher MDA can also be due to identified bias where permutation-based importance measures overestimates the true contribution of correlated variables (Gregorutti et al., 2017). In our study, temperature variables tend to have more correlation with other predictors than do the two precipitation variables. This is likely because temperature controls both the form of precipitation (snowfall vs. rainfall) and the timing of snowmelt. There is also an ongoing discussion regarding the stability of both measures, in which the two variable importance measures can yield noticeably different rankings, in simulated datasets (Calle and Urrea, 2010; Nicodemus, 2011; Ishwaran and Lu, 2019). Although results from MDI make more sense in our case, we suggest RF users to exert caution when interpreting outputs from these two measures.

## 4.6 Effects of watershed characteristics on model performance

To explore the role of catchment characteristics such as geology, topography, and land cover on the performance of RF model, we perform Pearson correlation test between the KGE scores and selected basin physical characteristics for each flow regime. These watershed characteristics were compiled as part of GAGES-II dataset using national data sources including US National Land Cover Database (NLCD) 2006 version, 100 m-resolution National Elevation Dataset (NED), and Digital General Soil Map of the United States (STATSGO2) (Table S1 in the Supplement). The results are shown in Table 5. There is a strong negative correlation ($p < 0.05$) between KGE scores and watershed slopes among rainfall-dominated and transient watersheds (Fig. 11a). As steeper hillslope often associates with faster surface and subsurface water movement during event-flow runoff, this can result in shorter response time. We observe a similar trend between KGE scores and percent of sand in the soil (Fig. 11b) where the RF performs worse in watersheds with higher hydraulic conductivity (i.e., higher sand content). This could be a result of rapid subsurface flow from soil profile enabled by soil macropores in mountainous forested area (Srivastava et al., 2017), where subsurface flow is the predominant mechanism. Without a quantification of the partition of discharge into surface flow and subsurface flow at individual watersheds, it is difficult to determine the relative importance of subsurface runoff mechanisms in regulating streamflow and how that may have affected the RF performance. The findings, however, suggest RF performance can deteriorate at watersheds with quick-response runoff when supplied with 1-day delayed observation data.

It appears that stream density and the amount of vegetation cover may also affect the performance of RF, but the relationships are not statistically significant at $\alpha = 0.05$. Aspect eastness, drainage area, and basin compactness are not determining factors to variability in the KGE scores. We also explored the impact of land-use and land-cover, which can be represented by the extent of impervious cover in each watershed. However, because we only selected unregulated watersheds that experienced minimal human disruption during the initial screening, most watersheds have very little impervious cover (less than 5 %). It is noted that these selected characteristics are not meant to be exhaustive, but rather representative of various types of factors that could help explain the variability in model performance. Furthermore, an alternative approach to Pearson's correlation is to use ANOVA to test for marginal significance of each catchment variable to KGE while accounting for their interaction. Because our objective is not to make inference on KGE based on these variables and ANOVA analysis can be complicated to interpret, we choose to compute correlation coefficient.

**Figure 11.** KGE scores plotted against (a) the average percent of slope and (b) the average percent of sand in soil at each watershed. Best-fit lines were determined using simple linear regression. Pearson correlation coefficients were computed with associated significance.

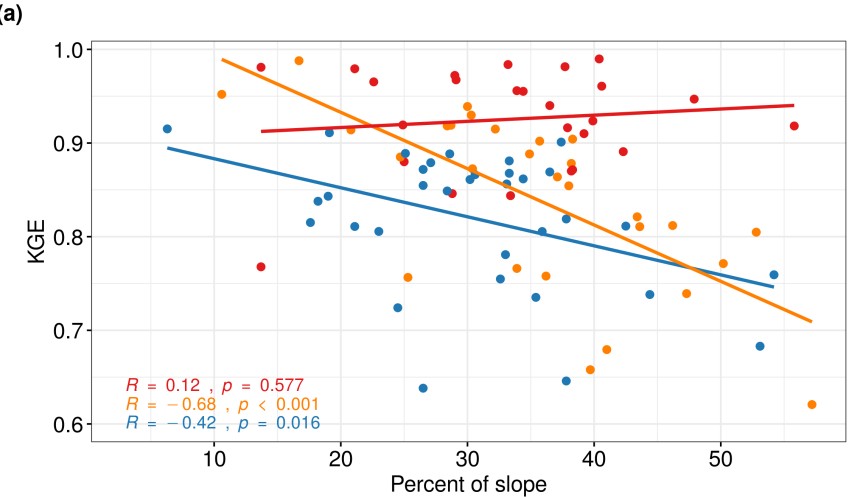

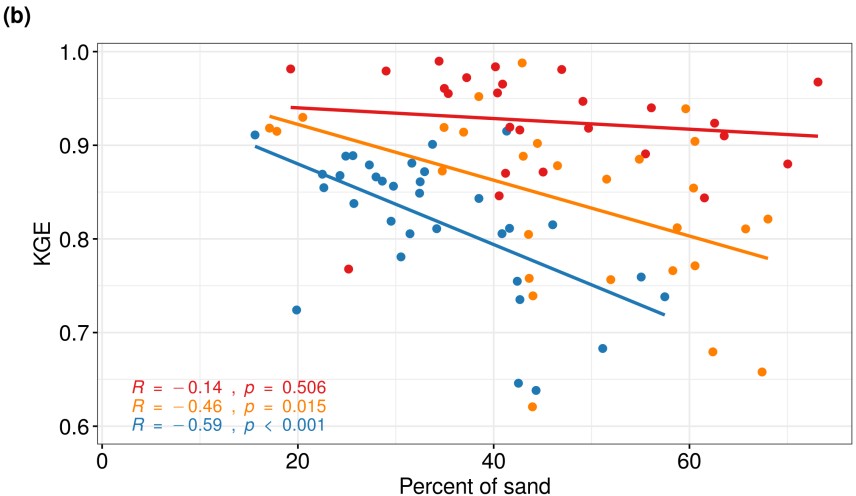

**Table 5.** Pearson correlation coefficient between KGE scores and selected basin physical characteristics. Highlighted red values indicate the relationship is significant at 5 percent or 1 percent level.

| Watershed characteristics | Hydrologic regime | | |
|---|---|---|---|
| | Rainfall dominant | Transient | Snowmelt dominant |
| Slope | -0.42 | -0.68 | 0.12 |
| Aspect eastness | -0.02 | 0.12 | -0.12 |
| Drainage area | 0.14 | -0.12 | 0.11 |
| Basin compactness | 0.09 | -0.12 | -0.16 |
| Stream density | -0.10 | 0.29 | -0.27 |
| Percent of sand | -0.59 | -0.46 | -0.14 |
| Percent of forested area | -0.11 | 0.32 | 0.32 |

## 4.7 Limitations and future research

There are some notable limitations in our study as well as RF in general. The classification of watersheds into three flow regimes was based on the timing of the climatological mean of the annual flow volume, which can fluctuate from year to year. This is particularly true for the watersheds in the transient group where streamflow is contributed by a mixture of runoff from winter rainfall and springtime snowmelt and the inter-annual variability is tremendous in both magnitude and timing (Lundquist et al., 2009). Therefore, the membership of the classified watersheds from this group can vary. In fact, Mantua et al. (2009) discussed the future shift of transient runoff watersheds towards rainfall-dominated in Washington State. Because we trained RF using the same input variables for all watersheds regardless of flow regimes and calculated performance criteria separately, the classification does not alter the results at individual watershed.

In the study, we used estimated precipitation from PRISM, which is an interpolation product and combines data from various rain gauges from multiple networks. Despite possible introduced errors and uncertainty, we believe the use of spatially distributed product better represents the areal estimation of precipitation over the watershed than a single rain gauge measurement. In real-time forecast, this would be not be feasible due to the added time to compile and process such data. Similarly, we provided RF model with a basin-average SWE from SNOTEL stations as an estimate of snowpack condition. Using a more spatially consistent SWE data such as the Snow Data Assimilation System (Pan et al., 2003) product would potentially improve model accuracy. As our results indicate that RF can produce reasonable forecasts, potential future research could explore the sensitivity of the model using satellite derived snow products and even include $t+1$ precipitation forecast as a predictor in the model.

An inherent limitation of RF is the lack of direct uncertainty quantification in prediction. In our case, the forecasted streamflow using RF does not yield a standard error comparable to that provided by traditional regression model, and hence no way to provide probabilistic confidence intervals on predictions. Methods to estimate confidence intervals have been proposed by Wager et al. (2014), Mentch and Hooker (2016), and Coulston et al. (2016), but they are not widely applied. For future work, computation of confidence interval in RF prediction will be useful in addressing and understanding uncertainty.

## 5 Conclusions

Accurate streamflow forecast has extensive applications across disciplines from water resources and planning to engineering design. In this study, we assessed the ability of RF to make daily streamflow forecasts at 86 watersheds in the Pacific Northwest Hydrologic Region. Key results are summarized below:

- Based on the KGE scores (ranging from 0.62 to 0.99), we show that RF is capable of producing skilfull forecasts across all watersheds.

- RF performs better in snowmelt-dominated watersheds, which can be attributed to stronger persistence in the streamflow time series. The largest improvements in forecast compared to naïve model are found among rainfall-dominated watersheds.

- The two approaches for measuring predictor importance yield noticeably different results. We recommend interpretation of the these two measures should be coupled with understanding of the physical processes and how these processes are connected.

- Increase in steepness of slope and amount of sand content are found to deteriorate RF performance in two flow regime groups. This demonstrates catchment characteristics can cause variability in performance of the model and should be considered in both predictor selection and evaluation of the model.

Considering the current and future vulnerabilities of the Pacific Northwest to flooding caused by extreme precipitation and significant snowmelt events (Ralph et al., 2014), skillful streamflow forecasts can have important implications. Due to its practical applications, RF and RF-based algorithms continue to gain popularity in hydrological studies (Tyralis et al., 2019). Given the promising results from our study, RF can be used as part of an ensemble of models to achieve better generalization ability and accuracy not only in streamflow forecast but also in other water-related applications in this region.

*Code and data availability.* Example code for building random forests model in R and data are available at https://github.com/leopham95/RandomForestStreamflowForecast

*Author contributions.* **Leo Pham**: Conceptualization, Data curation, Formal analysis, Funding acquisition, Investigation, Methodology, Project administration, Software, Validation, Visualization, Writing - original draft. **Lifeng Luo**: Conceptualization, Investigation, Methodology, Funding acquisition, Supervision, Project administration, Resources, Writing - original draft. **Andrew Finley**: Resources, Supervision, Funding acquisition, Writing - original draft.

*Competing interests.* The authors declare that they have no conflict of interest

*Acknowledgements.*  Leo Pham was supported by the Algorithms and Software for SUpercomputers with emerging aRchitEctures Fellowship funded by National Science Foundation Grant No.1827093. We wish to express deep gratitude to the researchers at the National Supercomputing Center in Wuxi, China and Tyler Willson at Michigan State University for the inital brainstorming and project development. Luo's effort was partially supported by the national science foundation (NSF-1615612).

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
