# Peer review of "Evaluation of random forests for short-term daily streamflow forecasting in rainfall and snowmelt-driven watersheds"

_Hydrology and Earth System Sciences, 2020_

## Referee Comment (RC1) · Francesco Avanzi (Referee) · 1 Aug 2020

Pham et al. developed a Random-Forest-based (RF) algorithm for day-ahead stream-flow forecasting. The algorithm employs 8 weather-snow-time features (see Table 2) and was tested across 86 watersheds in the Pacific Northwest (PNW), where it was compared with multilinear regression and previous-day streamflow as a minimum-information forecasting approach (so called Naïve method). Results show that RFs provide quite robust predictions across catchments with different climatology (rainfall dominated, snowfall dominated, or transient) and generally perform better than the two benchmark approaches – especially in rainfall-dominated and transient watersheds.

[Figure]

Drops in accuracy for RFs were correlated with watershed slope and sandiness.

This is an interesting, well-written, and concise paper about an emerging machine-learning technique in hydrology and its use for streamflow forecasting. While I have tangential knowledge of RF technical issues, I found the description of the algorithm clear and rigorous, which facilitates replicability and ultimately allows readers to learn more about RFs in general rather than only looking at is as a black box (note that these technical details are often bypassed or heavily summarized in other papers I have read on this matter). Also, RFs and machine-learning approaches in general are on the rise in hydrology, meaning I expect the paper to have some impact on the community. There are still some major and minor comments that I recommend for authors (see below), and I recommend the editor reconsider this manuscript after minor revisions.

MAJOR COMMENTS

1. The manuscript sometimes reads like a technical note, as it describes the algorithm and its implementation in great details but ultimately falls a little short on hydrological-process interpretation. I see that the main goal of the paper is testing an algorithm, and applied research is certainly within the scope of HESS. And yet, I feel like implementing RFs across 86 watersheds with different characteristics and 10 yrs with different climatology without looking more specifically at how performance changes across the landscape and between years with different characteristics is kind of a missed opportunity. For example, operational forecasters in the western US often calibrate multiple forecasting tools based on the concept of "water-year type" (for example, a set of parameters for dry yrs, another for wet yrs, etc.). Doing so here would allow authors to explore how predictive skills change between dry and wet yrs, or yrs with more or less snow or more or less rainfall, which would also have some future-climate implications. I ignore how demanding it would be to add an additional calibration experiment like this, so these were only examples. My bottom line is that I encourage authors to better explore their results from a hydrological-process perspective to add some process-based insights to an already interesting paper focused on an algorithm.

2. I was a little surprised by the choice of benchmark models and particularly by the fact that authors did not consider a full hydrologic model. I understand that authors would probably like to stay within the realm of data-driven models, but a Naïve approach looks very simplistic, especially at a daily time scale and in basins where rainfall and snowfall coexist. How can this approach predict, e.g., intense rain-on-snow events that are ubiquitous in the PNW? Most flood-forecasting tools I have been exposed to use full hydrologic models, and I encourage authors to at least discuss this matter in their manuscript.

3. Relatedly, I was also a little surprised that authors did not consider rainfall and snowfall as separate features in their model (see their Table 2). I am aware that PRISM only provides total precip, but it also provides temperature and relative humidity that could be employed to separate snowfall from rainfall. Perhaps considering SWE already makes up for this, but I encourage authors consider this at least for future work. This was again particularly puzzling to me given the well-known role of rain on snow in this region.

SPECIFIC COMMENTS

- Title: I have usually seen "rainfall-dominated" and "snowfall-dominated" being used, rather than rainfall and snowmelt driven. Consider revising.

- Line 3: I think "ease of application" might be relative, especially in ungauged areas or for users with limited computational capabilities. Consider revising or expanding.

- Line 8: transisent -> transient?

- Line 11: better than what?

- Line 24: I believe even ML algorithms need the formulation of some mathematical equations, although maybe not in a predictive role.

- Line 25: I am not sure if ML algorithms really require "fewer data" than, e.g., conceptual, minimal hydrologic models. Here again, a comparison with a hydrologic model

would be a great addition to the paper.

- Line 38ff: maybe also mention glaciers here, although they might not be an important driver for hydrology in your study region.

- Line 56: indeed, statistical forecasting models are widely used across the western US to predict summer flow (e.g., April to July total runoff). I understand this is out of the scope of your paper, but maybe mention this application to provide broader framing to your work.

- Algorithm 1 Step 3: I believe the case $X_i = t$ is missing, contrary to Figure 1.

- Section 2.2: maybe redefine acronyms for MDA and MDI here since you introduce them in the Introduction. This will be greatly appreciated by diagonal readers.

- Section 2.3: see my major comment 2.

- Line 149: Knoben at al. (https://hess.copernicus.org/articles/23/4323/2019/) have recently pointed out that KGE = 0 has a different implication from NSE = 0, and so KGE = 0 should be used with caution. Please revise as relevant.

- Line 172 and Table 2: please use SI rather than customary units.

- Line 176: maybe some more quantitative climatology would be more appropriate here. For instance, replace "ample amount of winter precipitation" with statistics of winter precip for your watersheds. Same for "mild temperature". It would also be interesting to provide some statistics of mean-max SWE across the basins.

- Line 186: have you tried to impute missing values? What's the impact of gaps in your framework?

- Line 196ff: how "place-based" is this classification based on the day of the water year? I would have expected one to classify basins based on proportion of rainfall over total precipitation, which look more general to me.

- Section 3.2.3: it has been reported that snow pillows have a certain bias in capturing the onset of snowmelt, basically because they isolate the overlying snowpack from the ground. I understand they are the only continuous-time data source to estimate snowmelt, but consider adding a warning about this bias here.

- Line 244: I believe at least some SNOTEL stations do measure soil moisture. Please specify this if relevant.

- Line 246: how were the validation and calibration period chosen? May this choice have played a role in your results? What were the climatological characteristics of these two periods? Please expand and support your choice here.

- Line 269: I might have missed this, but do you show any statistics of persistence for your catchments to support this statement? Again, I may be missing something here.

- Line 292: see my previous point regarding KGE

- Line 307: may these be due to rain on snow?

- 355: streamflow (typo)

- Figure 9: consider adding the scatter plot for slope

- Table 1: is there any reason why snowmelt-driven catchments have a larger range of drainage areas? Just out of my curiosity.

-Table 2: please add t or t -1 as relevant in the "predictors" column; you only did that for predictor 1.

- Table 5: what is the data source for these characteristics? Especially sandiness and forested area.

[Figure]

---

## Referee Comment (RC2) · Anonymous Referee #2 · 12 Aug 2020

I have included my review in the attached zip file.

Please also note the supplement to this comment:
https://hess.copernicus.org/preprints/hess-2020-305/hess-2020-305-RC2-supplement.zip

---

## Author Comment (AC1) · 3 Sep 2020

The authors would like to thank the reviewer for his informative and valuable comments. We appreciate the opportunity to provide further clarification and improve our manuscript for final submission and consideration. We included below the review's comments (italic black font) and our responses (plain blue font).

**1 Overall comments**

*Pham et al. developed a Random-Forest-based (RF) algorithm for day-ahead streamflow forecasting. The algorithm employs 8 weather-snow-time features (see Table 2) and was tested across 86 watersheds in the Pacific Northwest (PNW), where it was compared with multilinear regression and previous-day streamflow as a minimum information forecasting approach (so called Naïve method). Results show that RFs provide quite robust predictions across catchments with different climatology (rainfall dominated, snowfall dominated, or transient) and generally perform better than the two benchmark approaches – especially in rainfall-dominated and transient watersheds. Drops in accuracy for RFs were correlated with watershed slope and sandiness. This is an interesting, well-written, and concise paper about an emerging machine learning technique in hydrology and its use for streamflow forecasting. While I have tangential knowledge of RF technical issues, I found the description of the algorithm clear and rigorous, which facilitates replicability and ultimately allows readers to learn more about RFs in general rather than only looking at is as a black box (note that these technical details are often bypassed or heavily summarized in other papers I have read on this matter).*

Thank you for the review and supportive and constructive feedback. Yes, we too thought it would be helpful to provide this level of detail about the regression algorithms.

*Also, RFs and machine-learning approaches in general are on the rise in hydrology, meaning I expect the paper to have some impact on the community.*

Yes, we agree, these underused tools offer some exciting opportunities in this field. Tyralis et al. [2019] reviewed and compiled the applications of RF in water resources, where the number of papers has risen exponentially in the period between 2000 and 2019 (Figure 2). The authors of this study also acknowledged in their conclusion that, "it is quite remarkable that only a few studies recognize possible shortcomings of random forests and their variants." The analysis of the two commonly used measures of predictor importance(permutation-based Mean Decrease in Accuracy and Gini-based Mean Decrease in Node Impurity) in our manuscript suggests they might not be reliable and such analysis should be coupled with the understanding of the physical processes under study. We believe identifying and providing evidence for this shortcoming constitute a contribution to the current literature and for future research on the applicability of RF.

*There are still some major and minor comments that I recommend for authors (see below), and I recommend the editor reconsider this manuscript after minor revisions*

We appreciate the comments. We clarified and revised the manuscript according to your suggestions.

**2 Major comments**

*1.The manuscript sometimes reads like a technical note, as it describes the algorithm and its implementation in great details but ultimately falls a little short on hydrological process interpretation. I see that the main goal of the paper is testing an algorithm, and applied research is certainly within the scope of HESS. And yet, I feel like implementing RFs across 86 watersheds with different characteristics and 10 yrs with different climatology without looking more specifically at how performance changes across the landscape and between years with different characteristics is kind of a missed opportunity.*

In the manuscript, we tried to maintain a focus on (1) implementating and evaluating Random Forest for 1-day streamflow forecast and (2) exploring how the variations in hydrological processes across watersheds (e.g., contributions of rainfall and snowfall to streamflow, physical factors) might affect the model performance. In the Introduction, we provided an synopsis on the climatology of the Pacific Northwest and the roles of snowmelt and rainfall in driving streamflow dynamics. In Section 3.1 Study Area and Data, we described the regional hydrographic systems of the Columbia River Basin. We also included physical characteristics of the watersheds under study, which were compiled and documented in the GAGESII Dataset (https://water.usgs.gov/lookup/getspatial?gagesII_Sept2011) and available in Supplementary material. We further systematically classified the watersheds into three

hydrologic regimes: snowmelt-driven, rainfall-driven, and transient. As seen in Figure 2, the majority of the rainfall-dominated and transient watersheds are located on the west side of the Cascades while snowmelt-dominated watersheds are east of the Cascades and at higher elevation (Table 1). We reported the performance of RF for each watershed and explored how factors such as drainage area and slope could affect model performance in Section 4.6. Wenger et al. (2009) simulated and evaluated runoff at 55 watersheds in the Pacific Northwest using variable infiltration capacity (VIC) model and followed a similar classification scheme. Summary statistics of individual watersheds were reported similarly in Table A1 (`doi:10.1029/2009WR008839`). While we also thought about exploring the inter-annual and seasonal variability in model performance, reporting multiple sets of evaluation metrics (e.g., KGE, $R^2$, RMSE, MAE) for 86 watersheds would become confusing and dilute the scope of the paper.

*I was a little surprised by the choice of benchmark models and particularly by the fact that authors did not consider a full hydrologic model. I understand that authors would probably like to stay within the realm of data-driven models, but a Naïve approach looks very simplistic, especially at a daily time scale and in basins where rainfall and snowfall coexist.*

We did not consider a full hydrologic model and instead selected Naïve model as a benchmarking model for two reasons. First, as we briefly mentioned in the Introduction, both data-driven and hydrologic models have their advantages and disadvantages, and yet our objective was not to show one is superior to the other. Safeeq et al. [2014] provides an assessment of Variable Infiltration Capacity (VIC) model for predicting hydrologic regimes of 217 watersheds in the Pacific Northwest at 1/16 degree (6km×6km) grid-scale resolution. The author discussed the large underestimation of simulated SWE from meteorological data forcing compared to that at SNOTEL sites, which could be contributing to model bias. As we supplied RF with SNOTEL data, a comparison on runoff forecast between two models would be inappropriate. Second, we showed that, to our surprise, RF was not able to beat a simple Naïve model in its forecast among snowmelt-driven watersheds with strong persistence (Figure 5). We observed in some other papers where multiple ML models were tested and compared against one another without a baseline model. We wondered how much of the prediction were actually due to persistence in streamflow. As Pappenberger et al. [2015] pointed out, "benchmarking with simpler models can be viewed as a gain-based approach." For this reason, we believe the use of the Naïve model is justified.

*How can this approach predict, e.g., intense rain-on-snow events that are ubiquitous in the PNW?*

While this is a relevant question given the climatology of the PNW and the potential risk of rain-on-snow (ROS) floods, ROS events themselves are quite complex hydrometeorological phenomenon. Since we supplied RF with 1-day antecedent streamflow, SWE conditions, and meteorological data (precipitation and temperature), we would not expect the model to predict these ROS events. This would require additional future weather forecast information and fall under real-time forecast, which is out of the scope of our approach. We discussed the potential of including $t+1$ precipitation forecast on line 377-383 under Section 4.7 Limitations and future research.

*Most flood-forecasting tools I have been exposed to use full hydrologic models, and I encourage authors to at least discuss this matter in their manuscript.*

Good suggestion. We have now added the following discussion on the application of hydrologic models in this region under Introduction section. *Despite the promising results reported in existing literature, most ML streamflow forecast applications are limited to watersheds where rainfall is the major contributor. In many settings, particularly, non-arid mountainous regions in Western USA, a combination of rainfall and spring snowmelt can drive streamflow [Johnstone, 2011, Knowles et al., 2007]. The amount of snow accumulation and its contribution to discharge also vary among the watersheds [Knowles et al., 2006]. Both watershed-scale hydrologic models and statistical models have been used to access the current and future stream hydrology and associated flood risks [Salathé Jr et al., 2014, Wenger et al., 2010, Tohver et al., 2014, Pagano et al., 2009]. Safeeq et al. [2014] simulated streamflows using the Variable Infiltration Capacity (VIC) model at $1/16°$ and $1/20°$ spatial resolutions and evaluated against observed values from 217 watersheds at at annual and season time scales. The study found the model was able to capture the hydrologic behavior of the study watersheds with reasonable accuracy. Yet authors recommended careful site-specific model calibration using not only streamflow but also SWE data would be expected to improve model performance and reduce model bias. Pagano et al. [2009] applied Z-Score Regression to daily SWE from SNOTEL stations and year-to-date precipitation data to predict seasonal streamflow volume in unregulated streams in Western US. Authors reported the skill of these forecasts was comparable to the official published outlooks. A natural question is whether ML models can produce comparable performance in these watersheds*

*where streamflow contributions come from a mix of snowmelt and rainfall, as well as where snowmelt dominates sources.*

*3. Relatedly, I was also a little surprised that authors did not consider rainfall and snowfall as separate features in their model (see their Table 2). I am aware that PRISM only provides total precip, but it also provides temperature and relative humidity that could be employed to separate snowfall from rainfall. Perhaps considering SWE already makes up for this, but I encourage authors consider this at least for future work. This was again particularly puzzling to me given the well-known role of rain on snow in this region.*

We did consider the need to differentiating falling precipitation as rainfall from snowfall using surface air temperature data from PRISM as suggested. However, there are certain drawbacks of this approach. We expect the shifts in temperature can be dramatic at daily time scale, especially in the mountainous region with complex terrain, and may not be well captured by PRISM data. Moreover, watersheds in this study can span a wide range of elevation, making the determination of threshold temperature difficult. We chose not to explicitly differentiate the precipitation types and alternatively included maximum temperature (Tmax) and minimum temperature (Tmin) recorded at SNOTEL stations as predictors in the model. While it is uncertain to say whether the model was able to pick such signal due to the "black-box" nature of RF, we can see in Figure 8 that Tmin variable is ranked high among Snowmelt-dominant watersheds in variable importance. But this could also an indication of better prediction due to snowpack's sensitivity in temperature shifts rather than rain-snow differentiation.

**3 Specific comments**

We here address the specific comments. Simple typo fixes and clarifications were made directly in the revised manuscript.

*- Title: I have usually seen "rainfall-dominated" and "snowfall-dominated" being used, rather than rainfall and snowmelt driven. Consider revising.*

We used "snowmelt-dominated" and "snowmelt-driven" interexhangably throughout the manuscript. We included here two publications that employed similar usage, *"Soil moisture states, lateral flow, and streamflow generation in a semi-arid, snowmelt-driven catchment"* and *"Climate change impacts on the hydrology of a snowmelt driven basin in semiarid Chile"*.

*- Line 24: I believe even ML algorithms need the formulation of some mathematical equations, although maybe not in a predictive role.*

We agree and deleted "formulation of mathematical equations" from the sentence.

*I think "ease of application" might be relative, especially in ungauged areas or for users with limited computational capabilities. Consider revising or expanding*

The sentence has been revised to, "Among other qualities, the popularity of ML for such applications is due to the methods' competitive performance compared with alternative approaches, relative ease in implementation, and less strict distributional assumption."

*Line 38: maybe also mention glaciers here, although they might not be an important driver for hydrology in your study region.*

It's a good point and by saying, "most ML streamflow forecast applications are limited to watersheds where rainfall is the *major* contributor," we acknowledge there are other sources that can contribute to streamflow.

*- Line 56: indeed, statistical forecasting models are widely used across the western US to predict summer flow (e.g., April to July total runoff). I understand this is out of the scope of your paper, but maybe mention this application to provide broader framing to your work*

We appreciate the recommendation and mentioned a statistical forecasting model along with application of physical models in the Introduction.

*Line 149: Knoben at al. (https://hess.copernicus.org/articles/23/4323/2019/) have recently pointed out that KGE = 0 has a different implication from NSE = 0, and so KGE = 0 should be used with caution. Please revise as relevant.*

Thank you for poiting this out. The following is our revision, "KGE metric ranges between -inf and 1. While there currently is not a definitive KGE scale, Knoben et al. (2019) showed KGE values in the range between -0.41 and 1 indicate the model a model improves upon the mean flow benchmark , which assumes the predicted streamflow values equal to the mean of all observations. Generally, KGE value of 1 suggests the model can perfectly reproduce observations."

*Line 176: maybe some more quantitative climatology would be more appropriate here. For instance, replace "ample amount of winter precipitation" with statistics of winter precip for your watersheds. Same for "mild temperature". It would also be interesting to provide some statistics of mean-max SWE across the basins.*

We updated Table 1 to include summary statistics for mean annual temperature and mean annual precipitation across three hydrologic regimes. We also added the following plot to the manuscript. SWE from SNOTEL stations was calculated at HUC-6 level and min-max statistics would be available in Supplementary.

Figure 1: Gauge locations with color gradient indicating variations in (a) watershed drainage area, (b) mean watershed elevation, (c) mean watershed annual precipitation, and (d) mean watershed annual temperature.

[Figure]

**Table 1.** Number of streamflow gauges used in the study for each flow regime, ranges of mean watershed elevation and drainage area. Complete catchment physical and hydro-climatic characteristics for each watershed can be found in Appendix A.

| Hydrologic regime | Number of gauges | Mean watershed elevation (m) | Drainage area (km²) | Mean annual precipitation (cm) | Mean annual temperature (deg C) |
|---|---|---|---|---|---|
| Rainfall-dominated | 33 | 239 - 1207 | 58 - 703 | 122.0 - 367.0 | 5.4 - 11.5 |
| Transitional | 28 | 813 - 1477 | 58 - 1855 | 63.2 - 314.0 | 4.16 - 8.42 |
| Snowmelt-dominated | 25 | 1349 - 2509 | 51 - 3355 | 58.0 - 177.0 | 0.4 - 6.62 |

*Line 186: have you tried to impute missing values? What's the impact of gaps in your framework?*

In the initial screen, we selected gauges that "have less than 10 percent of missing data". The majority of 86 gauges that were eventually included in the study had continuous record or less than 3 percent missing data. Therefore, we do not think imputation was necessary.

*Line 196: how "place-based" is this classification based on the day of the water year? I would have expected one to classify basins based on proportion of rainfall over total precipitation, which look more general to me.*

This classification scheme were applied to streams in the Pacific Northwest from previous studies and based on the shape of the mean annual hydrograph. We provided examples of three types of hydrographs in Figure 2 (b-d). Classifying the watersheds based on proportion of rainfall over total precipitation is another approach but this also requires differentiating rainfall from snowfall at daily timescale. We addressed a shortcoming with this approach above.

*- Line 246: how were the validation and calibration period chosen? May this choice have played a role in your results? What were the climatological characteristics of these two periods? Please expand and support your choice here.*

Seven years and three years of data were used for training and validating the model, respectively. While there is no clear requirement, this selection is consistent with convention in which training dataset should be larger than validation data set to ensure that the model is exposed to a wide range

of hydrological conditions. We tested the another partition scheme where we reserved 8 years (2009-2016) for training and 2 years (2017-2018) for validating. The KGE scores across the watersheds typically improved within 0.05 margin. However, we felt 2 years of observations (approximately 700 data points) would not be sufficient to evaluate the model and decided to go with the former scheme, which might be more conservative. Moreover, our forecast is 1-day ahead so we don't expect the wet year vs dry-year characteristics, which is defined and takes place at annual time scale, would play a major role on the model's performance.

*- Line 269: I might have missed this, but do you show any statistics of persistence for your catchments to support this statement? Again, I may be missing something here.*

We discussed this at lines 279-285.

*- Line 307: may these be due to rain on snow?*

This is a possibility.

*Figure 9: consider adding the scatter plot for slope*

Thanks for the suggestion. We will add the following scatter plot slope vs. KGE in the revised manuscript.

Figure 2: KGE scores plotted against average percent of slope at each watershed. Best-fit lines were determined using simple linear regression.

[Figure]

*- Table 1: is there any reason why snowmelt-driven catchments have a larger range of drainage areas? Just out of my curiosity.*

We believe this is because the Columbia River and its major tributaries including the largest branch, the Snake River, flow through east of the Cascades and on the Oregon-Idaho border where most of the the snowmelt-dominated watersheds in our study are located.

*- Table 5: what is the data source for these characteristics? Especially sandiness and forested area*

The data from this table was part of the USGS GAGEII dataset, which was cited on line 200 and above. Percentage of sand in soil was estimated from the Department of Agriculture Digital General Soil Map of the United States or STATSGO2 Database. The data can be found in the submitted Supplementary document. Percentage of forested area was estimated using the National Land Cover Database 2006 classification. We added the following clarification to the manuscript, *"These watershed characteristics were compiled as part of GAGESII dataset using national data sources including US National Land Cover Database (NLCD) 2006 version, 100m-resolution National Elevation Dataset (NED), and Digital General Soil Map of the United States (STATSGO2)."*

**References**

J. A. Johnstone. A quasi-biennial signal in western us hydroclimate and its global teleconnections. *Climate dynamics*, 36(3-4):663–680, 2011.

N. Knowles, M. D. Dettinger, and D. R. Cayan. Trends in snowfall versus rainfall in the western united states. *Journal of Climate*, 19(18):4545–4559, 2006.

N. Knowles, M. Dettinger, and D. Cayan. Trends in snowfall versus rainfall for the western united states, 1949-2001. prepared for california energy commission public interest energy research program. *Trends in Snowfall Versus Rainfall for the Western United States, 1949-2001. Prepared for California Energy Commission Public Interest Energy Research Program*, 2007.

T. C. Pagano, D. C. Garen, T. R. Perkins, and P. A. Pasteris. Daily updating of operational statistical seasonal water supply forecasts for the western us 1. *JAWRA Journal of the American Water Resources Association*, 45(3):767–778, 2009.

F. Pappenberger, M.-H. Ramos, H. L. Cloke, F. Wetterhall, L. Alfieri, K. Bogner, A. Mueller, and P. Salamon. How do i know if my forecasts are better? using benchmarks in hydrological ensemble prediction. *Journal of Hydrology*, 522:697–713, 2015.

M. Safeeq, G. S. Mauger, G. E. Grant, I. Arismendi, A. F. Hamlet, and S.-Y. Lee. Comparing large-scale hydrological model predictions with observed streamflow in the pacific northwest: Effects of climate and groundwater. *Journal of Hydrometeorology*, 15(6):2501–2521, 2014.

E. P. Salathé Jr, A. F. Hamlet, C. F. Mass, S.-Y. Lee, M. Stumbaugh, and R. Steed. Estimates of twenty-first-century flood risk in the pacific northwest based on regional climate model simulations. *Journal of Hydrometeorology*, 15(5):1881–1899, 2014.

I. M. Tohver, A. F. Hamlet, and S.-Y. Lee. Impacts of 21st-century climate change on hydrologic extremes in the pacific northwest region of north america. *JAWRA Journal of the American Water Resources Association*, 50(6):1461–1476, 2014.

H. Tyralis, G. Papacharalampous, and A. Langousis. A brief review of random forests for water scientists and practitioners and their recent history in water resources. *Water*, 11(5):910, 2019.

S. J. Wenger, C. H. Luce, A. F. Hamlet, D. J. Isaak, and H. M. Neville. Macroscale hydrologic modeling of ecologically relevant flow metrics. *Water Resources Research*, 46(9), 2010.

---

## Author Comment (AC2) · 16 Sep 2020

Although the reviewer has chosen to reject the manuscript in current form, the authors appreciate the reviewer for his/her valuable comments and suggestions. We would like to take this opportunity to provide further clarification and improve our manuscript. We included below the review's comments (italic black font) and our responses (plain blue font).

**1 Overall comments**

*This paper describes the use of random forests (ensembles of decision trees) to predict streamflow in various basins in the Pacific Northwest at one-day lead time. The authors use two methods to understand the most important predictor variables, and they also investigate the effect of other basin characteristics (not included as predictor variables) on the performance of the random forest. With some improvements this work could be a valuable contribution to the literature, especially given the analyses of predictor importance and confounding variables (basin characteristics not included as predictors). However, at this time I have chosen to reject, due to several major issues with the paper. Major comments are summarized below, and inline comments are attached in a PDF.*

We appreciate the reviewer's kind words on the merit of the manuscript and its contribution to the literature. While we believe there are instances in the manuscript that can be improved, most of them can be addressed with better clarification. Please see the Inline comments section below.

**2 Major comments**

*1. The paper has serious grammatical issues, which make it difficult to follow. I have pointed all grammatical errors in the abstract (see `pham2020_annotations_abstract.pdf`). After the abstract, I have mostly abstained from pointing out every grammatical error. However, the frequency of grammatical errors is approximately the same throughout the paper. I would like to make it clear that I am not rejecting on the basis of grammar alone, but it does make the paper difficult to follow (there are many sentences that I simply do not understand).*

Thanks for carefully reviewing our manuscript. We have revised these errors based on your suggestions and believe the overall language is now clearer and improved.

*2. The explanation of machine-learning methods (the random forest and associated predictor-importance methods) is unclear and contains several false statements. See inline comments for more detail. The explanation is probably not clear enough for readers unfamiliar with ML to follow, and it contains enough false statements that readers familiar with ML will probably be left scratching their heads.*

We believe our description and discussion of random forests algorithm is sufficient given the scope of the paper. We also clarified statements the reviewer considered "false" and supported with cited literature.

*3. No significance-testing. The authors claim that their random forest outperforms the two baseline models (persistence, which they call the "naïve" model, and linear regression), but no significance-testing is conducted to support this claim. Especially for the comparison of the random forest with linear regression, the numbers are close enough (see line 277) that I doubt the differences are statistically significant.*

Our comparisons for the three models using correlation coefficient statistics are consistent with published literature. Specifically, Yaseen et al. [2015] conducted a review of ML models for streamflow forecasting in the period 2000–2015 and noted, "It was also found, the majority of the reviewed articles evaluated based on the correlation coefficient (R) and Root Mean Square Error (RMSE) in addition to other evaluation criteria (e.g., Relative error; RE, Mean Absolute Error; MAE, Mean Square Error; MSE, Nash–Sutcliffe Coefficient; NS, Sum of Square Error; SSE), clearly illustrated in (Fig. 1). The modeling that yields a maximum value of correlation coefficient and minimum values of RMSE and MAE presented a good evaluation of model performance." Also, result of a significance test for the differences between the correlation coefficients does not provide useful information. $R$, in this case, is an evaluation score for the models similar to KGE, RMSE, and MAE. Because we used persistence model as baseline, the positive difference in $R$ values between the two respective models can be interpreted as a gain in forecast skillfulness. It would not make sense to perform a statistical significance test for the difference between two evaluation scores of two models otherwise.

*4. Interpretation of model performance is lacking in detail and contains several confusing statements. See inline comments on lines 274, 283, 287, 293, 295, 298, 302, 304, 308, 309, and 311.*

*We addressed these comments and suggestions below.*

**3 Inline comments**

Simple typos and punctuation errors are corrected directly to the revised manuscript.

**3.1 Abstract**

*Line 1: "Data-driven machine learning" is redundant, since all machine learning is data-driven. Also, "machine learning" should not be capitalized.*

In this sentence, we included "data-driven" to contrast ML approach with the traditional hydrologic models, which are physically based and we later discussed this under the Introduction section.

*Line 3: What do you mean by "performance"? Forecast quality, or computational efficiency?*

We agree this was vague. Following is the revised sentence, "Among other qualities, the popularity of ML models for such applications is due to their relative ease in implementation, less strict distributional assumption, and competitive computational and predictive performance."

*Line 6: This phrase seems to be missing a word. Perhaps you mean many/diverse climatic conditions and physiographic settings?*

We revised it as following, "These watersheds cover diverse climatic conditions and physiographic settings."

*Line 8: What does this mean? (I know you define the term later, but it should be defined at first mention. Alternatively, if you do not want to define jargon in the abstract, you could use a different term.)*

We believe the current description, "timing of center-of-annual-flow volume" is adequate for the purpose of the Abstract.

*Line 12-13: What does this mean? Is 0.62-0.99 good, or bad? Since most readers are probably unfamiliar with KGE, I suggest reporting the other evaluation scores here as well. Alternatively, you could just report percent improvement over the baseline models (since this is what really tells you how good the random forest is).*

Although 0.62 - 0.99 KGE score range would be considered "good", we believe these values should be evaluated comparatively not absolutely. We also did not compute "percent improvement" in our analysis. In the interest of conciseness, we also only reported raw KGE score, which has increasingly been used as a metric to access model performance in hydrology and is more objective in our opinion, in the Abstract.

*Line 15-16: What are the "new insights"? This statement is very generic and could be found in almost any paper abstract, so you should make it a bit more specific.*

We were referring to these insights: (1) "RF performance deteriorates with increase in catchment slope and increase in soil sandiness" and (2) "We note disagreement between two popular measures of RFvariable importance and recommend jointly considering these measures with the physical processes under study."

**3.2 Manuscript body**

*Line 23-24: This is very close to the definition of ML, although you do not explicitly say so.*

You are right; we were giving the definition of ML models.

*Line 24-25: This is a controversial statement. Many methods have been developed to interpret ML models and use ML to understand the underlying physical processes. Entire books have been written on the subject (`https: // christophm. github. io/ interpretable-ml-book/ `).*

We believe ML models do not provide the same level of interpretation as physical models. We included below a comparison table from the EPA that provides an overview on different rainfall-runoff model types [Sitterson et al., 2018]. ML models would fall under "Empirical" category. Also, Breiman [2001] himself wrote, "A forest of trees is impenetrable as far as simple interpretations of its mechanism go. In some applications, analysis of medical experiments for example, it is critical to understand the interaction of variables that is providing the predictive accuracy." He later discussed the application of permutation-based variable importance as a proxy to understand the input-output relationship. We discussed the limitation of this method in our manuscript under Section 4.5.

*Table 1. Comparison of the basic structure for rainfall-runoff models*

|  | **Empirical** | **Conceptual** | **Physical** |
|---|---|---|---|
| Method | Non-linear relationship between inputs and outputs, black box concept | Simplified equations that represent water storage in catchment | Physical laws and equations based on real hydrologic responses |
| Strengths | Small number of parameters needed, can be more accurate, fast run time | Easy to calibrate, simple model structure | Incorporates spatial and temporal variability, very fine scale |
| Weaknesses | No connection between physical catchment, input data distortion | Does not consider spatial variability within catchment | Large number of parameters and calibration needed, site specific |
| Best Use | In ungauged watersheds, runoff is the only output needed | When computational time or data are limited. | Have great data availability on a small scale |
| Examples | Curve Number, Artificial Neural Networks[a] | HSPF[b], TOPMODEL[a], HBV[a], Stanford[a] | MIKE-SHE[a], KINEROS[c], VIC[a], PRMS[d] |

a] Devi et al. (2015)
[b] Johnson et al. (2003)
[c] Woolhiser et al. (1990)
[d] Singh (1995)

*Line 27: What are other possible goals for ML? This will not be obvious to readers unfamiliar with ML.*

We revised the sentence to, "ML models are particularly useful when accurate prediction is the central inferential goal (Dibike and Solomatine, 2001), whereas conceptual rainfall-runoff model can help gain a better understanding of hydrologic phenomena and catchment yields and responses (Sitterson et al., 2018).

*Line 28: Neuro-fuzzy what? This is an adjective in a list of nouns.*

Neuro-fuzzy refers to combinations of artificial neural networks and fuzzy logic and is commonly used as a noun in the literature. For example, Mosavi et al. [2018] discussed flood forecasting using ML models and wrote, "Many ML algorithms, e.g., artificial neural networks (ANNs), neuro-fuzzy, support vector machine (SVM), and support vector regression (SVR) were reported as effective for both short-term and long-term flood forecast."

*Line 33: Which other models? Be specific. Your literature review should motivate the methods that you end up using.*

We were referring to the two models, SVM and GP, from the previous sentence. In this paragraph, we would like to review the applications of various ML models in streamflow forecasting. We specifically discussed the reasons we used RF for our study on line 74-81.

*Line 35: Define.*

We feel like this is not relevant because we did not use "baseflow separation" in our study. Readers who are interested in this method can look into it the referenced paper.

*Line 46: If this the region shown in Figure 2? If yes, you should reference Figure 2 here. If no, you should include a map of the region in a different figure.*

Yes, it is the same region in Figure 2 and we referenced it here in the revised manuscript.

*Line 48: Does "unregulated" mean "not human-modified"?*

Yes.

*Line 56: "RF can be trained to forecast streamflow at various timescales, depending on the selection of input variables." I don't know what you mean by this. Random forests (and the individual trees therein) perform variable selection automatically, so random forests should not "depend on the selection of input variables".*

The performance of the model to forecast "at various timescales" does depend on selection of predictor variables. For example, a study focuses on seasonal streamflow forecasting would consider including climate indices such as Southern Oscillation Index as one of the predictor variables.

*Line 58: I don't understand why this prevents you from forecasting at longer lead times. All forecasts are made with antecedent information, but some phenomena can still be forecast skillfully at lead times much longer than 1 day.*

In [Rasouli et al., 2012], the authors forecasted streamflow at 1-7 day lead times using three models: Bayesian neural network, support vector regression, and Gaussian process, and data from

combinations of climate indices and local meteo-hydrologic observations. They concluded local observations as predictors were generally best at shorter lead times while local observations plus climate indices were best at longer lead times of 5–7 days. Also, the skillfulness of all three models decreased with increasing lead times. We cited this study on line 35. In our study, we focused on 1-day lead time forecasting and therefore did not include long-term climate information.

*Line 63: This is a controversial point. Interpretation methods have been developed for many ML models. Also, many interpretation methods are model-agnostic and therefore can be applied to any ML model. In fact, one could argue that neural networks are more interpretable than random forests, since many interpretation methods rely on gradients (of the prediction with respect to model weights or input variables) and therefore cannot be applied to random forests, which are gradient-free models.*

We understand you think the statement is controversial. However, we provided the reason why we believe RF "allows for some level of interpretability." This is delivered through its permutation-based and Gini-based variable importance measures, which have been used across disciplines (citied on lines 76-78). The permutation-based feature importance in particular was developed in the original paper and also included as one of the model-agnostic methods from the book you cited above. The author also discussed the advantages and disadvantages of each method. While you are suggesting, "one could argue that neural networks are more interpretable than random forests, since many interpretation methods rely on gradients," it doesn't seem fair as this implies gradient-based methods are better than permutation-based or Gini-based methods.

*Line 66: What do you mean by this? Internal parameters (those adjusted by training), or hyper-parameters?*

While we are aware of the term "hyperparameter" that is used in the ML literature, we chose to adhere to the original usage of "parameter" in [Breiman, 2001] and randomForest R package description [Liaw et al., 2002]. These two parameters are discussed under Section 2 Methodology.

*Line 75: This is false. Random forests (at least the ones you trained) are supervised learning, because the correct answer is supplied for each training example.*

As you said, we trained RF to perform supervised learning in our study. However, random forests can be used to perform supervised and unsupervised learning [Liaw et al., 2002, Criminisi et al., 2012].

*Line 75: What do you mean by this? Random forests have two parameters for each split node (the predictor variable and threshold), so a forest with 2000 trees has ($10^4$) parameters at least.*

The term "non-parametric" does not suggest the model doesn't contain parameters. Rather, "Non-parametric methods do not assume any particular family for the distribution of the data and so do not estimate any parameters for such a distribution" [Altman and Bland, 1999].

*Line 76: This is false. The trees are always somewhat correlated, because there is overlap among training sets for the different trees (since training sets are resampled \*with replacement\* from the full training set).*

A fundamental element of random forests is the randomization of predictor selection at each split, thus minimizing the correlation among the trees. Breiman [2001] himself wrote, "The randomness used in tree construction has to aim for low correlation $\rho$ while maintaining reasonable strength." Bharathidason and Venkataeswaran [2014] discussed the idea of only including uncorrelated trees in the forest, which was shown to improve the performance of the model. As you pointed out, "trees are always somewhwat correlated," we believe this is true and changed "uncorrelated" to "decorrelated". This is consistent with the description of RF in the Elements of Statistical Learning text [Friedman et al., 2001].

*Line 83: How does this happen? How does each tree make a prediction for a new example? Please clarify.*

We added the following sentence to the manuscript, "After all the trees are grown, the forests make prediction on a new data point by having all trees run through the predictors. In the end, the trees cast a majority vote on a label class for classification task or produce a value for regression task by averaging all predictions."

*Algorithm 1: This algorithm will probably not be intuitive for readers unfamiliar with ML. I think a plain-language explanation, along with a figure, would be much better.*

In the submitted manuscript, we included both plain-language explanation (lines 75:85) and a figure (Figure 1).

*Line 88: This is not an estimate. It is called the "0.632" rule and has been mathematically proven: https://www.jstor.org/stable/2965703?seq=1*

In the cited paper, the author discussed that the value "0.632+" was "estimated" using bootstrap. It is also shown in [Albert et al., 2008] that the probability of not selecting an event in the bootstrap

procedure becomes $e^{-1} \approx 0.369$ or approximately 37%.

*Line 97: Unnecessary detail, since the hyperparameter experiment you described could be implemented with a simple for-loop (does not require a special library).*

We believe citation of packages used in the study is important for reproducibility.

*Line 104: So you compute a different MSE for each tree, rather than computing one MSE for the whole random forest?*

MSE is calculated at tree level because the OOB sample used to compute MSE is different for each tree.

*Line 105: Is this method any different than the permutation test created by Breiman (2001)?*

It is the same method.

*Line 108: Not necessarily. If there are two highly correlated predictor variables ($x_1$ and $x_2$), permuting one of the two may not decrease the model's performance. For example, if you permute only $x_1$, even if $x_1$ is highly important, the model may still perform well by relying on $x_2$, since $x_1$ and $x_2$ contain a lot of redundant information.*

We did not consider this and thank you for pointing it out. Boulesteix et al. [2012] discussed the challenge of accurately measuring variable importance in computational biology and bioinformatics studies when highly correlated predictor variables are involved. It is relevant in our study as we supplied the model with maximum temperature and minimum temperature, which are correlated. We will add a discussion of this issue under Section 4.5 Variable importance analysis.

*Line 111: ? I generally don't know what you mean in this sentence.*

We provided details in the next sentence. In regression decision tree, split only occurs when the residual sum of squares (from Step 3 in Algorithm 1) of two descendent nodes is less than that of their parent node. In other words, there is a reduction in residual errors and the MDI measures this reduction.

*Line 114: This shouldn't matter if you have only one response variable, right? It should matter only if you have multiple response variables with different scales (e.g., one response variable that ranges from 0...1 and another that ranges from 500...5000).*

We standardized all variables in the training and validation sets. Because of this, raw MDI does not have an associated unit and provides little interpretation. Scaled MDI, on the other hand, can be interpreted as the relative contribution, in percentage, of each predictor to the total reduction in node impurities.

*Line 125: I suggest calling this the "persistence baseline," rather than the naïve model. The word "naïve" evokes naïve Bayes for many people.*

This is valid but we believe naïve model is commonly used in the context of hydrologic forecasting. We also defined this terminology.

*Line 125: What are these limitations? Please discuss. Model evaluation is very important, and the methods you use should be explicitly justified.*

We added the following clarification, "Among the limitations, these measures were reported to be especially oversensitive to extreme values (outliers)."

*Line 136: How?*

We explained this for each evaluation metric in the following paragraphs in the manuscript.

*Line 139: Please define all variables in this equation, including the N and i.*

We added the following to the sentence, " $\hat{y}_i$ and $y_i$ are the forecasted and observed values at day $i$ respectively, and N is total number of the observations during the validation period."

*Line 145: What do you mean by this? More sensitive to outliers?*

By "error", we were referring to the difference between the predicted and observed values ($|\hat{y}_i - y_i|$). Due to the squared operation, RMSE is therefore more sensitive to large errors.

*Line 147: Please state the ranges and optimal values for MAE and RMSE, like you did for $R^2$. (I know that it's probably obvious to most readers, but it's a small amount of additional text and worth specifying.)*

Actually, both MAE and RMSE depend on the raw value of response variable, $y$, which will vary from one study to another. They are better interpreted comparatively.

*Line 160: I don't know what this means.*

For the validation period, we calculated the 90th, 95th, and 99th percentile streamflow values at each watershed. These are considered thresholds. If an observed daily streamflow exceeded this threshold, it would be considered an extreme event.

*Line 180: A buffering effect to what? What does the ocean "buffer".*

We acknowledge the term "buffering effect" might be vague and revised the sentence to, "Proximity to the ocean creates a more moderate climate with a narrower temperature range, particularly in the winter."

*Line 221- 223: Predictors of what? Streamflow, or SWE?*

They were included as predictors for the RF model. All eight predictors are listed in Table 2.

*Line 223: Why only the last measurement of each day?*

We only supplied the last measurement from SNOTEL stations because not all predictors have sub-daily values.

*Line 225: How big are these basins? Please show a map.*

We added the following map and table to Supplementary material.

[Figure]

Figure 1: Map of basins within the Pacific Northwest Hydrological Unit. Blue basins contained at least one watershed and were included in the the study.

| HUC-6 | Name | Area ($km^2$) | Number of SNOTEL |
|---|---|---|---|
| 170102 | Pend Oreille | 67598.70 | 30 |
| 170200 | Upper Columbia | 119755.57 | 10 |
| 170300 | Yakima | 15928.20 | 9 |
| 170401 | Snake Headwaters | 14812.20 | 11 |
| 170501 | Middle Snake-Boise | 85150.16 | 27 |
| 170601 | Lower Snake | 30198.02 | 7 |
| 170602 | Salmon | 36248.15 | 11 |
| 170603 | Clearwater | 24318.13 | 9 |
| 170701 | Middle Columbia | 29124.57 | 11 |
| 170703 | Deschutes | 27789.56 | 8 |
| 170800 | Lower Columbia | 16120.04 | 15 |
| 170900 | Willamette | 29697.66 | 15 |
| 171003 | Southern Oregon Coastal | 34510.01 | 6 |
| 171100 | Puget Sound | 52958.23 | 26 |
| 171200 | Oregon Closed Basins | 45143.34 | 6 |

*Line 227: Can you discuss how much this affects the accuracy of your model? It seems like a major caveat.*

This represents a shortcoming of the study due to limited spatial coverage of SNOTEL stations and the introduced uncertainty likely affects the accuracy of the model. We acknowledged this by

stating, "The SNOTEL averages, therefore, represent first-order estimates of snow coverage and temperature conditions." We also considered an alternative approach by drawing information only from the SNOTEL station closest to gauge location but decided the basin-average better represented SWE conditions. Using basin-average SNOTEL SWE is consistent with previous studies in that focused on streamflow forecast [Abudu et al., 2011] as well as contribution of snowmelt to streamflow [Zheng et al., 2018] in western USA. Nevertheless, we believe supplying the RF with a more spatially consistent SWE data would improve model accuracy and is certainly worthy of future research. The reported RF performance in our study might be an underestimation.

*Line 232: Why not use all predictors available? Random forests are not computationally expensive and perform predictor selection automatically.*

We believe the current selection is appropriate. We also had to consider the practical purpose of the model. While there are other variables we could include such as soil moisture content, many would not be available for 1-day ahead forecasting in real time.

*Line 235: Why use $T_{min}$ and $T_{max}$ as predictors if their only influence is on SWE (another predictor)? In that case you should just use SWE.*

It's worth mentioning that these predictors are at 1-day lag. SWE only reflects the current state of snow condition and melting of snow is often triggered by changes in temperature. Given that there is high temporal correlation in daily temperatures, $T_{min}$ and $T_{max}$ data can provide useful signal to our streamflow forecast.

*Line 237: This only tells me what a pentad is, not what the "pentad index" is. Please define the "pentad index".*

In this case, the "pentad index" refers to the numerical sequence of pentads in a calendar year (1 to 73).

*Line 239: What do you mean by "across gauges"? Is this correlation a spatial correlation, or is it computed at each gauge (in which case it's temporal but not spatial)?*

Temporal correlation between daily streamflow and Pentad Index was computed at each gauge here.

*Line 247: How? There are many ways to do min-max scaling. For example, what are the min and max values after standardization? Also, which dataset do you use to compute the min and max values for scaling? Just the training set, or both training and validation?*

We added the following clarification to the manuscript, "We standardized training and validation data at each gauge using min-max scaling. The new data for all variables have values between zero and one."

*Line 251: Random forests have many other hyperparameters: minimum sample size per split node, minimum sample size per leaf node, maximum depth, cost function, etc.).*

We were aware of this but preferred to focus on the two parameters discussed in [Breiman, 2001].

*What is a "sample of training data sets"? I thought you had only one training set (2009-15).* Thanks for allowing us to clarify this. We tested RF on training data sets of 30 randomly chosen watersheds and observed that the reduction in error is negligible after 2000 trees. Then we set the number of trees to 2000 for all watersheds.

*Line 256: Why optimize for MAE, instead of one of the other scores you looked at (RMSE, $R^2$, or KGE)?*

We actually optimized using both MAE and RMSE. The results were similar except for a few watersheds. The reason we moved forward based on MAE results was because RMSE penalizes larger errors and it was our interest to minimize the average errors, not large errors. KGE and $R^2$, on the other hand, do not directly capture the magnitude of errors but rather the overall performance of the model. They are also not commonly used in parameter tuning.

*Line 261: On line 95 you said that the default is M/3, where M = number of predictors.*

Yes, we have 8 predictors so round-up of 8/3 is 3.

*Anonymous: Is this shown in the figures?*

Yes, this is shown in figure 5a with correlation coefficient of RF (y-axis) plotted against correlation coefficient of naïve model (x-axis). For rainfall-driven and transient watersheds, most points lie on the left of the 1-to-1 line, suggesting RF outperforms naïve model. For snowmelt-driven watersheds, the points lie on the 1-to-1 line, which indicates there is marginal difference in the models' performance.

*Line 271: Is this shown in the figures?*

Yes.

*Line 274: ? I don't understand this sentence.*

Sorry for the typo. The sentence should read, "Without accounting for persistence, it would be inadequate to conclude that RF delivered better performance compared to the other two groups."

*Line 275: How many of these differences are statistically significant? In general, all comparisons between two models should be accompanied by a significance test.*

We addressed this comment in the Major comments section.

*Line 283: Could you verify this hypothesis by analyzing the data?*

Yes, the hydrographs of snowmelt-driven watersheds tend to be less flashy compared to rainfall-driven watersheds.

*Line 287: How can large errors and mean errors have the same distribution? By definition, large errors are greater than mean errors.*

By "distribution", we were referring to the dispersions of the RMSE and MAE score values for 3 groups in Figure 6b. For example, the MAE scores are heavily skewed towards 0 while RMSE scores are more evenly spread among snowmelt-driven watersheds.

*Line 293: The opposite of "poor" is not "satisfactory". Does the Rogelis paper define other ranges of KGE as "fair," "good," "excellent," etc.? Or does it just say that the 0-0.5 range is "poor"?*

As we explained above, these scores should be evaluated comparatively rather than absolutely.

*Line 295: ? Define.*

We addressed this comment above.

*Line 299: Is this a fair comparison? The sets of watersheds is your paper vs. Tongal and Booij are completely different, no?*

You are right. For this very reason, we simply reported the KGE scores in our study and theirs without making a comparison of the two models.

*Line 301: Figure 7 should be plotted on a performance diagram. This would allow you to show POD, FAR, frequency bias, and CSI all in the same figure. For example, see Figure 12 in this paper: https://journals.ametsoc.org/waf/article/35/4/1523/347594*

We think this is a good suggestion. We included here the relative operating characteristic (ROC) plot, which measures the ability of forecast model to discriminate between events and no-events across thresholds. This is similar to the performance diagram in the paper you referenced. We will modify the discussion on POD and FAR based on this new plot.

[Figure]

Figure 2: Probability of detection plotted is plotted against false alarm rate for three extreme thresholds: $90^{\text{th}}$, $95^{\text{th}}$, and $99^{\text{th}}$ percentiles.

*Line 302: What is the actual value corresponding to each percentile?*

The actual values corresponding to each of the three thresholds varied across watersheds. Because of this, we did not record the actual values them and focused on the FAR and POD values.

*Line 308: This hypothesis seems like just a guess. Can you verify it by looking at the data (i.e., explicitly looking at predictions for cases with large surges vs. cases without large surges)?*

This is not a guess but suggested by our understanding of the hydrology of the region, examination of hydrographs, and the POD rate of RF among snowmelt-driven watersheds. The large surges of runoff from these watersheds likely occur during spring and early summer (March-June in Figure 2d).

*Line 309: What does this mean? FAR and POD measure very different things, so what does it mean for them to be "in agreement"?*

Although POD and FAR provide different measurements, high POD and low FAR suggest skillful forecast. This is shown on slide 25 here (`https://www.nws.noaa.gov/oh/hrl/hsmb/docs/hep/events_announce/STEWksp_Training_Hydro_Verification_30Nov06.pdf`).

*Line 311: You don't know this until you have calculated frequency bias (which is shown in performance diagrams).*

We're not sure how "frequency bias" is calculated as the paper cited above did not mention it. However, based on our ROC plot, if there were systematic overestimation, FAR would be exceed POD and the ROC curve would fall below the no-skill line (slide 38 in the document from NOAA we referenced above).

**References**

S. Abudu, J. P. King, and A. S. Bawazir. Forecasting monthly streamflow of spring-summer runoff season in rio grande headwaters basin using stochastic hybrid modeling approach. *Journal of Hydrologic Engineering*, 16(4):384–390, 2011.

J. Albert, E. Aliu, H. Anderhub, P. Antoranz, A. Armada, M. Asensio, C. Baixeras, J. Barrio, H. Bartko, D. Bastieri, et al. Implementation of the random forest method for the imaging atmospheric cherenkov telescope magic. *Nuclear Instruments and Methods in Physics Research Section A: Accelerators, Spectrometers, Detectors and Associated Equipment*, 588(3):424–432, 2008.

D. G. Altman and J. M. Bland. Statistics notes variables and parameters. *Bmj*, 318(7199):1667, 1999.

S. Bharathidason and C. J. Venkataeswaran. Improving classification accuracy based on random forest model with uncorrelated high performing trees. *Int. J. Comput. Appl*, 101(13):26–30, 2014.

A.-L. Boulesteix, S. Janitza, J. Kruppa, and I. R. König. Overview of random forest methodology and practical guidance with emphasis on computational biology and bioinformatics. *Wiley Interdisciplinary Reviews: Data Mining and Knowledge Discovery*, 2(6):493–507, 2012.

L. Breiman. Random forests. *Machine learning*, 45(1):5–32, 2001.

A. Criminisi, J. Shotton, and E. Konukoglu. Decision forests: A unified framework for classification, regression, density estimation, manifold learning and semi-supervised learning. *Foundations and Trends® in Computer Graphics and Vision*, 7(2–3):81–227, 2012.

J. Friedman, T. Hastie, and R. Tibshirani. *The elements of statistical learning*, volume 1. Springer series in statistics New York, 2001.

A. Liaw, M. Wiener, et al. Classification and regression by randomforest. *R news*, 2(3):18–22, 2002.

A. Mosavi, P. Ozturk, and K.-w. Chau. Flood prediction using machine learning models: Literature review. *Water*, 10(11):1536, 2018.

K. Rasouli, W. W. Hsieh, and A. J. Cannon. Daily streamflow forecasting by machine learning methods with weather and climate inputs. *Journal of Hydrology*, 414:284–293, 2012.

J. Sitterson, C. Knightes, R. Parmar, K. Wolfe, B. Avant, and M. Muche. An overview of rainfall-runoff model types. 2018.

Z. M. Yaseen, A. El-Shafie, O. Jaafar, H. A. Afan, and K. N. Sayl. Artificial intelligence based models for stream-flow forecasting: 2000–2015. *Journal of Hydrology*, 530:829–844, 2015.

X. Zheng, Q. Wang, L. Zhou, Q. Sun, and Q. Li. Predictive contributions of snowmelt and rainfall to streamflow variations in the western united states. *Advances in Meteorology*, 2018, 2018.

---

## Editor Comment (EC1) · Dimitri Solomatine (Editor) · 28 Sep 2020

Referees did an excellent job and provided a lot of comments and useful suggestions. The authors recognise and appreciate most of them and it seems they have a good plan for revision. Success! Please give special attention to interpretation and discussion of results, clear conslusions and recommendations, and to English (grammar, use of words and clarity).

———————————————

---

## Referee Report (RR1)

I thank the authors for their hard work in revising the manuscript, and I find it much improved. However, I still have several major comments (listed below) and many minor comments (made inline in the manuscript; document below), so I recommend major revisions before publication. Note that I have made inline comments on some of the authors' responses, as well as the manuscript itself.

**Major comments**

1. Results still contain no significance tests or error bars. The authors claim that their random forest outperforms the two baseline models (persistence, which they call the "naïve" model, and linear regression), but there are no significance tests or error bars to support this claim. I made the same comment in round 1, and I find the authors' response (explaining why they chose not to include significance tests as requested) unsatisfactory. See page 6 of the document below for the authors' response (and my response to their response).

2. The authors' claim that random forests are more interpretable than other machine-learning models, is highly debatable. For details, see my first two comments on page 9 of the document below. This comment should be easy to address – I would just like to see the authors add a few sentences to the manuscript, discussing the controversy of model interpretability. *i.e.,* There are properties of random forests that make them more interpretable than other ML models, but there are also properties that make them less interpretable, so it's unfair to simply say "RF allows for some level of interpretability" (as their justification for using random forests) and then move on.

3. The authors have not clarified which data (training only or training plus validation) they use to compute standardization parameters – *i.e.,* to compute the minimum and maximum for min-max scaling. See my fourth comment on page 12 of the document below. The distinction is important. Only training data should be used to compute standardization parameters; if the validation data are also used to compute standardization parameters, this means that validation data are used to pre-process training data, which means that information from the validation data has "leaked" into the training data, which means that the two datasets are no longer independent.

4. The authors should justify why they experimented with only two hyperparameters. See my fifth comment on page 12 of the document below.

5. The authors use ROC curves to diagnose (the absence of) systematic overestimation, which is an invalid interpretation of ROC curves. See my comment on line 14 of the document below.

6. The "no-skill line" in the ROC curves (Figure 7) is misplaced. The no-skill line should be the *x = y* line (or POD = false-alarm rate), which is the ROC curve that would be achieved by a random model such as a coin flip.

7. I don't understand why the authors include only 3 probability thresholds in their ROC curves (90%, 95%, and 99%). A typical ROC curve includes probability thresholds spanning the range from 0-100% (typically in increments of 10% at most – usually in increments of 1% or 0.1%, which allows for a smoother curve). Seeing such a small portion of the full ROC curve, it is difficult to assess the models' performance. I request that the authors plot the full ROC curve for each model, like the ones shown/explained here: https://developers.google.com/machine-learning/crash-course/classification/roc-and-auc. Plotting the full ROC curves would also allow

the authors to compute area under the ROC curve (AUC), which is a scalar typically used to quantify the goodness of a ROC curve.

8.  As in round 1, I request that the authors include performance diagrams along with the ROC curves (see my last comment on page 13 of the document below).  Performance diagrams plot probability of detection (POD) on the *y*-axis vs. success ratio (1 minus false-alarm **ratio**, which is different than false-alarm **rate**) on the *x*-axis.  Since frequency bias = POD / success ratio, performance diagrams can be used to diagnose systematic overestimation, which the authors have identified as a goal.  Performance diagrams look like Figure 4 in this paper: https://journals.ametsoc.org/view/journals/wefo/34/6/waf-d-19-0094_1.xml?tab_body=fulltext-display.  If it proves too complicated to plot the contours in the background (frequency bias and critical success index), I would be okay with the authors leaving the contours out.

9.  The discussion on lines 384-391 is unclear throughout.  See inline comments.

10. Mean-flow benchmark.  On lines 331-332, the authors cite a previous paper to claim that a Kling-Gupta efficiency (KGE) ≥ -0.41 improves upon the "mean-flow benchmark," which is the KGE achieved by always predicting the time-mean flow ("climatology") at the given basin. I request that the authors compute the mean-flow benchmark for their own dataset, since it may be different than the dataset used in the paper they cite.

We would like to thank Editor Dimitri Solomatine and two reviewers for their time and valuable feedback to our manuscript. The manuscript has benefited from the informative suggestions. We appreciate the opportunity to submit a revised version for further consideration. Please find below a list of our answers to the comments and changes to the paper carried out to carefully address the remarks and the suggestions proposed by the two referees. The comments are shown below in black italic font. Our responses and revisions are presented below each comment. A marked-up version of the manuscript showing the specific changes we made is submitted along with this letter. We hope that this will allow the editor to assess the revision.

**1 Referee 1**

**1.1 Overall comments**

*Pham et al. developed a Random-Forest-based (RF) algorithm for day-ahead streamflow forecasting. The algorithm employs 8 weather-snow-time features (see Table 2) and was tested across 86 watersheds in the Pacific Northwest (PNW), where it was compared with multilinear regression and previous-day streamflow as a minimum information forecasting approach (so called Naïve method). Results show that RFs provide quite robust predictions across catchments with different climatology (rainfall dominated, snowfall dominated, or transient) and generally perform better than the two benchmark approaches – especially in rainfall-dominated and transient watersheds. Drops in accuracy for RFs were correlated with watershed slope and sandiness. This is an interesting, well-written, and concise paper about an emerging machine learning technique in hydrology and its use for streamflow forecasting. While I have tangential knowledge of RF technical issues, I found the description of the algorithm clear and rigorous, which facilitates replicability and ultimately allows readers to learn more about RFs in general rather than only looking at is as a black box (note that these technical details are often bypassed or heavily summarized in other papers I have read on this matter).*

Thank you for the review and supportive and constructive feedback. Yes, we too thought it would be helpful to provide this level of detail about the regression algorithms.

*Also, RFs and machine-learning approaches in general are on the rise in hydrology, meaning I expect the paper to have some impact on the community.*

Yes, we agree, these underused tools offer some exciting opportunities in this field. Tyralis et al. [2019] reviewed and compiled the applications of RF in water resources, where the number of papers has risen exponentially in the period between 2000 and 2019 (Figure 2). The authors of this study also acknowledged in their conclusion that, "it is quite remarkable that only a few studies recognize possible shortcomings of random forests and their variants." The analysis of the two commonly used measures of predictor importance(permutation-based Mean Decrease in Accuracy and Gini-based Mean Decrease in Node Impurity) in our manuscript suggests they might not be reliable and such analysis should be coupled with the understanding of the physical processes under study. We believe identifying and providing evidence for this shortcoming constitute a contribution to the current literature and for future research on the applicability of RF.

*There are still some major and minor comments that I recommend for authors (see below), and I recommend the editor reconsider this manuscript after minor revisions*

We appreciate the comments. We clarified and revised the manuscript according to your suggestions.

**1.2 Major comments**

*1.The manuscript sometimes reads like a technical note, as it describes the algorithm and its implementation in great details but ultimately falls a little short on hydrological process interpretation. I see that the main goal of the paper is testing an algorithm, and applied research is certainly within the scope of HESS. And yet, I feel like implementing RFs across 86 watersheds with different characteristics and 10 yrs with different climatology without looking more specifically at how performance changes across the landscape and between years with different characteristics is kind of a missed opportunity.*

In the manuscript, we tried to maintain a focus on (1) implementing and evaluating Random Forest for 1-day streamflow forecast and (2) exploring how the variations in hydrological processes across watersheds (e.g., contributions of rainfall and snowfall to streamflow, physical factors) might affect the model performance. In the Introduction, we provided an synopsis on the climatology of the Pacific Northwest and the roles of snowmelt and rainfall in driving streamflow dynamics. In Section 3.1 Study

Area and Data, we described the regional hydrographic systems of the Columbia River Basin. We also included physical characteristics of the watersheds under study, which were compiled and documented in the GAGESII Dataset (`https://water.usgs.gov/lookup/getspatial?gagesII_Sept2011`) and available in Supplementary material. We further systematically classified the watersheds into three hydrologic regimes: snowmelt-driven, rainfall-driven, and transient. As seen in Figure 2, the majority of the rainfall-dominated and transient watersheds are located on the west side of the Cascades while snowmelt-dominated watersheds are east of the Cascades and at higher elevation (Table 1). We reported the performance of RF for each watershed and explored how factors such as drainage area and slope could affect model performance in Section 4.6. Wenger et al. (2009) simulated and evaluated runoff at 55 watersheds in the Pacific Northwest using variable infiltration capacity (VIC) model and followed a similar classification scheme. Summary statistics of individual watersheds were reported similarly in Table A1 (`doi:10.1029/2009WR008839`). While we also thought about exploring the inter-annual and seasonal variability in model performance, reporting multiple sets of evaluation metrics (e.g., KGE, $R^2$, RMSE, MAE) for 86 watersheds would become confusing and dilute the scope of the paper.

*I was a little surprised by the choice of benchmark models and particularly by the fact that authors did not consider a full hydrologic model. I understand that authors would probably like to stay within the realm of data-driven models, but a Naïve approach looks very simplistic, especially at a daily time scale and in basins where rainfall and snowfall coexist.*

We did not consider a full hydrologic model and instead selected Naïve model as a benchmarking model for two reasons. First, as we briefly mentioned in the Introduction, both data-driven and hydrologic models have their advantages and disadvantages, and yet our objective was not to show one is superior to the other. Safeeq et al. [2014] provides an assessment of Variable Infiltration Capacity (VIC) model for predicting hydrologic regimes of 217 watersheds in the Pacific Northwest at 1/16 degree (6km×6km) grid-scale resolution. The author discussed the large underestimation of simulated SWE from meteorological data forcing compared to that at SNOTEL sites, which could be contributing to model bias. As we supplied RF with SNOTEL data, a comparison on runoff forecast between two models would be inappropriate. Second, we showed that, to our surprise, RF was not able to beat a simple Naïve model in its forecast among snowmelt-driven watersheds with strong persistence (Figure 5). We observed in some other papers where multiple ML models were tested and compared against one another without a baseline model. We wondered how much of the prediction were actually due to persistence in streamflow. As Pappenberger et al. [2015] pointed out, "benchmarking with simpler models can be viewed as a gain-based approach." For this reason, we believe the use of the Naïve model is justified.

*How can this approach predict, e.g., intense rain-on-snow events that are ubiquitous in the PNW?*

While this is a relevant question given the climatology of the PNW and the potential risk of rain-on-snow (ROS) floods, ROS events themselves are quite complex hydrometeorological phenomenon. Since we supplied RF with 1-day antecedent streamflow, SWE conditions, and meteorological data (precipitation and temperature), we would not expect the model to predict these ROS events. This would require additional future weather forecast information and fall under real-time forecast, which is out of the scope of our approach. We discussed the potential of including $t+1$ precipitation forecast on line 377-383 under Section 4.7 Limitations and future research.

*Most flood-forecasting tools I have been exposed to use full hydrologic models, and I encourage authors to at least discuss this matter in their manuscript.*

Good suggestion. We have now added the following discussion on the application of hydrologic models in this region under Introduction section. *Despite the promising results reported in existing literature, most ML streamflow forecast applications are limited to watersheds where rainfall is the major contributor. In many settings, particularly, non-arid mountainous regions in Western USA, a combination of rainfall and spring snowmelt can drive streamflow [Johnstone, 2011, Knowles et al., 2007]. The amount of snow accumulation and its contribution to discharge also vary among the watersheds [Knowles et al., 2006]. Both watershed-scale hydrologic models and statistical models have been used to access the current and future stream hydrology and associated flood risks [Salathé Jr et al., 2014, Wenger et al., 2010, Tohver et al., 2014, Pagano et al., 2009]. Safeeq et al. [2014] simulated streamflows using the Variable Infiltration Capacity (VIC) model at 1/16° and 1/20° spatial resolutions and evaluated against observed values from 217 watersheds at at annual and season time scales. The study found the model was able to capture the hydrologic behavior of the study watersheds with reasonable accuracy. Yet authors recommended careful site-specific model calibration using not only streamflow but also SWE data would be expected to improve model performance and reduce model*

*bias. Pagano et al. [2009] applied Z-Score Regression to daily SWE from SNOTEL stations and year-to-date precipitation data to predict seasonal streamflow volume in unregulated streams in Western US. Authors reported the skill of these forecasts was comparable to the official published outlooks. A natural question is whether ML models can produce comparable performance in these watersheds where streamflow contributions come from a mix of snowmelt and rainfall, as well as where snowmelt dominates sources.*

*3. Relatedly, I was also a little surprised that authors did not consider rainfall and snowfall as separate features in their model (see their Table 2). I am aware that PRISM only provides total precip, but it also provides temperature and relative humidity that could be employed to separate snowfall from rainfall. Perhaps considering SWE already makes up for this, but I encourage authors consider this at least for future work. This was again particularly puzzling to me given the well-known role of rain on snow in this region.*

We did consider the need to differentiating falling precipitation as rainfall from snowfall using surface air temperature data from PRISM as suggested. However, there are certain drawbacks of this approach. We expect the shifts in temperature can be dramatic at daily time scale, especially in the mountainous region with complex terrain, and may not be well captured by PRISM data. Moreover, watersheds in this study can span a wide range of elevation, making the determination of threshold temperature difficult. We chose not to explicitly differentiate the precipitation types and alternatively included maximum temperature (Tmax) and minimum temperature (Tmin) recorded at SNOTEL stations as predictors in the model. While it is uncertain to say whether the model was able to pick such signal due to the "black-box" nature of RF, we can see in Figure 8 that Tmin variable is ranked high among Snowmelt-dominant watersheds in variable importance. But this could also an indication of better prediction due to snowpack's sensitivity in temperature shifts rather than rain-snow differentiation.

**1.3   Specific comments**

We here address the specific comments. Simple typo fixes and clarifications were made directly in the revised manuscript.

*- Title: I have usually seen "rainfall-dominated" and "snowfall-dominated" being used, rather than rainfall and snowmelt driven. Consider revising.*

We used "snowmelt-dominated" and "snowmelt-driven" interexhangably throughout the manuscript. We included here two publications that employed similar usage, *"Soil moisture states, lateral flow, and streamflow generation in a semi-arid, snowmelt-driven catchment"* and *"Climate change impacts on the hydrology of a snowmelt driven basin in semiarid Chile"*.

*- Line 24: I believe even ML algorithms need the formulation of some mathematical equations, although maybe not in a predictive role.*

We agree and deleted "formulation of mathematical equations" from the sentence.

*I think "ease of application" might be relative, especially in ungauged areas or for users with limited computational capabilities. Consider revising or expanding*

The sentence has been revised to, "Among other qualities, the popularity of ML for such applications is due to the methods' competitive performance compared with alternative approaches, relative ease in implementation, and less strict distributional assumption."

*Line 38: maybe also mention glaciers here, although they might not be an important driver for hydrology in your study region.*

It's a good point and by saying, "most ML streamflow forecast applications are limited to watersheds where rainfall is the *major* contributor," we acknowledge there are other sources that can contribute to streamflow.

*- Line 56: indeed, statistical forecasting models are widely used across the western US to predict summer flow (e.g., April to July total runoff). I understand this is out of the scope of your paper, but maybe mention this application to provide broader framing to your work*

We appreciate the recommendation and mentioned a statistical forecasting model along with application of physical models in the Introduction.

*Line 149: Knoben at al. (https://hess.copernicus.org/articles/23/4323/2019/) have recently pointed out that KGE = 0 has a different implication from NSE = 0, and so KGE = 0 should be used with caution. Please revise as relevant.*

Thank you for poiting this out. The following is our revision, "KGE metric ranges between -inf and 1. While there currently is not a definitive KGE scale, Knoben et al. (2019) showed KGE values

in the range between -0.41 and 1 indicate the model a model improves upon the mean flow benchmark , which assumes the predicted streamflow values equal to the mean of all observations. Generally, KGE value of 1 suggests the model can perfectly reproduce observations."

*Line 176: maybe some more quantitative climatology would be more appropriate here. For instance, replace "ample amount of winter precipitation" with statistics of winter precip for your watersheds. Same for "mild temperature". It would also be interesting to provide some statistics of mean-max SWE across the basins.*

We updated Table 1 to include summary statistics for mean annual temperature and mean annual precipitation across three hydrologic regimes. We also added the following plot to the manuscript. SWE from SNOTEL stations was calculated at HUC-6 level and min-max statistics would be available in Supplementary.

Figure 1: Gauge locations with color gradient indicating variations in (a) watershed drainage area, (b) mean watershed elevation, (c) mean watershed annual precipitation, and (d) mean watershed annual temperature.

[Figure]

**Table 1.** Number of streamflow gauges used in the study for each flow regime, ranges of mean watershed elevation and drainage area. Complete catchment physical and hydro-climatic characteristics for each watershed can be found in Appendix A.

| Hydrologic regime | Number of gauges | Mean watershed elevation (m) | Drainage area (km²) | Mean annual precipitation (cm) | Mean annual temperature (deg C) |
|---|---|---|---|---|---|
| Rainfall-dominated | 33 | 239 - 1207 | 58 - 703 | 122.0 - 367.0 | 5.4 - 11.5 |
| Transitional | 28 | 813 - 1477 | 58 - 1855 | 63.2 - 314.0 | 4.16 - 8.42 |
| Snowmelt-dominated | 25 | 1349 - 2509 | 51 - 3355 | 58.0 - 177.0 | 0.4 - 6.62 |

*Line 186: have you tried to impute missing values? What's the impact of gaps in your framework?*

In the initial screen, we selected gauges that "have less than 10 percent of missing data". The majority of 86 gauges that were eventually included in the study had continuous record or less than 3 percent missing data. Therefore, we do not think imputation was necessary.

*Line 196: how "place-based" is this classification based on the day of the water year? I would have expected one to classify basins based on proportion of rainfall over total precipitation, which look more general to me.*

This classification scheme were applied to streams in the Pacific Northwest from previous studies and based on the shape of the mean annual hydrograph. We provided examples of three types of hydrographs in Figure 2 (b-d). Classifying the watersheds based on proportion of rainfall over total precipitation is another approach but this also requires differentiating rainfall from snowfall at daily timescale. We addressed a shortcoming with this approach above.

*- Line 246: how were the validation and calibration period chosen? May this choice have played a role in your results? What were the climatological characteristics of these two periods? Please expand and support your choice here.*

Seven years and three years of data were used for training and validating the model, respectively. While there is no clear requirement, this selection is consistent with convention in which training dataset should be larger than validation data set to ensure that the model is exposed to a wide range of hydrological conditions. We tested the another partition scheme where we reserved 8 years (2009-2016) for training and 2 years (2017-2018) for validating. The KGE scores across the watersheds typically improved within 0.05 margin. However, we felt 2 years of observations (approximately 700 data points) would not be sufficient to evaluate the model and decided to go with the former scheme, which might be more conservative. Moreover, our forecast is 1-day ahead so we don't expect the wet year vs dry-year characteristics, which is defined and takes place at annual time scale, would play a major role on the model's performance.

*- Line 269: I might have missed this, but do you show any statistics of persistence for your catchments to support this statement? Again, I may be missing something here.*

We discussed this at lines 279-285.

*- Line 307: may these be due to rain on snow?*

This is a possibility.

*Figure 9: consider adding the scatter plot for slope*

Thanks for the suggestion. We will add the following scatter plot slope vs. KGE in the revised manuscript.

Figure 2: KGE scores plotted against average percent of slope at each watershed. Best-fit lines were determined using simple linear regression.

[Figure]

*- Table 1: is there any reason why snowmelt-driven catchments have a larger range of drainage areas? Just out of my curiosity.*

We believe this is because the Columbia River and its major tributaries including the largest branch, the Snake River, flow through east of the Cascades and on the Oregon-Idaho border where most of the the snowmelt-dominated watersheds in our study are located.

*- Table 5: what is the data source for these characteristics? Especially sandiness and forested area*

The data from this table was part of the USGS GAGEII dataset, which was cited on line 200 and above. Percentage of sand in soil was estimated from the Department of Agriculture Digital General Soil Map of the United States or STATSGO2 Database. The data can be found in the submitted Supplementary document. Percentage of forested area was estimated using the National Land Cover Database 2006 classification. We added the following clarification to the manuscript, *"These watershed characteristics were compiled as part of GAGESII dataset using national data sources including US National Land Cover Database (NLCD) 2006 version, 100m-resolution National Elevation Dataset (NED), and Digital General Soil Map of the United States (STATSGO2)."*

**2 Referee 2**

**2.1 Overall comments**

*This paper describes the use of random forests (ensembles of decision trees) to predict streamflow in various basins in the Pacific Northwest at one-day lead time. The authors use two methods to understand the most important predictor variables, and they also investigate the effect of other basin characteristics (not included as predictor variables) on the performance of the random forest. With some improvements this work could be a valuable contribution to the literature, especially given the analyses of predictor importance and confounding variables (basin characteristics not included as predictors). However, at this time I have chosen to reject, due to several major issues with the paper. Major comments are summarized below, and inline comments are attached in a PDF.*

We appreciate the reviewer's kind words on the merit of the manuscript and its contribution to the literature. While we believe there are instances in the manuscript that can be improved, most of them can be addressed with better clarification. Please see the Inline comments section below.

**2.2 Major comments**

*1. The paper has serious grammatical issues, which make it difficult to follow. I have pointed all grammatical errors in the abstract (see `pham2020_annotations_abstract.pdf`). After the abstract, I have mostly abstained from pointing out every grammatical error. However, the frequency of grammatical errors is approximately the same throughout the paper. I would like to make it clear that I am not rejecting on the basis of grammar alone, but it does make the paper difficult to follow (there are many sentences that I simply do not understand).*

Thanks for carefully reviewing our manuscript. We have revised these errors based on your suggestions and believe the overall language is now clearer and improved.

*2. The explanation of machine-learning methods (the random forest and associated predictor-importance methods) is unclear and contains several false statements. See inline comments for more detail. The explanation is probably not clear enough for readers unfamiliar with ML to follow, and it contains enough false statements that readers familiar with ML will probably be left scratching their heads.*

We believe our description and discussion of random forests algorithm is sufficient given the scope of the paper. We also clarified statements the reviewer considered "false" and supported with cited literature.

*3. No significance-testing. The authors claim that their random forest outperforms the two baseline models (persistence, which they call the "naïve" model, and linear regression), but no significance-testing is conducted to support this claim. Especially for the comparison of the random forest with linear regression, the numbers are close enough (see line 277) that I doubt the differences are statistically significant.*

Our comparisons for the three models using correlation coefficient statistics are consistent with published literature. Specifically, Yaseen et al. [2015] conducted a review of ML models for streamflow forecasting in the period 2000–2015 and noted, "It was also found, the majority of the reviewed articles evaluated based on the correlation coefficient (R) and Root Mean Square Error (RMSE) in addition to other evaluation criteria (e.g., Relative error; RE, Mean Absolute Error; MAE, Mean Square Error; MSE, Nash–Sutcliffe Coefficient; NS, Sum of Square Error; SSE), clearly illustrated in (Fig. 1). The modeling that yields a maximum value of correlation coefficient and minimum values of RMSE and MAE presented a good evaluation of model performance." Also, result of a significance test for the differences between the correlation coefficients does not provide useful information. $R$, in this case, is an evaluation score for the models similar to KGE, RMSE, and MAE. Because we used persistence model as baseline, the positive difference in $R$ values between the two respective models can be interpreted as a gain in forecast skillfulness. It would not make sense to perform a statistical significance test for the difference between two evaluation scores of two models otherwise.

*4. Interpretation of model performance is lacking in detail and contains several confusing statements. See inline comments on lines 274, 283, 287, 293, 295, 298, 302, 304, 308, 309, and 311.*

We addressed these comments and suggestions below.

**2.3 Inline comments**

Simple typos and punctuation errors are corrected directly to the revised manuscript.

> Anonymous: This response justifies the evaluation scores used, which is a straw-man argument. I have no issue with the evaluation scores; my issue is with the lack of significance-testing. It is currently impossible to tell if any of the inter-model differences are statistically significant or due to noise.

> Anonymous: ? I don't understand what "otherwise" means in this sentence, and throughout this response I see no justification for the lack of significance tests.

**2.3.1 Abstract**

*Line 1: "Data-driven machine learning" is redundant, since all machine learning is data-driven. Also, "machine learning" should not be capitalized.*

In this sentence, we included "data-driven" to contrast ML approach with the traditional hydrologic models, which are physically based and we later discussed this under the Introduction section.

*Line 3: What do you mean by "performance"? Forecast quality, or computational efficiency?*

We agree this was vague. Following is the revised sentence, "Among other qualities, the popularity of ML models for such applications is due to their relative ease in implementation, less strict distributional assumption, and competitive computational and predictive performance."

*Line 6: This phrase seems to be missing a word. Perhaps you mean many/diverse climatic conditions and physiographic settings?*

We revised it as following, "These watersheds cover diverse climatic conditions and physiographic settings."

*Line 8: What does this mean? (I know you define the term later, but it should be defined at first mention. Alternatively, if you do not want to define jargon in the abstract, you could use a different term.)*

We believe the current description, "timing of center-of-annual-flow volume" is adequate for the purpose of the Abstract.

*Line 12-13: What does this mean? Is 0.62-0.99 good, or bad? Since most readers are probably unfamiliar with KGE, I suggest reporting the other evaluation scores here as well. Alternatively, you could just report percent improvement over the baseline models (since this is what really tells you how good the random forest is).*

Although 0.62 - 0.99 KGE score range would be considered "good", we believe these values should be evaluated comparatively not absolutely. We also did not compute "percent improvement" in our analysis. In the interest of conciseness, we also only reported raw KGE score, which has increasingly been used as a metric to access model performance in hydrology and is more objective in our opinion, in the Abstract.

*Line 15-16: What are the "new insights"? This statement is very generic and could be found in almost any paper abstract, so you should make it a bit more specific.*

We were referring to these insights: (1) "RF performance deteriorates with increase in catchment slope and increase in soil sandiness" and (2) "We note disagreement between two popular measures of RFvariable importance and recommend jointly considering these measures with the physical processes under study."

**2.3.2 Manuscript body**

*Line 23-24: This is very close to the definition of ML, although you do not explicitly say so.*

You are right; we were giving the definition of ML models.

*Line 24-25: This is a controversial statement. Many methods have been developed to interpret ML models and use ML to understand the underlying physical processes. Entire books have been written on the subject (`https://christophm.github.io/interpretable-ml-book/`).*

We believe ML models do not provide the same level of interpretation as physical models. We included below a comparison table from the EPA that provides an overview on different rainfall-runoff model types [Sitterson et al., 2018]. ML models would fall under "Empirical" category. Also, Breiman [2001] himself wrote, "A forest of trees is impenetrable as far as simple interpretations of its mechanism go. In some applications, analysis of medical experiments for example, it is critical to understand the interaction of variables that is providing the predictive accuracy." He later discussed the application of permutation-based variable importance as a proxy to understand the input-output relationship. We discussed the limitation of this method in our manuscript under Section 4.5.

Table 1. Comparison of the basic structure for rainfall-runoff models

| | Empirical | Conceptual | Physical |
|---|---|---|---|
| Method | Non-linear relationship between inputs and outputs, black box concept | Simplified equations that represent water storage in catchment | Physical laws and equations based on real hydrologic responses |
| Strengths | Small number of parameters needed, can be more accurate, fast run time | Easy to calibrate, simple model structure | Incorporates spatial and temporal variability, very fine scale |
| Weaknesses | No connection between physical catchment, input data distortion | Does not consider spatial variability within catchment | Large number of parameters and calibration needed, site specific |
| Best Use | In ungauged watersheds, runoff is the only output needed | When computational time or data are limited. | Have great data availability on a small scale |
| Examples | Curve Number, Artificial Neural Networks[a] | HSPF[b], TOPMODEL[a], HBV[a], Stanford[a] | MIKE-SHE[a], KINEROS[c], VIC[a], PRMS[d] |

a] Devi et al. (2015)
[b] Johnson et al. (2003)
[c] Woolhiser et al. (1990)
[d] Singh (1995)

*Line 27: What are other possible goals for ML? This will not be obvious to readers unfamiliar with ML.*

We revised the sentence to, "ML models are particularly useful when accurate prediction is the central inferential goal (Dibike and Solomatine, 2001), whereas conceptual rainfall-runoff model can help gain a better understanding of hydrologic phenomena and catchment yields and responses (Sitterson et al., 2018).

*Line 28: Neuro-fuzzy what? This is an adjective in a list of nouns.*

Neuro-fuzzy refers to combinations of artificial neural networks and fuzzy logic and is commonly used as a noun in the literature. For example, Mosavi et al. [2018] discussed flood forecasting using ML models and wrote, "Many ML algorithms, e.g., artificial neural networks (ANNs), neuro-fuzzy, support vector machine (SVM), and support vector regression (SVR) were reported as effective for both short-term and long-term flood forecast."

*Line 33: Which other models? Be specific. Your literature review should motivate the methods that you end up using.*

We were referring to the two models, SVM and GP, from the previous sentence. In this paragraph, we would like to review the applications of various ML models in streamflow forecasting. We specifically discussed the reasons we used RF for our study on line 74-81.

*Line 35: Define.*

We feel like this is not relevant because we did not use "baseflow separation" in our study. Readers who are interested in this method can look into it the referenced paper.

*Line 46: If this the region shown in Figure 2? If yes, you should reference Figure 2 here. If no, you should include a map of the region in a different figure.*

Yes, it is the same region in Figure 2 and we referenced it here in the revised manuscript.

*Line 48: Does "unregulated" mean "not human-modified"?*

Yes.

*Line 56: "RF can be trained to forecast streamflow at various timescales, depending on the selection of input variables." I don't know what you mean by this. Random forests (and the individual trees therein) perform variable selection automatically, so random forests should not "depend on the selection of input variables".*

The performance of the model to forecast "at various timescales" does depend on selection of predictor variables. For example, a study focuses on seasonal streamflow forecasting would consider including climate indices such as Southern Oscillation Index as one of the predictor variables.

*Line 58: I don't understand why this prevents you from forecasting at longer lead times. All forecasts are made with antecedent information, but some phenomena can still be forecast skillfully at lead times much longer than 1 day.*

In [Rasouli et al., 2012], the authors forecasted streamflow at 1-7 day lead times using three models: Bayesian neural network, support vector regression, and Gaussian process, and data from

Anonymous: Okay, but it still won't make sense to most readers to use "neuro-fuzzy" as a noun. Please replace in the text with "neuro-fuzzy (a combination of ANNs and fuzzy logic)".

Anonymous: This is still unclear in the text. Please replace "two other competing ML models" with "the other two ML models" or something similar. Currently it's not clear that "two other competing ML models" are the ones you mentioned in the previous sentence.

Anonymous: If you don't think "baseflow separation" is worth defining, please don't mention it. If it's worth mentioning, I think it's worth defining.

Anonymous: Please include this explanation in the text. The following sentence is a non-sequitur, because like I said in the first round of reviews, using only antecedent information as predictors should not limit your lead time.

"We focus on 1-day lead time because we assume only antecedent information for predictors are available at the time forecast is made."

combinations of climate indices and local meteo-hydrologic observations. They concluded local observations as predictors were generally best at shorter lead times while local observations plus climate indices were best at longer lead times of 5–7 days. Also, the skillfulness of all three models decreased with increasing lead times. We cited this study on line 35. In our study, we focused on 1-day lead time forecasting and therefore did not include long-term climate information.

*Line 63: This is a controversial point. Interpretation methods have been developed for many ML models. Also, many interpretation methods are model-agnostic and therefore can be applied to any ML model. In fact, one could argue that neural networks are more interpretable than random forests, since many interpretation methods rely on gradients (of the prediction with respect to model weights or input variables) and therefore cannot be applied to random forests, which are gradient-free models.*

We understand you think the statement is controversial. However, we provided the reason why we believe RF "allows for some level of interpretability." This is delivered through its permutation-based and Gini-based variable importance measures, which have been used across disciplines (citied on lines 76-78). The permutation-based feature importance in particular was developed in the original paper and also included as one of the model-agnostic methods from the book you cited above. The author also discussed the advantages and disadvantages of each method. While you are suggesting, "one could argue that neural networks are more interpretable than random forests, since many interpretation methods rely on gradients," it doesn't seem fair as this implies gradient-based methods are better than permutation-based or Gini-based methods.

*Line 66: What do you mean by this? Internal parameters (those adjusted by training), or hyperparameters?*

While we are aware of the term "hyperparameter" that is used in the ML literature, we chose to adhere to the original usage of "parameter" in [Breiman, 2001] and randomForest R package description [Liaw et al., 2002]. These two parameters are discussed under Section 2 Methodology.

*Line 75: This is false. Random forests (at least the ones you trained) are supervised learning, because the correct answer is supplied for each training example.*

As you said, we trained RF to perform supervised learning in our study. However, random forests can be used to perform supervised and unsupervised learning [Liaw et al., 2002, Criminisi et al., 2012].

*Line 75: What do you mean by this? Random forests have two parameters for each split node (the predictor variable and threshold), so a forest with 2000 trees has $(10^4)$ parameters at least.*

The term "non-parametric" does not suggest the model doesn't contain parameters. Rather, "Non-parametric methods do not assume any particular family for the distribution of the data and so do not estimate any parameters for such a distribution" [Altman and Bland, 1999].

*Line 76: This is false. The trees are always somewhat correlated, because there is overlap among training sets for the different trees (since training sets are resampled \*with replacement\* from the full training set).*

A fundamental element of random forests is the randomization of predictor selection at each split, thus minimizing the correlation among the trees. Breiman [2001] himself wrote, "The randomness used in tree construction has to aim for low correlation $\rho$ while maintaining reasonable strength." Bharathidason and Venkataeswaran [2014] discussed the idea of only including uncorrelated trees in the forest, which was shown to improve the performance of the model. As you pointed out, "trees are always somewhat correlated," we believe this is true and changed "uncorrelated" to "decorrelated". This is consistent with the description of RF in the Elements of Statistical Learning text [Friedman et al., 2001].

*Line 83: How does this happen? How does each tree make a prediction for a new example? Please clarify.*

We added the following sentence to the manuscript, "After all the trees are grown, the forests make prediction on a new data point by having all trees run through the predictors. In the end, the trees cast a majority vote on a label class for classification task or produce a value for regression task by averaging all predictions."

*Algorithm 1: This algorithm will probably not be intuitive for readers unfamiliar with ML. I think a plain-language explanation, along with a figure, would be much better.*

In the submitted manuscript, we included both plain-language explanation (lines 75:85) and a figure (Figure 1).

*Line 88: This is not an estimate. It is called the "0.632" rule and has been mathematically proven: https://www.jstor.org/stable/2965703?seq=1*

In the cited paper, the author discussed that the value "0.632+" was "estimated" using bootstrap. It is also shown in [Albert et al., 2008] that the probability of not selecting an event in the bootstrap

Anonymous: You cannot use the existence of the permutation test as an argument that random forests are more interpretable than other ML models, because like you say in this paragraph, the permutation test is model-agnostic. So your only valid argument here is that the Gini-based importance test can be applied to random forests and not other models. However, there are many interpretation methods that can be applied to other models (e.g., neural networks) and not random forests.

Anonymous: Please include some of this discussion in the main text. ML interpretability is a very controversial topic, and I think it's unfair to simply say "RF allows for some level of interpretability" and then move on.

Anonymous: Please clarify this in the main text. ML specialists might be confused when they see the word "parameter" being used to mean hyperparameter.

Anonymous: "Semi-supervised training" means training with a small amount of labeled data and a large amount of unlabeled data, which you are not doing. In any case, I'm happy with the change made to the main text.

Anonymous: Please clarify this in the main text.

procedure becomes $e^{-1} \approx 0.369$ or approximately 37%.

*Line 97: Unnecessary detail, since the hyperparameter experiment you described could be implemented with a simple for-loop (does not require a special library).*

We believe citation of packages used in the study is important for reproducibility.

*Line 104: So you compute a different MSE for each tree, rather than computing one MSE for the whole random forest?*

MSE is calculated at tree level because the OOB sample used to compute MSE is different for each tree.

*Line 105: Is this method any different than the permutation test created by Breiman (2001)?*

It is the same method.

*Line 108: Not necessarily. If there are two highly correlated predictor variables ($x_1$ and $x_2$), permuting one of the two may not decrease the model's performance. For example, if you permute only $x_1$, even if $x_1$ is highly important, the model may still perform well by relying on $x_2$, since $x_1$ and $x_2$ contain a lot of redundant information.*

We did not consider this and thank you for pointing it out. Boulesteix et al. [2012] discussed the challenge of accurately measuring variable importance in computational biology and bioinformatics studies when highly correlated predictor variables are involved. It is relevant in our study as we supplied the model with maximum temperature and minimum temperature, which are correlated. We will add a discussion of this issue under Section 4.5 Variable importance analysis.

*Line 111: ? I generally don't know what you mean in this sentence.*

We provided details in the next sentence. In regression decision tree, split only occurs when the residual sum of squares (from Step 3 in Algorithm 1) of two descendent nodes is less than that of their parent node. In other words, there is a reduction in residual errors and the MDI measures this reduction.

*Line 114: This shouldn't matter if you have only one response variable, right? It should matter only if you have multiple response variables with different scales (e.g., one response variable that ranges from 0...1 and another that ranges from 500...5000).*

We standardized all variables in the training and validation sets. Because of this, raw MDI does not have an associated unit and provides little interpretation. Scaled MDI, on the other hand, can be interpreted as the relative contribution, in percentage, of each predictor to the total reduction in node impurities.

*Line 125: I suggest calling this the "persistence baseline," rather than the naïve model. The word "naïve" evokes naïve Bayes for many people.*

This is valid but we believe naïve model is commonly used in the context of hydrologic forecasting. We also defined this terminology.

*Line 125: What are these limitations? Please discuss. Model evaluation is very important, and the methods you use should be explicitly justified.*

We added the following clarification, "Among the limitations, these measures were reported to be especially oversensitive to extreme values (outliers)."

*Line 136: How?*

We explained this for each evaluation metric in the following paragraphs in the manuscript.

*Line 139: Please define all variables in this equation, including the N and i.*

We added the following to the sentence, " $\hat{y}_i$ and $y_i$ are the forecasted and observed values at day $i$ respectively, and N is total number of the observations during the validation period."

*Line 145: What do you mean by this? More sensitive to outliers?*

By "error", we were referring to the difference between the predicted and observed values ($|\hat{y}_i - y_i|$). Due to the squared operation, RMSE is therefore more sensitive to large errors.

*Line 147: Please state the ranges and optimal values for MAE and RMSE, like you did for $R^2$. (I know that it's probably obvious to most readers, but it's a small amount of additional text and worth specifying.)*

Actually, both MAE and RMSE depend on the raw value of response variable, $y$, which will vary from one study to another. They are better interpreted comparatively.

*Line 160: I don't know what this means.*

For the validation period, we calculated the $90^{th}$, $95^{th}$, and $99^{th}$ percentile streamflow values at each watershed. These are considered thresholds. If an observed daily streamflow exceeded this threshold, it would be considered an extreme event.

*Line 180: A buffering effect to what? What does the ocean "buffer".*

Anonymous: Okay, but what did you do with the Caret package? Did you use the Caret package to train the random forests, or just to loop through hyperparameters?

Anonymous: In this case, please add "as explained in the remainder of this section" to the end of the sentence.

Anonymous: For both MAE and RMSE, the range is always [0, inf) and the optimal value is 0. (Of course most models are good enough that they do not produce errors of infinity, but they could.)

We acknowledge the term "buffering effect" might be vague and revised the sentence to, "Proximity to the ocean creates a more moderate climate with a narrower temperature range, particularly in the winter."

*Line 221- 223: Predictors of what? Streamflow, or SWE?*

They were included as predictors for the RF model. All eight predictors are listed in Table 2.

*Line 223: Why only the last measurement of each day?*

We only supplied the last measurement from SNOTEL stations because not all predictors have sub-daily values.

*Line 225: How big are these basins? Please show a map.*

We added the following map and table to Supplementary material.

[Figure]

Figure 3: Map of basins within the Pacific Northwest Hydrological Unit. Blue basins contained at least one watershed and were included in the the study.

| HUC-6 | Name | Area $(km^2)$ | Number of SNOTEL |
|---|---|---|---|
| 170102 | Pend Oreille | 67598.70 | 30 |
| 170200 | Upper Columbia | 119755.57 | 10 |
| 170300 | Yakima | 15928.20 | 9 |
| 170401 | Snake Headwaters | 14812.20 | 11 |
| 170501 | Middle Snake-Boise | 85150.16 | 27 |
| 170601 | Lower Snake | 30198.02 | 7 |
| 170602 | Salmon | 36248.15 | 11 |
| 170603 | Clearwater | 24318.13 | 9 |
| 170701 | Middle Columbia | 29124.57 | 11 |
| 170703 | Deschutes | 27789.56 | 8 |
| 170800 | Lower Columbia | 16120.04 | 15 |
| 170900 | Willamette | 29697.66 | 15 |
| 171003 | Southern Oregon Coastal | 34510.01 | 6 |
| 171100 | Puget Sound | 52958.23 | 26 |
| 171200 | Oregon Closed Basins | 45143.34 | 6 |

*Line 227: Can you discuss how much this affects the accuracy of your model? It seems like a major caveat.*

This represents a shortcoming of the study due to limited spatial coverage of SNOTEL stations and the introduced uncertainty likely affects the accuracy of the model. We acknowledged this by

stating, "The SNOTEL averages, therefore, represent first-order estimates of snow coverage and temperature conditions." We also considered an alternative approach by drawing information only from the SNOTEL station closest to gauge location but decided the basin-average better represented SWE conditions. Using basin-average SNOTEL SWE is consistent with previous studies in that focused on streamflow forecast [Abudu et al., 2011] as well as contribution of snowmelt to streamflow [Zheng et al., 2018] in western USA. Nevertheless, we believe supplying the RF with a more spatially consistent SWE data would improve model accuracy and is certainly worthy of future research. The reported RF performance in our study might be an underestimation.

*Line 232: Why not use all predictors available? Random forests are not computationally expensive and perform predictor selection automatically.*

We believe the current selection is appropriate. We also had to consider the practical purpose of the model. While there are other variables we could include such as soil moisture content, many would not be available for 1-day ahead forecasting in real time.

*Line 235: Why use $T_{min}$ and $T_{max}$ as predictors if their only influence is on SWE (another predictor)? In that case you should just use SWE.*

It's worth mentioning that these predictors are at 1-day lag. SWE only reflects the current state of snow condition and melting of snow is often triggered by changes in temperature. Given that there is high temporal correlation in daily temperatures, $T_{min}$ and $T_{max}$ data can provide useful signal to our streamflow forecast.

*Line 237: This only tells me what a pentad is, not what the "pentad index" is. Please define the "pentad index".*

In this case, the "pentad index" refers to the numerical sequence of pentads in a calendar year (1 to 73).

*Line 239: What do you mean by "across gauges"? Is this correlation a spatial correlation, or is it computed at each gauge (in which case it's temporal but not spatial)?*

Temporal correlation between daily streamflow and Pentad Index was computed at each gauge here.

*Line 247: How? There are many ways to do min-max scaling. For example, what are the min and max values after standardization? Also, which dataset do you use to compute the min and max values for scaling? Just the training set, or both training and validation?*

We added the following clarification to the manuscript, "We standardized training and validation data at each gauge using min-max scaling. The new data for all variables have values between zero and one."

*Line 251: Random forests have many other hyperparameters: minimum sample size per split node, minimum sample size per leaf node, maximum depth, cost function, etc.).*

We were aware of this but preferred to focus on the two parameters discussed in [Breiman, 2001].

*What is a "sample of training data sets"? I thought you had only one training set (2009-15).*
Thanks for allowing us to clarify this. We tested RF on training data sets of 30 randomly chosen watersheds and observed that the reduction in error is negligible after 2000 trees. Then we set the number of trees to 2000 for all watersheds.

*Line 256: Why optimize for MAE, instead of one of the other scores you looked at (RMSE, $R^2$, or KGE)?*

We actually optimized using both MAE and RMSE. The results were similar except for a few watersheds. The reason we moved forward based on MAE results was because RMSE penalizes larger errors and it was our interest to minimize the average errors, not large errors. KGE and $R^2$, on the other hand, do not directly capture the magnitude of errors but rather the overall performance of the model. They are also not commonly used in parameter tuning.

*Line 261: On line 95 you said that the default is M/3, where M = number of predictors.*
Yes, we have 8 predictors so round-up of 8/3 is 3.
*Anonymous: Is this shown in the figures?*
Yes, this is shown in figure 5a with correlation coefficient of RF (y-axis) plotted against correlation coefficient of naïve model (x-axis). For rainfall-driven and transient watersheds, most points lie on the left of the 1-to-1 line, suggesting RF outperforms naïve model. For snowmelt-driven watersheds, the points lie on the 1-to-1 line, which indicates there is marginal difference in the models' performance.
*Line 271: Is this shown in the figures?*
Yes.
*Line 274: ? I don't understand this sentence.*
* * *
**Margin comments:**

- Anonymous: Please state explicitly that this is a limitation of your work.
- Anonymous: Please put this sentence in the conclusion, along with other future work.
- Anonymous: Please put this in the main text.
- Anonymous: You still have not answered this question.
- Anonymous: Please clarify in the main text that there are other hyperparameters. Also, please justify why you experimented with only two hyperparameters.
- Anonymous: What do you mean by "error"? Out-of-bag MAE? Whatever the answer is, please state it in the main text.
- Anonymous: Please put this justification in the main text.
- Anonymous: Yes, in this case M/3 and ceil(sqrt(M)) are the same, but in general they are not. So my comment remains.
- Anonymous: Please put this explanation in the text.

Sorry for the typo. The sentence should read, "Without accounting for persistence, it would be inadequate to conclude that RF delivered better performance compared to the other two groups."

*Line 275: How many of these differences are statistically significant? In general, all comparisons between two models should be accompanied by a significance test.*

We addressed this comment in the Major comments section.

*Line 283: Could you verify this hypothesis by analyzing the data?*

Yes, the hydrographs of snowmelt-driven watersheds tend to be less flashy compared to rainfall-driven watersheds.

*Line 287: How can large errors and mean errors have the same distribution? By definition, large errors are greater than mean errors.*

By "distribution", we were referring to the dispersions of the RMSE and MAE score values for 3 groups in Figure 6b. For example, the MAE scores are heavily skewed towards 0 while RMSE scores are more evenly spread among snowmelt-driven watersheds.

*Line 293: The opposite of "poor" is not "satisfactory". Does the Rogelis paper define other ranges of KGE as "fair," "good," "excellent," etc.? Or does it just say that the 0-0.5 range is "poor"?*

As we explained above, these scores should be evaluated comparatively rather than absolutely.

*Line 295: ? Define.*

We addressed this comment above.

*Line 299: Is this a fair comparison? The sets of watersheds is your paper vs. Tongal and Booij are completely different, no?*

You are right. For this very reason, we simply reported the KGE scores in our study and theirs without making a comparison of the two models.

*Line 301: Figure 7 should be plotted on a performance diagram. This would allow you to show POD, FAR, frequency bias, and CSI all in the same figure. For example, see Figure 12 in this paper: https://journals.ametsoc.org/waf/article/35/4/1523/347594*

We think this is a good suggestion. We included here the relative operating characteristic (ROC) plot, which measures the ability of forecast model to discriminate between events and no-events across thresholds. This is similar to the performance diagram in the paper you referenced. We will modify the discussion on POD and FAR based on this new plot.

[Figure]

Figure 4: Probability of detection plotted is plotted against false alarm rate for three extreme thresholds: 90th, 95th, and 99th percentiles.

*Line 302: What is the actual value corresponding to each percentile?*

The actual values corresponding to each of the three thresholds varied across watersheds. Because of this, we did not record the actual values them and focused on the FAR and POD values.

*Line 308: This hypothesis seems like just a guess. Can you verify it by looking at the data (i.e., explicitly looking at predictions for cases with large surges vs. cases without large surges)?*

This is not a guess but suggested by our understanding of the hydrology of the region, examination of hydrographs, and the POD rate of RF among snowmelt-driven watersheds. The large surges of runoff from these watersheds likely occur during spring and early summer (March-June in Figure 2d).

Anonymous: Please put this in the main text, to clarify that the statement "there is less variability in flow behaviors at individual gauges in this group" is backed up by data, rather than being pure conjecture.

Anonymous: Please clarify this in the main text. Without the clarification, I imagine a lot of readers will have the same question I did.

Anonymous: The ROC curve and performance diagram are very different things. The ROC curve plots POD vs. false-alarm rate (sometimes called probability of false detection or POFD), which is b / (b + d) in the contingency table. The performance diagram plots POD vs. false-alarm *ratio*, which is b / (a + b) in the contingency table. False-alarm *rate* and *ratio* tend to be very different for rare events, because b + d (the number of actual non-events) tends to be much greater than a + b (the number of forecast events). Thus, models with very good ROC curves often have poor performance diagrams. This is why it is crucial to show both.

*Line 309: What does this mean? FAR and POD measure very different things, so what does it mean for them to be "in agreement"?*

Although POD and FAR provide different measurements, high POD and low FAR suggest skillful forecast. This is shown on slide 25 here (`https://www.nws.noaa.gov/oh/hrl/hsmb/docs/hep/events_announce/STEWksp_Training_Hydro_Verification_30Nov06.pdf`).

*Line 311: You don't know this until you have calculated frequency bias (which is shown in performance diagrams).*

We're not sure how "frequency bias" is calculated as the paper cited above did not mention it. However, based on our ROC plot, if there were systematic overestimation, FAR would be exceed POD and the ROC curve would fall below the no-skill line (slide 38 in the document from NOAA we referenced above).

> Anonymous: You cannot detect systematic overestimation from a ROC curve. Systematic overestimation occurs when frequency bias > 1, and frequency bias = (a + b) / (a + c), where a = number of true positives; b = number of false positives; and c = number of false negatives. In other words, frequency bias is (number of forecast events) / (number of actual events).
>
> Frequency bias can be contoured in the background of a performance diagram, which is another reason that I've requested you include performance diagrams along with ROC curves.

[revised manuscript text omitted]

Anonymous: less

Anonymous: Insert colon here.

Anonymous: Do not capitalize.

Anonymous: The fractions are awkward. Please replace with "0.0625" and "0.05".

Anonymous: comma

Anonymous: comma

performance in these watersheds where streamflow contributions come from a mix of snowmelt and rainfall, as well as where snowmelt dominates sources. Considering the prominent role of snowpack in water management and contribution of rapid snowmelt in flood events, such question is worth exploring. To this end, we evaluate the potential of RF in making short-term streamflow forecast at 1-day lead time across 86 watersheds in the Pacific Northwest Hydrologic Region (Fig. 2). The U.S. Geological Survey (2020) defines this region as hydrologic region 17 or HUC 17. HUC-17 consists of sub-basins and watersheds of the Columbia River that span varying hydrologic regimes. The selected watersheds have long-term record of unregulated streamflow and different streamflow contributions of rainfall and snowmelt. Drainage basin factors such as topography, vegetation, and soil can affect the response time and mechanisms of runoff (Dingman, 2015). Few studies attempted to account for or report these effects on models' performance. Without such consideration, it is difficult to determine if a data-driven model can be generalized to watersheds not included in the given study. Therefore, our objectives are (1) to examine and compare the performance of RF in a number of watersheds across hydrologic regimes and (2) to explore the role of catchment characteristics in model performance that are overlooked in previous studies.

In practice, RF can be trained to forecast streamflow at various timescales, depending on the selection of input variables. We focus on 1-day lead time because we assume only antecedent information for predictors are available at the time forecast is made. At longer lead times, changes in weather conditions would likely exert much greater control on runoff and the performance of the model.

We select RF to forecast streamflow for two reasons. First, RF has been referenced to deliver high performance in short-term streamflow forecasts (Mosavi et al., 2018; Papacharalampous and Tyralis, 2018; Li et al., 2019; Shortridge et al., 2016), making it a good candidate for our study. Second, RF allows for some level of interpretability compared with other ML models. This is delivered through two measures of predictive contribution of variables: mean decrease in accuracy (MDA) and mean decrease in node impurity (MDI). These two measures have been widely used as means for variable selection in classification and regression studies in bioinformatics (Chen and Ishwaran, 2012), remote sensing classification (Pal, 2005), and flood hazard risk assessment (Wang et al., 2015). This can be considered an advantage of RF compared with the more "black-box" nature of competing ML algorithms. While the referred interpretability does not directly translate to interpretation of the physical processes, it can provide insight into relationships among predictor and streamflow response.

The remainder of the paper is arranged as follows. Section 2 provides a brief introduction to RF , relevant parameters, and selected evaluation  criteria. Section 3 describes the study area, datasets, and predictor selection. Results and discussion are given in Sect. 4 along with limitations and recommendation for future research. A summary and indication of future work  is provided in Sect. 5.

> Anonymous: Please replace with "depending on the input variables provided". The current phrasing makes it unclear whether "the selection of input variables" is done by the user or by the random forest.

**2 Methodology**

**2.1 Random forests**

Proposed by Breiman (2001), RF is a  supervised, non-parametric algorithm within the decision tree family that comprises an ensemble of  decorrelated trees to yield prediction for classification and regression tasks. Since a single decision tree can produce high variance and is prone to noise (James et al., 2013), RF addresses this limitation by generating multiple trees where each tree is built on a bootstrapped sample of the training data. Each time a binary split is made in a tree (also known as split node), a random subset of predictors (without replacement) from the full set of predictor variables is considered. One predictor from these candidates is used to make the split where the expected sum variances of the response variable in the two resulting nodes is minimized. The randomization process in generating the subset of the features prevents one or more particularly strong predictor from getting repeatedly chosen at each split, resulting in highly correlated trees (James et al., 2013). After all the trees are grown,  the forests make prediction on a new data point by having all trees run through the predictors. In the end, the forests cast a majority vote on a label class for classification task or produce a value for regression task by averaging all predictions. Breiman (2001) provided full details on RF and its merit. The `randomForest` package in R developed by Liaw et al. (2002) was used for model training and validation in our study. The step-by-step of building a regression RF follows:
* * *
**Algorithm 1** Building a regression RF
* * *
**Step 1:** $n$ bootstrap samples are drawn from training set, each has the same size as the training sample. This is also known as `ntree` or number of trees in the forest.

[revised manuscript text omitted]

$$RMSE = \sqrt{\frac{\sum\limits_{i=1}^{N}(\hat{y}_i - y_i)^2}{N}} \tag{5}$$

KGE metric ranges between  $-\infty$ and 1. While there currently is not a definitive KGE scale,
175  Knoben et al. (2019) showed KGE values in the range between -0.41 and 1 indicate the model improves upon the mean flow benchmark, which assumes the predicted streamflow values equal to the mean of all observations. KGE value of 1 suggests the model can perfectly reproduce observations. KGE is calculated as follows:

$$KGE = 1 - \sqrt{(r-1)^2 + (\alpha-1)^2 + (\beta-1)^2} \tag{6}$$

180    where $r$ is the correlation coefficient, $\alpha$ is a measure of relative variability in the forecasted and observed values, and $\beta$ represents the bias:

Anonymous: Please insert "Pearson" here, since you have been calling it the "Pearson correlation coefficient" elsewhere.

$$\alpha = \frac{\sigma_{\hat{y}}}{\sigma_y} \quad \text{and} \quad \beta = \frac{\mu_{\hat{y}}}{\mu_y} \tag{7}$$

where $\sigma_{\hat{y}}$ is the standard deviation in observations, $\sigma_y$ is the standard deviation in forecasted values, $\mu_{\hat{y}}$ is the forecasted mean, and $\mu_y$ is observation mean.

185    In hydrological forecast, one might be interested in the ability of the model to capture more extreme events rather than the overall performance. The definition of "extreme" depends on the objective of the study. Here, we adopt the peak-over-threshold method  For the validation period, we calculated the 90th, 95th, and 99th percentile streamflow values at each watershed. These are considered thresholds. If an observed daily streamflow exceeded this threshold, it would be considered an extreme event. We
190 measure the ability of RF to capture these events using two additional criteria: probability of detection (POD) and false alarm rate (FAR). The calculation followed as in (Karran et al., 2013).

$$POD = \frac{P(\hat{y}_i > \omega | y_i > \omega)}{P(y_i > \omega)} \tag{8}$$

and

$$FA = \frac{P(\hat{y}_i > \omega | y_i < \omega)}{P(y_i < \omega)} \tag{9}$$

where $\omega$ is a specified threshold.

**3 Study Area and data**

**3.1 Watersheds in the Pacific Northwest Hydrologic Region**

In this study, we focus on watersheds in the Pacific Northwest Hydrologic Region . This region covers an area of 836,517 km$^2$ and encompasses all of Washington, six other states, and British Columbia, Canada. For the purpose of maintaining consistency in monitoring protocol and data, we only consider watersheds on the US territory. The Columbia River and its tributaries make up the majority of the drainage area, traveling more than  2000 km with an extensive network of more than 100 hydroelectric dams and reservoirs have been built along these river channels. Hydropower in the Columbia River Basin supplies approximately 70 percent of Pacific Northwest energy (Payne et al., 2004). Flood control is also an important aspect of reservoir operation in this region.

The north-south running Cascade Mountain Range divides the region into eastern and western parts and strongly influence the regional climate. The western windward side of the mountain receives an ample amount of winter precipitation compared to the leeward side. When temperature falls near freezing point, precipitation comes in the form of snow and provides water storage for dry summer months. Summers tend to be cool and comparatively dry. East of the Cascades, summer rainfall result from rapidly built thunderstorm and convective events that can produce flash floods (Mass, 2015).  Proximity to the ocean creates a more moderate climate with a narrower seasonal temperature range, particularly in the winter. Spatial trends and variations in annual mean temperature, mean precipitation, drainage area, and elevation of the watersheds are shown in Fig. 3.

**3.2 Data**

**3.2.1 Streamflow**

Our analysis uses streamflow data available through the USGS National Water Information System (NWIS) (https://waterdata.usgs.gov/nwis/sw). From NWIS, we selected daily streamflow time series for gauges using the following criteria: 1) continuous operation during the 10-year period between 2009 and 2018, 2) have less than 10 percent of missing data, and 3) positioned in watersheds with "natural" flow that is minimally interrupted by anthropogenic intervention such reservoirs. The third criterion was met using the GAGES-II: Geospatial Attributes of gauges for Evaluating Streamflow dataset (Falcone, 2011) classification to identify watersheds with least-disturbed hydrologic condition and represented natural flow.  We performed additional screening by computing correlation coefficient between the respective gauge and mean basin streamflow and removed those with a correlation of less than 0.5. We also excluded small creeks with drainage area less than 50 km$^2$. In total, 86 watersheds were selected (Fig. 2).

Anonymous: Figure 2 should be referenced much earlier (at the beginning of this section and also in the introduction, where you first mention the HUC 17 region). It is difficult to understand all this description without a map.

(I also made this comment in round 1.)

Anonymous: Replace with "windward (west)".

Anonymous: Replace with "leeward (east)".

Anonymous: More moderate than what?

Anonymous: Does this statement apply to the whole domain, just the west side of the Cascades, or what?

Anonymous: Delete.

[revised manuscript text omitted]

Anonymous: I think you mean MDA?

Anonymous: According to Figure 8, for snow-melt-dominated watersheds, pentad is either the 3rd- or 4th-most important predictor.

Anonymous: Does "in this group" mean "for snow-melt-dominated watersheds"?

Anonymous: is

Anonymous: was

Anonymous: Do you mean the PDF (probability-density function)?

Anonymous: Why would MDA, but not MDI, underestimate the importance of variables with a non-normal distribution?

Anonymous: What is a "potential" predictor variable?

Anonymous: This sentence is a non-sequitur. In the previous sentence you talked about normal vs.

correlated variables (Gregorutti et al., 2017). There is also an ongoing discussion regarding the stability of both measures across different datasets (Calle and Urrea, 2010; Nicodemus, 2011; Ishwaran and Lu, 2019). Although results from MDI make more sense in our case, we suggest RF users to exert caution when interpreting outputs from these two measures.

**4.6 Effects of watershed characteristics on model performance**

To explore the role of catchment characteristics such as geology, topography, and land cover on the performance of RF model, we perform Pearson correlation test between the KGE scores and selected basin physical characteristics for each flow regime. These watershed characteristics were compiled as part of GAGES-II dataset using national data sources including US National Land Cover Database (NLCD) 2006 version, 100m-resolution National Elevation Dataset (NED), and Digital General Soil Map of the United States (STATSGO2) (Table 1 in the Supplement). The results are shown in Table 5. There is a strong negative correlation ($p < 0.05$) between KGE scores and watershed slopes among rainfall-dominated and transient watersheds (Fig. 9b). As steeper hillslope often associates with faster surface and subsurface water movement during event-flow runoff, this can result in shorter response time. We observe a similar trend between KGE scores and percent of sand in the soil (Fig. 9a) where the RF performs worse in watersheds with higher hydraulic conductivity (i.e., higher sand content). This could be a result of rapid subsurface flow from soil profile enabled by soil macropores in mountainous forested area (Srivastava et al., 2017), where subsurface flow is the predominant mechanism. Without a quantification of the partition of discharge into surface flow and subsurface flow at individual watersheds, it is difficult to determine the relative importance of subsurface runoff mechanisms in regulating streamflow and how that may have affected the RF performance. The findings, however, suggest RF performance can deteriorate at watersheds with quick-response runoff when supplied with 1-day delayed observation data.

It appears that stream density and the amount of vegetation cover may also affect the performance of RF, but the relationships are not statistically significant at $\alpha = 0.05$. Aspect eastness, darainage area, and basin compactness are not determining factors to variability in the KGE scores. We also explored the impact of land-use/land-cover, which can be represented by  of impervious cover in each watershed. However, because we only selected unregulated watersheds that experienced minimal human disruption during the initial screening, most watersheds have very little impervious cover (less than 5 %). It is noted that these selected characteristics are not meant to be exhaustive, but rather representative of various types of factors that could help explain the variability in model performance. Furthermore, an alternative approach to Pearson's correlation is to use ANOVA to test for marginal significance of each catchment variable to KGE while accounting for their interaction. Because our objective is not to make inference on KGE based on these variables and ANOVA analysis can be complicated to interpret, we choose to compute correlation coefficient .

**4.7 Limitations and future research**

There are some notable limitations in our study as well as RF in general. The classification of watersheds into three flow regimes was based on the timing of the climatological mean of the annual flow volume, which can fluctuate from year to year. This is particularly true for the watersheds in the transient group where streamflow is contributed by a mixture of runoff
* * *
non-normal distributions -- the normality of a distribution has nothing to do with the scale of measurement (e.g., if you multiply a [non-]normally distributed variable by 1000, it is still [non-]normally distributed). Also, why would you mention number of categories here? Both temperature and precipitation are continuous, not categorical, variables.

Anonymous: Are you suggesting that the two temperature variables (min and max) have more correlation with other predictors than do the two precip variables (1-day and 3-day)? If so, have you verified this by computing the correlations in your dataset?

Anonymous: What do you mean by the "stability [of a measure] across different datasets"?

Anonymous: drainage

Anonymous: Replace with "land use and land cover".

[revised manuscript text omitted]

The following materials are submitted with the manuscript, *"Evaluation of random forests for short-term daily streamflow forecasting in rainfall and snowmelt-driven watersheds."*

1. Table 1. USGS Gaging stations used in the study, classified regime, and selected physical characteristics that were used to compute Pearson correlation coefficient presented in Table 5 in the manuscript.

2. Figure 1. Delineation of basins (USGS HUC-6) within the Pacific Northwest Hydrologic Region.

3. Table 2. HUC-6 basins, drainage area, number of available SNOTEL stations, and elevation range of the SNOTEL stations.

Table 1: USGS Gaging stations used in the study, classified regime, and selected physical characteristics.

| Station ID | Regime | Drainage area ($km2$) | Compactness | Mean elevation (m) | % slope | Aspect eastness | Stream density | % sand in soil | % forested area | % impervious cover |
|---|---|---|---|---|---|---|---|---|---|---|
| 10396000 | Snowmelt dominant | 528.9 | 1.92 | 1890.5 | 13.7 | -0.78 | 0.81 | 25.18 | 14.31 | 0.12 |
| 12043300 | Rainfall dominant | 135.2 | 1.97 | 238.5 | 24.5 | -0.98 | 0.78 | 19.87 | 58.83 | 0.1 |
| 12048000 | Transient | 405 | 2.22 | 1265.7 | 46.2 | 0.4 | 0.56 | 58.75 | 71.13 | 0.04 |
| 12054000 | Rainfall dominant | 171.7 | 1.69 | 1073.8 | 54.2 | 0.77 | 0.53 | 55.07 | 74.9 | 0.09 |
| 12056500 | Rainfall dominant | 147 | 1.93 | 990.6 | 53.1 | -0.36 | 0.54 | 51.14 | 82.06 | 0.06 |
| 12060500 | Rainfall dominant | 198.3 | 1.46 | 601.9 | 42.5 | 0.21 | 0.74 | 41.62 | 78.88 | 0.63 |
| 12079000 | Rainfall dominant | 224.1 | 1.31 | 437.9 | 21.1 | 0.2 | 0.75 | 34.18 | 59.36 | 1.27 |
| 12082500 | Transient | 350 | 2.03 | 1182.5 | 33.9 | -0.86 | 0.8 | 58.3 | 68.05 | 0.43 |
| 12092000 | Transient | 240.9 | 2.84 | 1420.3 | 39.7 | -0.88 | 0.67 | 67.42 | 59.01 | 0.39 |
| 12094000 | Transient | 205.2 | 1.77 | 1263.9 | 41 | -0.28 | 0.77 | 62.39 | 69.29 | 0.17 |
| 12095000 | Rainfall dominant | 205.8 | 2.8 | 680.3 | 23 | -0.66 | 0.74 | 40.85 | 65.38 | 1.16 |
| 12096500 | Transient | 1142.8 | 2.34 | 812.6 | 25.3 | -0.65 | 0.67 | 51.96 | 57.65 | 2.5 |
| 12097500 | Transient | 190.2 | 2.07 | 1214.1 | 35.7 | -0.64 | 0.69 | 44.47 | 79.78 | 0.53 |
| 12097850 | Transient | 970.4 | 2.51 | 1262.7 | 38.2 | 0.21 | 0.75 | 46.52 | 72.32 | 0.51 |
| 12108500 | Rainfall dominant | 71.1 | 1.59 | 262.2 | 6.3 | -0.86 | 0.52 | 41.33 | 28.96 | 6.79 |
| 12114500 | Transient | 66.6 | 2.93 | 1073.5 | 37.1 | -0.98 | 0.81 | 51.54 | 77.27 | 0.75 |
| 12141300 | Transient | 401.5 | 2.05 | 1038.5 | 50.2 | -0.99 | 0.7 | 60.57 | 76.88 | 0.06 |
| 12142000 | Rainfall dominant | 165.6 | 2.76 | 931.7 | 44.4 | -0.99 | 0.67 | 57.48 | 77.76 | 0.61 |
| 12143400 | Transient | 107.9 | 1.9 | 1042.5 | 43.4 | -0.92 | 0.64 | 68.02 | 69.04 | 1.52 |
| 12143600 | Transient | 165 | 1.47 | 966.3 | 43.6 | -0.91 | 0.7 | 65.74 | 70.34 | 1.56 |
| 12144000 | Transient | 210.1 | 1.19 | 827.9 | 38 | -0.48 | 0.68 | 60.42 | 70.66 | 2.33 |
| 12145500 | Rainfall dominant | 79 | 2.13 | 460.9 | 19 | -0.57 | 0.87 | 38.49 | 73.84 | 1.23 |
| 12167000 | Rainfall dominant | 683.8 | 1.64 | 655.4 | 32.6 | -0.89 | 0.59 | 42.41 | 78.89 | 0.54 |
| 12179900 | Transient | 128.3 | 2.38 | 1110.1 | 57.2 | 0.02 | 0.54 | 43.96 | 59.73 | 0.11 |
| 12186000 | Transient | 398.4 | 1.92 | 1176 | 52.8 | -0.63 | 0.63 | 43.56 | 76.4 | 0.13 |
| 12189500 | Transient | 1855.3 | 1.88 | 1150.9 | 47.3 | -0.78 | 0.58 | 44.01 | 74.44 | 0.25 |
| 12201500 | Rainfall dominant | 224.2 | 1.42 | 274.9 | 17.6 | -0.92 | 0.73 | 46.03 | 65.32 | 1.82 |
| 12209490 | Rainfall dominant | 58.1 | 1.92 | 932.9 | 37.8 | -0.95 | 0.62 | 42.53 | 66.85 | 0.62 |
| 12210000 | Rainfall dominant | 332.1 | 2.37 | 838.1 | 35.4 | -0.94 | 0.64 | 42.68 | 71.42 | 0.35 |
| 12210700 | Transient | 1524.8 | 1.81 | 873.5 | 36.2 | -0.91 | 0.69 | 43.63 | 69.46 | 0.5 |
| 12323670 | Snowmelt dominant | 102.4 | 2.27 | 2222.5 | 29 | 0.95 | 0.72 | 37.24 | 46.62 | 0.16 |
| 12354000 | Snowmelt dominant | 828.4 | 2.14 | 1383.4 | 34.4 | 0.45 | 0.68 | 35.36 | 85.62 | 0.38 |
| 12358500 | Snowmelt dominant | 2939.2 | 1.12 | 1723.7 | 40.4 | -0.99 | 0.72 | 34.42 | 69.91 | 0.07 |
| 12374250 | Snowmelt dominant | 50.8 | 2.56 | 1408.3 | 24.9 | 0.98 | 0.84 | 41.65 | 94.87 | 0.14 |
| 12390700 | Snowmelt dominant | 470.2 | 2.23 | 1348.9 | 37.9 | 0.89 | 0.52 | 42.67 | 91.73 | 0.1 |
| 12392155 | Snowmelt dominant | 326.2 | 2.33 | 1389.8 | 38.3 | -0.42 | 0.61 | 45.05 | 87.93 | 0.09 |

Table 1: USGS Gaging stations used in the study, classified regime, and selected physical characteristics.

| Station ID | Regime | Drainage area ($km2$) | Compactness | Mean elevation (m) | % slope | Aspect eastness | Stream density | % sand in soil | % forested area | % impervious cover |
|---|---|---|---|---|---|---|---|---|---|---|
| 12448500 | Snowmelt dominant | 2683.6 | 1.83 | 1591.5 | 38.2 | 0.1 | 0.79 | 41.21 | 51.58 | 0.42 |
| 12451000 | Snowmelt dominant | 830.6 | 2.12 | 1534 | 55.8 | -0.61 | 0.67 | 49.69 | 48.34 | 0.1 |
| 12452800 | Snowmelt dominant | 526.4 | 1.7 | 1519.8 | 42.3 | -0.36 | 0.7 | 55.51 | 64.63 | 0.65 |
| 12458000 | Snowmelt dominant | 499.4 | 2.59 | 1547.5 | 47.9 | 0.9 | 0.67 | 49.12 | 61.75 | 0.09 |
| 12488500 | Snowmelt dominant | 205.2 | 1.77 | 1465.8 | 39.2 | 0.97 | 0.77 | 63.55 | 81.66 | 0.27 |
| 13010065 | Snowmelt dominant | 1222.3 | 1.11 | 2508.5 | 13.7 | -0.8 | 0.49 | 46.95 | 41.54 | 0.01 |
| 13162225 | Snowmelt dominant | 76.1 | 2.65 | 2474.2 | 40.6 | -0.59 | 0.69 | 34.96 | 47.25 | 0.25 |
| 13185000 | Snowmelt dominant | 2154.4 | 1.9 | 1955.1 | 36.5 | -1 | 0.79 | 56.11 | 43.97 | 0.11 |
| 13235000 | Snowmelt dominant | 1163.2 | 1.39 | 2079 | 39.9 | -0.97 | 0.75 | 62.57 | 52.91 | 0.12 |
| 13237920 | Snowmelt dominant | 874.8 | 1.39 | 1658.2 | 29.1 | -0.41 | 0.83 | 73.13 | 77.43 | 0.11 |
| 13296500 | Snowmelt dominant | 2090.9 | 1.17 | 2375.2 | 28.8 | 0.44 | 0.87 | 40.56 | 61.14 | 0.11 |
| 13309220 | Snowmelt dominant | 2696.6 | 1.55 | 2192.4 | 33.4 | 1 | 0.83 | 61.53 | 59.36 | 0.04 |
| 13313000 | Snowmelt dominant | 561.9 | 2.37 | 2180.4 | 25 | -0.54 | 0.76 | 70.04 | 79.39 | 0.07 |
| 13331500 | Snowmelt dominant | 618.9 | 1.15 | 1736.3 | 37.7 | -0.39 | 0.63 | 19.26 | 76.92 | 0.04 |
| 13334450 | Transient | 269.8 | 2.62 | 1271.8 | 30.3 | 0.89 | 0.82 | 20.5 | 51.92 | 0.21 |
| 13337000 | Snowmelt dominant | 3053.4 | 1.08 | 1584.1 | 33.2 | -0.9 | 0.72 | 40.18 | 86.37 | 0.07 |
| 13338500 | Snowmelt dominant | 3027 | 1.28 | 1384.6 | 21.1 | 0.57 | 0.88 | 29.01 | 70.53 | 0.17 |
| 13340600 | Snowmelt dominant | 3354.6 | 2.18 | 1442.7 | 33.9 | -1 | 0.72 | 40.39 | 75.57 | 0.05 |
| 14020000 | Transient | 341.4 | 2.22 | 1209.3 | 32.2 | -0.79 | 0.65 | 17.85 | 73.37 | 0.31 |
| 14020300 | Transient | 456.4 | 1.53 | 1187.3 | 28.4 | -0.87 | 0.79 | 17.1 | 67.74 | 0.42 |
| 14092750 | Transient | 57.5 | 2.64 | 1477.3 | 24.7 | 1 | 0.84 | 54.88 | 86.46 | 0.02 |
| 14096850 | Transient | 374.6 | 1.76 | 942.7 | 10.6 | 0.53 | 0.75 | 38.48 | 57.42 | 0.19 |
| 14107000 | Snowmelt dominant | 393.8 | 2.19 | 1428.8 | 22.6 | -0.05 | 0.84 | 40.91 | 74.79 | 1.29 |
| 14137000 | Transient | 674.2 | 2.54 | 1006.4 | 30.4 | -0.98 | 0.86 | 34.75 | 83.69 | 0.19 |
| 14141500 | Rainfall dominant | 59.9 | 1.45 | 724.7 | 18.2 | -0.96 | 0.65 | 25.72 | 84.58 | 0.01 |
| 14150800 | Rainfall dominant | 113.3 | 2.26 | 765.5 | 26.5 | -1 | 0.62 | 22.65 | 86.26 | 0.09 |
| 14154500 | Rainfall dominant | 546.8 | 2.41 | 857.5 | 33.3 | -0.5 | 0.68 | 24.3 | 84.93 | 0.04 |
| 14158500 | Transient | 237.1 | 1.82 | 1253.9 | 16.7 | -0.94 | 0.46 | 42.91 | 80.99 | 0.15 |
| 14159200 | Transient | 414.3 | 1.97 | 1280.8 | 28.7 | -0.66 | 0.7 | 34.93 | 92.32 | 0.02 |
| 14161500 | Rainfall dominant | 62.4 | 2.7 | 982.7 | 33.3 | -0.97 | 0.55 | 31.64 | 93.6 | 0.02 |
| 14166500 | Rainfall dominant | 226.5 | 1.73 | 253.8 | 19.1 | 0.18 | 0.58 | 15.62 | 57.02 | 0.94 |
| 14179000 | Transient | 272.5 | 1.73 | 1149.5 | 34.9 | -0.88 | 0.85 | 43.02 | 85.41 | 0.05 |
| 14180300 | Rainfall dominant | 66.6 | 2.72 | 1027.9 | 26.5 | -0.6 | 0.62 | 32.94 | 87.63 | 0.08 |
| 14182500 | Rainfall dominant | 286.8 | 1.73 | 818.3 | 35.9 | -0.62 | 0.68 | 31.45 | 84.37 | 0.06 |
| 14185000 | Rainfall dominant | 458.2 | 2.4 | 889.5 | 30.6 | -0.89 | 0.69 | 27.98 | 83.15 | 0.08 |
| 14185900 | Rainfall dominant | 258.2 | 2.11 | 918.7 | 37.8 | -0.91 | 0.68 | 29.52 | 82.16 | 0.05 |

Table 1: USGS Gaging stations used in the study, classified regime, and selected physical characteristics.

| Station ID | Regime | Drainage area ($km2$) | Compactness | Mean elevation (m) | % slope | Aspect eastness | Stream density | % sand in soil | % forested area | % impervious cover |
|---|---|---|---|---|---|---|---|---|---|---|
| 14187000 | Rainfall dominant | 134.7 | 2.56 | 732.7 | 28.6 | -0.09 | 0.63 | 24.89 | 71.81 | 0.02 |
| 14216000 | Transient | 594.6 | 1.84 | 1080.2 | 20.8 | -0.91 | 0.75 | 36.93 | 83.55 | 0.97 |
| 14216500 | Transient | 349.5 | 2.33 | 921 | 30 | -0.09 | 0.82 | 59.62 | 60.01 | 1.07 |
| 14219000 | Rainfall dominant | 167.4 | 1.79 | 687 | 30.2 | -0.5 | 0.51 | 32.49 | 67.64 | 0.67 |
| 14222500 | Rainfall dominant | 323.9 | 1.85 | 573.3 | 25.1 | -0.97 | 0.68 | 25.62 | 69.44 | 0.43 |
| 14231000 | Transient | 1378 | 1.94 | 1128.6 | 38.3 | -0.99 | 0.82 | 60.56 | 76.23 | 0.5 |
| 14236200 | Rainfall dominant | 361 | 1.79 | 672.9 | 33.1 | -0.84 | 0.58 | 29.76 | 61.65 | 1.02 |
| 14308990 | Rainfall dominant | 167.8 | 2.59 | 917.6 | 26.5 | -0.36 | 0.75 | 44.33 | 84.29 | 0.04 |
| 14309500 | Rainfall dominant | 224.9 | 2.4 | 736.3 | 33 | 0.97 | 0.66 | 30.53 | 82.45 | 0.12 |
| 14316495 | Rainfall dominant | 79 | 3.22 | 1207.1 | 37.4 | -0.5 | 0.75 | 33.75 | 83.85 | 0.03 |
| 14316700 | Rainfall dominant | 587.9 | 2.76 | 944 | 34.4 | -0.36 | 0.69 | 28.62 | 87.24 | 0.04 |
| 14318000 | Rainfall dominant | 459.5 | 2.23 | 858.9 | 27.1 | -0.26 | 0.71 | 27.3 | 85.85 | 0.04 |
| 14325000 | Rainfall dominant | 443.1 | 2.27 | 651.9 | 28.4 | -0.31 | 0.65 | 32.43 | 75.87 | 0.31 |
| 14400000 | Rainfall dominant | 702.6 | 2.39 | 671.2 | 36.5 | -0.88 | 0.61 | 22.5 | 57.15 | 0.1 |

[Figure]

Figure 1: Delineation of basins (USGS HUC-6) within the Pacific Northwest Hydrologic Region. Blue basins contain at least one of the 86 chosen watershed included in the study.

Table 2: Basin names, drainage area, number of available SNOTEL stations, and elevation range of the SNOTEL stations.

| HUC-6 | Basin name | Area ($km^2$) | Number of SNOTEL | Elevation range (m) |
|---|---|---|---|---|
| 170102 | Pend Oreille | 67598.70 | 30 | 4350-8250 |
| 170200 | Upper Columbia | 119755.57 | 10 | 3590-6490 |
| 170300 | Yakima | 15928.20 | 9 | 3430-5920 |
| 170401 | Snake Headwaters | 14812.20 | 11 | 6770-9820 |
| 170501 | Middle Snake-Boise | 85150.16 | 27 | 4800-8360 |
| 170601 | Lower Snake | 30198.02 | 7 | 4000-5760 |
| 170602 | Salmon | 36248.15 | 11 | 5350-9150 |
| 170603 | Clearwater | 24318.13 | 9 | 4600-6320 |
| 170701 | Middle Columbia | 29124.57 | 11 | 3310-5580 |
| 170703 | Deschutes | 27789.56 | 8 | 3810-5850 |
| 170800 | Lower Columbia | 16120.04 | 15 | 2140-5800 |
| 170900 | Willamette | 29697.66 | 15 | 2420-4950 |
| 171003 | Southern Oregon Coastal | 34510.01 | 6 | 3240-6050 |
| 171100 | Puget Sound | 52958.23 | 26 | 2250-5130 |
| 171200 | Oregon Closed Basins | 45143.34 | 6 | 5250-7660 |

---

## Referee Report (RR2)

Once again, I find the manuscript much-improved.  I have a handful of minor comments and two major comments.  As usual, the major comments are listed below, while the minor comments are made in the manuscript itself (see attached file).

**Major comments**

1. To me, the discussion at the end of Section 4.5 is still very unclear.  See inline comments.

2. The authors neglected to make a lot of the recommended changes in round 2, and they did not make this clear.  If you disagree with a recommendation, you are free to push to back at me.  After all, that's part of the review process.  However, I find it disrespectful that the authors just ignored a handful of comments.  I put a lot of time into the review process, and I don't like having to sleuth around and figure out which comments the authors ignored.

[revised manuscript text omitted]

Anonymous: Replace with "at most" or "at most individual".

Anonymous: Is this shown anywhere in the paper, or does the conclusion from a separate analysis? I'm not asking for another figure, but if the conclusion comes from a separate analysis, please specify "(not shown)" at the end of the sentence.

Anonymous: Replace with "the r-value trend".

Anonymous: Insert comma after "Fig. 6".

Anonymous: Why use someone else's mean-flow benchmark, instead of computing the mean-flow benchmark for your own dataset? I imagine that this wouldn't take a lot of effort, and the mean-flow benchmark could differ a lot between your dataset and theirs.
Anonymous: I just read your response to reviewers, where you said the following:

"To clarify, the author derived and concluded that a Kling-Gupta efficiency (KGE) > −0.41 improves upon the mean-flow benchmark, which is the KGE achieved by always predicting the time-mean flow ('climatology') at any basin."

My response to that:

**Figure 8.** Streamflow daily forecast scores computed over the validation period for RF model in four metrics: R-squared, KGE, MAE, and RMSE.

[Figure]

**Table 4.** Descriptive statistics of the four criteria used to evaluate the overall performance of RF: $R^2$, KGE, MAE, and RMSE.

| Metric | Flow regime | Min | Q1 | Median | Q3 | Max |
|---|---|---|---|---|---|---|
| $R^2$ | Rainfall-dominated | 0.59 | 0.71 | 0.77 | 0.81 | 0.87 |
| | Transient | 0.57 | 0.71 | 0.80 | 0.87 | 0.99 |
| | Snowmelt-dominated | 0.88 | 0.95 | 0.97 | 0.98 | 0.99 |
| KGE | Rainfall-dominated | 0.64 | 0.78 | 0.84 | 0.87 | 0.92 |
| | Transient | 0.62 | 0.77 | 0.86 | 0.91 | 0.99 |
| | Snowmelt-dominated | 0.77 | 0.89 | 0.94 | 0.97 | 0.99 |
| MAE | Rainfall-dominated | 0.0061 | 0.0096 | 0.0131 | 0.0161 | 0.0245 |
| | Transient | 0.0070 | 0.0097 | 0.0109 | 0.0143 | 0.0189 |
| | Snowmelt-dominated | 0.0065 | 0.0087 | 0.0092 | 0.0114 | 0.0168 |
| RMSE | Rainfall-dominated | 0.0157 | 0.0241 | 0.0326 | 0.0395 | 0.0609 |
| | Transient | 0.0144 | 0.0227 | 0.0275 | 0.0331 | 0.0468 |
| | Snowmelt-dominated | 0.0160 | 0.0218 | 0.0270 | 0.0315 | 0.0436 |

"So the mean-flow benchmark is truly independent of the basin? If so, please make this clear in the manuscript. Otherwise, it's unclear why you're using someone else's mean-flow benchmark, rather than computing the benchmark for your own data."

**Figure 9.** The probability of detection (POD) plotted against the false alarm rate (FAR) for three extreme thresholds: 90th, 95th, and 99th percentiles. Thin black line connects values from the same watershed. (Vertical axis) Number of times RF *correctly* forecasted events that exceeded the threshold divided by the total number of exceedance. (Horizontal axis) Number of times RF *incorrectly* forecasted events that exceeded the threshold divided by the total number of non-exceedance.

[Figure]

Anonymous: Please explain (either in the caption or in the main body where you introduce Figure 9) that these ROC curves are different than typical ROC curves. i.e., Since you plot only 3 thresholds, the x-axis does not go all the way from 0 to 1. For people used to looking at ROC curves, like me, this looked very wrong until you explained it.

[revised manuscript text omitted]

---

## Referee Report (RR3)

Thank you again for your hard work in revising the paper.  I have only a few minor comments left, which all take the form of "put your response to reviewers in the main text of the paper, so that it's clear to the reader" (see below).

Again, we thank the reviewer 2 for valuable comments and suggestions. Please see below our responses to the comments. The reviewer's comments are in black font and our responses are in blue font. Changes in the manuscript are highlighted in red.

**1  Major comments**

**Major comment 1**. To me, the discussion at the end of Section 4.5 is still very unclear. See inline comments.

Please see below our responses to comments related to Section 4.5.

Line 378: Does "in this group" mean "for snow-melt-dominated watersheds"? (I made this comment in round 2 as well.)

Yes, this is the case and we did miss this comment. We added the clarification to the text, "Precipitation does not seem to have significant contribution to the model's accuracy among the snowmelt-dominated watersheds."

Lines 394-396: Why would MDA, but not MDI, underestimate the importance of variables with a non-normal distribution? (I made this comment in round 2 as well.)

We addressed this comment in our response to the reviewer in Round 2 (page 8). Here was our response, "*Precipitation data is generally zero inflated (at least 30 percent in our dataset depending on the watershed). As a result, there is a high likelihood that the day with zero precipitation ends up with the same value during the shuffling process used to compute MDA. While we did not perform additional simulation to explore this as it is out of the scope of our paper, we believe it is worth discussing and can be investigated in future research.*" This is the reason our conclusion is that, "*We suggest RF users to exert caution when interpreting outputs from these two measures.*"

The following discussion was added to the text after Round 1 revision, "*In our precipitation data (both training and validation), at least 30 percent of the daily observations are 0 across the watersheds. There is a high likelihood that the day with zero precipitation ends up with the same value during the shuffling process, thus potentially affecting the randomness created to compute MDA. While we did not perform additional simulation to further confirm whether MDA and MDI measures are sensitive to highly-skewed and zero-inflated variables, this can be a topic of future research.*"

Lines 400-403: But you standardized all the predictors to range from 0 to 1, right? So the random forest saw predictors only in normalized units, not in physical units. So the "scale of measurement" should have had no impact on your importance measures.

Actually, "scale of measurement" does not only refer to the numeric range but also the nature of the data (for example, ordinal vs continuous). You are correct that RF saw predictors only in standardized units, not in physical units. However, standardization and normalization do not change the nature of the data. For example, precipitation data is zero-inflated and the standardization does not make it more normally distributed like temperature data. These zero precipitation data points are just scaled into different values. We addressed this point in the previous revision (page 8). Here was our response, "*We agree that both variables, precipitation and temperature, are not categorical variables and removed "their number of categories" from the text. However, among our 8 predictors in our study, pentad is considered an ordinal variable. Also, the scales of measurement of precipitation and temperature variables are slightly different. Precipitation is a flux variable and comprises discrete and continuous components in that if it does not rain the amount of rainfall is discrete whereas if it rains the amount is continuous. Temperature is a state variable and always continuous. Therefore, we believe the findings in Strobl et al. (2007) are relevant for our discussion on variable importance.*" Again, the cited literature does suggest that the scale of measurement can have an impact on RF variable importance measures and we believe it's important that the readers are aware of this.

Line 401: What is a "potential predictor variable"? (I made this comment in round 2 as well.)

Sorry, we did miss this. "Potential" has been deleted from the text.

Lines 401-403: Are you suggesting that the two temperature variables (min and max) have more correlation with other predictors than do the two precip variables (1-day and 3 day)? If so, have you verified this by computing the correlations in your dataset? (I made this comment in round 2 as well.)

We addressed this comment in our previous response (page 8). Here was our response, "*Thanks for the opportunity to clarify this. Yes, temperature variables tend to have more correlation with other predictors than do the two precipitation variables in our dataset. This is likely because temperature controls both the form of precipitation (snowfall vs rainfall) and the timing of snowmelt. However, due to the blackbox nature of ML models, we don't know for sure if this is directly related to the observed*

Anonymous: Please explain this in the main text. What you mean by "scale of measurement" will likely not be obvious to the reader.

Anonymous: Please include this explanation in the main text as well.

*patterns in MDI and MDA*. Therefore, the key takeaway of this discussion was that, "*We suggest RF users to exert caution when interpreting outputs from these two measures.*" We added the following discussion to the manuscript, "In our study, temperature variables tend to have more correlation with other predictors than do the two precipitation variables. This is likely because temperature controls both the form of precipitation (snowfall vs rainfall) as well as the timing of snowmelt."

Line 403: I still don't know what you mean by "stability" here.

We added the following clarification, "There is also an ongoing discussion regarding the stability of both measures, in which the two variable importance measures can yield noticeably different rankings, in different simulated datasets (Calle and Urrea, 2010; Nicodemus, 2011; Ishwaran and Lu, 2019)".

**Major comment 2**. The authors neglected to make a lot of the recommended changes in round 2, and they did not make this clear. If you disagree with a recommendation, you are free to push to back at me. After all, that's part of the review process. However, I find it disrespectful that the authors just ignored a handful of comments. I put a lot of time into the review process, and I don't like having to sleuth around and figure out which comments the authors ignored.

We truly appreciate the reviewer's time in providing us with valuable comments and believe that the manuscript has improved thanks to the changes made in the two rounds of revision. It was not our intention to ignore the comments but we do acknowledge that we missed a handful of them in the revision process. We apologize for this. In this revision, we addressed all of comments point by point.

**2 Other inline comments**

Line 28: Replace with "less" (I made this comment in round 2 as well).

We agree that "less" is more appropriate and replaced "fewer" with "less" in the manuscript."

Line 35: Insert colon (I made this comment in round 2 as well).

Sorry we did miss this. The colon has been added.

Line 48: Do not capitalize (I made this comment in round 2 as well). The fractions are awkward. Please replace with "0.0625" and "0.05".(I made this comment in round 2 as well.)

We address to these comments in our response to the reviewer (page 7). Here was our response, "*We believe the current specification is consistent with the current literature on VIC model.*" In this paper published by HESS, Variable Infiltration Model (VIC) is capitalized and spatial resolution is in degrees (`https://hess.copernicus.org/articles/17/721/2013/`).

Lines 50-51: Insert comma (I made this comment in round 2 as well). Insert comma (I made this comment in round 2 as well).

We did miss these. The commas have been added.

Lines 81-82: The nested parentheses are awkward. Please rearrange the sentence in a way that gets rid of them.

We have modified the sentence," Both model-agnostic, such as permutation-based feature importance (Breiman, 2001), and model-specific, such as gini-based for RF (Breiman et al., 1984) and gradient-based for ANNs (Shrikumar et al., 2017), interpretation methods can provide useful insights into how the ML models make their predictions."

Line 87: Replace with "referred to".

We replaced "referred" with "referred to".

Line 99: Please explain this more clearly. I doubt that readers unfamiliar with random forests and ML will understand this. (I made the same comment in rounds 1 and 2, and it was ignored.)

We addressed this comment in our response in Round 2 revision (page 7). Here was our response in Round 2 revision, "*This is a bit vague for us to provide clarification. Our current explanation, "One predictor from these candidates is used to make the split where the expected sum variances of the response variable in the two resulting nodes is minimized," is consistent with the principle of regression tree explained in the Elements of Statistical Learning text (Friedman et al., 2001)*". We reviewed other papers on the applications of RF and believe that current explanation in the sentence is accurate and intuitive in terms of how a decision tree regression works. This is also illustrated mathematically in Algorithm 1 Step 3. We added this reference to the manuscript, "One predictor from these candidates is used to make the split where the expected sum variances of the response variable in the two resulting nodes is minimized (Algorithm 1, Step 3)."

Line 119: You need punctuation here. I suggest inserting a comma here, which means you will need to replace the colon before "ntree" with a comma.

The commas were added. We also removed the colon. Here is the new sentence, "While all considered parameters might have an effect on the performance of RF, we chose to focus on two parameters, `ntree` and `mtry`, for a number of reasons."

Line 121: What do you mean by this? Any hyperparameter is tunable.

The goal of tuning is to obtain the optimal value of the hyperparameter based on a criteria (lowest MAE for example). It has been theoratically proven (please see the cited literature) that more trees are always better (yielding lower MAE). In other words, optimal ntree value can go to infinity. The reduction in error, however, becomes negligible after a sufficiently large number of trees (we discussed this on lines 111-112). We added the following modification to the manuscript, "Second, `ntree` in a forest is a parameter that is tunable but not optimized and should be set sufficiently high (Oshiro et al., 2012; Probst et al., 2019) for RF to achieve good performance."

Line 121: I don't understand how this sentence justifies experimenting with ntree instead of the many other hyperparameters available.

In the manuscript, we explicitly stated that "*all considered parameters might have an effect on the performance of RF*" and we chose to focus on ntree and mtry. The main reason is that these two parameters were originally introduced by Breiman (2001) in the development of RF algorithm. In this sentence, we cited the literature which suggests that "*ntree should be set sufficiently high (Oshiro et al., 2012; Probst et al., 2019) for RF to achieve optimal performance*". In other words, a high number of trees is essential for optimal RF model performance. Yet what "sufficiently high" means will differ from one dataset to another. This is why we needed to experiment with ntree in our study.

Lines 187-188: After standardization, both the training and validation data range from 0 to 1, right?

This is not the case. Only training data range from 0 to 1. Validation data are considered new data and therefore can have values outside of this range.

Line 310: are

This has been fixed.

Line 315: Replace with "at most" or "at most individual".

We replaced "at individual watersheds" with "at most individual watersheds"

Line 315-316: Is this shown anywhere in the paper, or does the conclusion from a separate analysis? I'm not asking for another figure, but if the conclusion comes from a separate analysis, please specify "(not shown)" at the end of the sentence.

Yes, this is shown in Figure 7a and does not come from a separate analysis. Watersheds that yield lower $r$ values in the naive model are considered to have lower persistence (we defined persistence as the correlation between streamflow of day $t$ and $t+1$ on line 156-157). We added the following clarification to the manuscript, "In Fig. 7a, we observe most points lie on the left of the 1-to-1 line, suggesting that RF outperforms naïve model at most individual watersheds in rainfall-driven and transient regimes. We also discern that large improvement, defined as the positive difference in $r$ values between RF and naïve model, tends to occur with lower persistence (lower $r$ values from the naïve model)."

Line 328: Replace with "the r-value trend".

We replaced "$r$ values trend" with "$r$-value trend".

Line 328: Insert comma after "Fig. 6".

We added the comma.

Lines 345-346: Why use someone else's mean-flow benchmark, instead of computing the mean-flow benchmark for your own dataset? I imagine that this wouldn't take a lot of effort, and the mean-flow benchmark could differ a lot between your dataset and theirs.

I just read your response to reviewers, where you said the following: "To clarify, the author derived and concluded that a Kling-Gupta efficiency (KGE) > -0.41 improves upon the mean-flow benchmark, which is the KGE achieved by always predicting the time-mean flow ('climatology') at any basin."

My response to that: "So the mean-flow benchmark is truly independent of the basin? If so, please make this clear in the manuscript. Otherwise, it's unclear why you're using someone else's mean-flow benchmark, rather than computing the benchmark for your own data."

Yes, the mean-flow benchmark (such as the Nash–Sutcliffe efficiency (NSE and the KGE) are both independent of the basin. We made the following modification to the manuscript, "As observed mean flow is used in the calculation of KGE, Knoben et al. (2019) suggested that a KGE score greater

Anonymous: Please put this explanation in the main text. What you mean by "tunable but not optimized" will likely be unclear to the reader.

Anonymous: Please put this in the main text. Currently, your justification for experimenting only with ntree and mtry will likely be unclear to the reader.

than -0.41 indicates a hydrologic model improves upon the forecast with mean flow, independent of the basin."

Figure 9: Please explain (either in the caption or in the main body where you introduce Figure 9) that these ROC curves are different than typical ROC curves. i.e., Since you plot only 3 thresholds, the x-axis does not go all the way from 0 to 1. For people used to looking at ROC curves, like me, this looked very wrong until you explained it.

We have added the following clarification to the caption of the plot, "It is noted that the scales of the horizontal and vertical axes are not 1-to-1 in the plotted partial receiver operating characteristic (ROC) curve."

The full range, from min to max? I also asked this question in round 2, and it was not answered.

We addressed this question in Round 2 revision (page 9). And that is correct. The bar represents the full range of values. We added the clarification to the manuscript, "Figure 10. Barplots show importance of predictor variables using (a-c) MDA and (d-f) MDI criteria. Length of the blue bars indicates the median value across the watersheds for each flow regime and the thin black bar represents the full range of the values."

---

## Author Response (AR2)

We thank the reviewer 2 for his/her valuable comments and suggestions. The manuscript has benefited immensely from the review process. We appreciate the opportunity to provide further clarifications and add additional changes to improve the manuscript. We included below the reviewer's comments (italic black font) and our responses (plain blue font). Changes in the manuscript are highlighted in red.

**1 Major comments**

1. Results still contain no significance tests or error bars. The authors claim that their random forest outperforms the two baseline models (persistence, which they call the "naïve" model, and linear regression), but there are no significance tests or error bars to support this claim. I made the same comment in round 1, and I find the authors' response (explaining why they chose not to include significance tests as requested) unsatisfactory. See page 6 of the document below for the authors' response (and my response to their response).

We acknowledge the reviewer's point of view in requesting statistical significance tests for model comparison. We performed two-sample Wilcoxon rank-sum tests to compare the correlation coefficients obtained from the three models : RF, naïve, and MLR for each flow regime. Their distributions are also plotted below with p-value included. We also modified the text to reflect this change, "Figure 1 shows the distributions of Pearson correlation coefficient ($r$) between forecasted and observed values obtained from the three models: RF, naïve, and MLR. Non-parametric, two-sample Wilcoxon rank-sum significance tests [Wilcoxon et al., 1970], which is used to assess whether the values obtained between two separate groups on are systematically different from one another, suggest that the pair-wise differences in $r$ values between RF and the other two models are statistically significant ($p < 0.05$) in two flow regimes. RF is observed to outperform both naïve and MLR models in rainfall-driven and transient watersheds. Among snowmelt-driven watersheds, the three models yield similar correlation coefficients ($p > 0.05$). In Fig. 2a and 2b, we observe most points lie on the left of the 1-to-1 line, suggesting that RF outperforms naïve model at individual watersheds in rainfall-driven and transient regimes. We also discern that large improvement, defined as the positive difference in $r$ values between RF and naïve model, tends to occur with lower persistence. This suggests that application of RF would be most benefiting at watersheds where next-day streamflow is less dependent on the condition of the current day. Among snowmelt-driven watersheds, the data points lie on the 1-to-1 line, indicating that the three models show marginal difference in $r$ values." We also performed the two-sample t-test, which assumes the data has a normal distribution, but decided to go with the Wilcoxon test due to its non-parametric nature. The two tests, however, yield very similar results where the Wilcoxon test is even more conservative (giving larger p-values).

Figure 1: Boxplots for Pearson correlation coefficient between forecasted and observed values for three models: RF, naïve, and MLR across three flow regimes. Pair-wise Wilcoxon rank-sum significance tests are performed and p-value (in black) are included for each pair of models.

[Figure]

Figure 2: Pairwise scatter plots of Pearson correlation coefficient between forecasted and observed values for (a) RF vs. naïve model, (b) RF vs MLR, and (c) MLR vs. naïve model. Each dot represents a watershed (n=86).

[Figure]

*2. The authors' claim that random forests are more interpretable than other machine-learning models, is highly debatable. For details, see my first two comments on page 9 of the document below. This comment should be easy to address – I would just like to see the authors add a few sentences to the manuscript, discussing the controversy of model interpretability. i.e., There are properties of random forests that make them more interpretable than other ML models, but there are also properties that make them less interpretable, so it's unfair to simply say "RF allows for some level of interpretability" (as their justification for using random forests) and then move on.*

We agree that saying one ML model is more interpretable than the others may be controversial and made the following modification to the manuscript, "We select RF to forecast streamflow for two reasons. First, RF has been referenced to deliver high performance in short-term streamflow forecasts [Mosavi et al., 2018, Papacharalampous and Tyralis, 2018, Li et al., 2019, Shortridge et al., 2016], making it a good candidate for our study. Second, RF allows for some level of interpretability . This is delivered through two measures of predictive contribution of variables: mean decrease in accuracy (MDA) and mean decrease in node impuritiy (MDI). These two measures have been widely used as means for variable selection in classification and regression studies in bioinformatics [Chen and Ishwaran, 2012], remote sensing classification [Pal, 2005], and flood hazard risk assessment [Wang et al., 2015]. The interpretability of a ML model, however, can be a controversial subject and remains an active area of study [Ribeiro et al., 2016, Carvalho et al., 2019]. Both model-agnostic (e.g., permutation-based feature importance [Breiman, 2001]) and model-specific (e.g., gini-based for RF [Breiman et al., 1984], gradient-based for ANNs [Shrikumar et al., 2017]) interpretation methods can provide useful insights into how the ML models make their predictions.  While the referred interpretability does not directly translate to interpretation of the physical processes, it can provide insight into relationships among predictors and streamflow response.

3. The authors have not clarified which data (training only or training plus validation) they use to compute standardization parameters – i.e., to compute the minimum and maximum for min-max scaling. See my fourth comment on page 12 of the document below. The distinction is important. Only training data should be used to compute standardization parameters; if the validation data are also used to compute standardization parameters, this means that validation data are used to pre-process training data, which means that information from the validation data has "leaked" into the training data, which means that the two datasets are no longer independent.

You are right that only training data should be used to compute the min-max values for variable scaling. This was our approach. First, we computed the min and max values from training data sets for each of the predictor and response variables at each watershed. These min and max values were then used to standardize both training and validation data sets. The training data, which were used to compute min-max values for standardization, therefore have values between 0 and 1. We added this clarification to the text.

4. The authors should justify why they experimented with only two hyperparameters. See my fifth comment on page 12 of the document below.

We added the following justification to the manuscript, "While all considered parameters might have an effect on the performance of RF, we chose to focus on two parameters: `ntree` and `mtry` for a number of reasons. First, these two parameters were originally introduced by [Breiman, 2001] in

the development of RF algorithm. Second, `ntree` in a forest is a parameter that is generally not tunable but should be set sufficiently high [Oshiro et al., 2012, Probst et al., 2019] for RF to achieve optimal performance. Furthermore, empirical results provided in previous works suggest that `mtry` is the most influential out of parameters in RF [Bernard et al., 2009, Van Rijn and Hutter, 2018, Probst et al., 2019].

5. The authors use ROC curves to diagnose (the absence of) systematic overestimation, which is an invalid interpretation of ROC curves. See my comment on line 14 of the document below.

We acknowledge that ROC curve may not be used to diagnose and detect systematic overestimation. In the revised manuscript submitted after round 1, we removed this part of the discussion (please see lines 354-356) for the following reasons. First, as you suggested, detecting systematic overestimation requires plotting the performance diagram, which is not commonly used to evaluate hydrologic models, both data-driven and physically-based, based on our review of the literature (we further discussed why this may be the case below). Second, it also requires introducing both false alarm ratio and false alarm rate and their differences, which you clearly explained and we appreciate it. However, this may prove confusing to the readers considering it's not a major focus of the paper.

6. The "no-skill line" in the ROC curves (Figure 7) is misplaced. The no-skill line should be the x = y line (or POD = false-alarm rate), which is the ROC curve that would be achieved by a random model such as a coin flip.

We apologize for the confusion. The "no-skill line" in the ROC curves is not misplaced. You are right that the no-skill line should be the x = y line (or POD = false-alarm rate FAR) and this was the case for our figure. In Figure 7 of our manuscript, we showed a partial ROC curve (please note the scales of the x-axis and y-axis are not 1-to-1 here) with FAR < 0.065, which is slightly larger than the maximum FAR value for the three threshold across all watersheds. When we set xlim = ylim, the data points are clumped together. We provide the two plots below to illustrate where we believe the bottom plot, which is included in the manuscript, presents the data better.

Figure 3: The probability of detection (POD) plotted against the false alarm rate (FAR) for three extreme thresholds: 90th, 95th, and 99th percentiles. Thin black line connects values from the same watershed. (Vertical axis) Number of times RF *correctly* forecasted events that exceeded the threshold divided by the total number of exceedance. (Horizontal axis) Number of times RF *incorrectly* forecasted events that exceeded the threshold divided by the total number of non-exceedance.

[Figure]

Figure 4: The probability of detection (POD) plotted against the false alarm rate (FAR) for three extreme thresholds: 90th, 95th, and 99th percentiles. Thin black line connects values from the same watershed. (Vertical axis) Number of times RF *correctly* forecasted events that exceeded the threshold divided by the total number of exceedance. (Horizontal axis) Number of times RF *incorrectly* forecasted events that exceeded the threshold divided by the total number of non-exceedance.

[Figure]

*7. I don't understand why the authors include only 3 probability thresholds in their ROC curves (90%, 95%, and 99%). A typical ROC curve includes probability thresholds spanning the range from 0-100% (typically in increments of 10% at most – usually in increments of 1which allows for a smoother curve). Seeing such a small portion of the full ROC curve, it is difficult to assess the models' performance. I request that the authors plot the full ROC curve for each model, like the ones shown/explained here: https://developers.google.com/machinelearning/crash-course/classification/roc-and-auc. Plotting the full ROC curves would also allow the authors to compute area under the ROC curve (AUC), which is a scalar typically used to quantify the goodness of a ROC curve.*

We intend to use this partial ROC curve to facilitate our discussion of section 4.4, which focuses on the performance of RF on detecting extreme streamflows. We also justified our selection of the three thresholds on lines 185-192. As you mentioned that we could technically compute thresholds in the increments of 10%, plot the full ROC curve, compute the area under curve, and evaluate the skillfulness of RF using this quantity. However, we believe that it is more appropriate to limit the discussion of POD, FAR, and the ROC curve only to extreme events as this is related to flood forecasting (where floods are the results of extreme streamflow exceeding specific thresholds and clearly defined as binary flood/no-flood events). The thresholds we chose are often used to study flood characteristics and have physical interpretations. But this also depends on the watershed and the evaluation of historical floods. Flood forecasting, however, is not the focus of our paper. Our study and the ones you cited below and in round 1 (both focus on storm detection) have very different aims. The main outcome in these studies is binary and the events are clearly defined (e.g., cyclones). Therefore, the use of performance diagram and ROC provides valuable insights. In our study, the outcome is a continuous variable and it does not make sense to evaluate the performance of the model using these tools. For example, if the observed streamflow is 30 $m^3/s$, which also happens to be the 10th percentile streamflow at a watershed, and our prediction is 32 $m^3/s$. The prediction, in this case, can neither be considered a "hit" or a "miss". Therefore, correlation coefficient $r$, $R^2$, KGE, MAE, and RMSE are better criteria and were used to evaluate the overall performance of the RF model. To make it more clear, we added the following text to manuscript, "In hydrological forecast, one might be interested in the ability of the model to capture more extreme events rather than the overall performance. This is particularly relevant in flood risk assessment and flood forecasting where floods are associated with discharge exceeding a high percentile (typically $\geq$ 90th)[Cayan et al., 1999]."

*As in round 1, I request that the authors include performance diagrams along with the ROC curves (see my last comment on page 13 of the document below). Performance diagrams plot probability of detection (POD) on the y-axis vs. success ratio (1 minus false-alarm ratio, which is different than false-alarm rate) on the x-axis. Since frequency bias = POD / success ratio, performance diagrams can be used to diagnose systematic overestimation, which the authors have identified as*

*a goal. Performance diagrams look like Figure 4 in this paper:* `https: // journals. ametsoc. org/ view/ journals/ wefo/ 34/ 6/ waf-d-19-0094_ 1. xml? tab_ body=fulltextdisplay`. *If it proves too complicated to plot the contours in the background (frequency bias and critical success index), I would be okay with the authors leaving the contours out.*

We provided our reasons for not including full ROC curve and performance diagram above.

*9. The discussion on lines 384-391 is unclear throughout. See inline comments.*

We addressed these comments below.

*10. Mean-flow benchmark. On lines 331-332, the authors cite a previous paper to claim that a Kling-Gupta efficiency (KGE) $> -0.41$ improves upon the "mean-flow benchmark," which is the KGE achieved by always predicting the time-mean flow ("climatology") at the given basin. I request that the authors compute the mean-flow benchmark for their own dataset, since it may be different than the dataset used in the paper they cite.*

To clarify, the author derived and concluded that a Kling-Gupta efficiency (KGE) $> -0.41$ improves upon the mean-flow benchmark, which is the KGE achieved by always predicting the time-mean flow ("climatology") at *any* basin. We added the following modification to the text to make it less ambiguous, "Knoben et al. (2019) suggested that, at any given basin, a KGE score greater than -0.41 indicates a hydrologic model improves upon the mean flow benchmark. Therefore, RF can be seen to give satisfactory performance at all watersheds in our study based on the KGE scores.

**2 Reviewer's response to our response after round 1**

*Okay, but it still won't make sense to most readers to use "neuro-fuzzy" as a noun. Please replace in the text with "neuro-fuzzy (a combination of ANNs and fuzzy logic)".*

We have added your suggestion to the manuscript, " Artificial neural networks (ANN), neuro-fuzzy (a combination of ANNs and fuzzy logic), support vector machine (SVM), and decision trees (DT) are reported to be among the most popular and effective for both short-term and long-term flood forecast [Mosavi et al., 2018]."

*This is still unclear in the text. Please replace "two other competing ML models" with "the other two ML models" or something similar. Currently it's not clear that "two other competing ML models" are the ones you mentioned in the previous sentence.*

We made the following modification to the text, "They found BNN outperformed multiple linear regression (MLR) as well as the other two ML models.

*If you don't think "baseflow separation" is worth defining, please don't mention it. If it's worth mentioning, I think it's worth defining.*

We defined what baseflow saeration as you suggested, "Tongal and Booij [2018] forecasted daily streamflow in four rivers in the United States with SVR, ANN, and RF coupled with a baseflow separation (i.e., separating the two different components of streamflow into baseflow and surface flow) method.

*Please include this explanation in the text. The following sentence is a non-sequitur, because like I said in the first round of reviews, using only antecedent information as predictors should not limit your lead time.*

We added the following explanation to the manuscript, "In [Rasouli et al., 2012], the authors forecasted streamflow at 1-7 day lead times using three ML models and data from combinations of climate indices and local meteo-hydrologic observations. The authors concluded that models with local observations as predictors were generally best at shorter lead times while models with local observations plus climate indices were best at longer lead times of 5–7 days. Also, the skillfulness of all three models decreased with increasing lead times. In our study, we focused on 1-day lead time forecasting and therefore did not include long-term climate information.  "

*You cannot use the existence of the permutation test as an argument that random forests are more interpretable than other ML models, because like you say in this paragraph, the permutation test is model-agnostic. So your only valid argument here is that the Gini-based importance test can be applied to random forests and not other models. However, there are many interpretation methods that can be applied to other models (e.g., neural networks) and not random forests. Please include some of this discussion in the main text. ML interpretability is a very controversial topic, and I think it's unfair to simply say "RF allows for some level of interpretability" and then move on. Please include some*

*of this discussion in the main text. ML interpretability is a very controversial topic, and I think it's unfair to simply say "RF allows for some level of interpretability" and then move on.*

We addressed these comments in major comment 2.

*Please clarify this in the main text. ML specialists might be confused when they see the word "parameter" being used to mean hyperparameter.*

We added the following clarification to the text, "The remainder of the paper is arranged as follows. Section 2 provides a brief introduction to RF, relevant parameters (which can also be referred as "hyper-parameters" in the ML literature), and selected evaluation criteria. Section 3 describes the study area, datasets, and predictor selection. Resultsand discussion are given in Sect. 4 along with limitations and recommendation for future research. A summary and indication of future work are is provided in Sect 5."

*Please clarify this in the main text.*

We added the clarification to the manuscript, "Non-parametric methods do not assume any particular family for the distribution of the data[Altman and Bland, 1999]."

*Okay, but what did you do with the Caret package? Did you use the Caret package to train the random forests, or just to loop through hyperparameters?*

We used Caret to loop through the `mtry` parameter. This was stated on lines 117-118. "In this study, we select the optimal `mtry` using an exhaustive search strategy, in which all possible values of `mtry` are considered, using R package `Caret` [Kuhn et al., 2008]".

*In this case, please add "as explained in the remainder of this section" to the end of the sentence.*

We have added your suggestion to the text.

*For both MAE and RMSE, the range is always [0, inf) and the optimal value is 0. (Of course most models are good enough that they do not produce errors of infinity, but they could.)*

We have added the following clarification to the text ,"MAE and RMSE values range between 0 and $\infty$ where a score of 0 indicates a perfect match between predicted and observed data".

*Please replace "as predictors" in this sentence with "as predictors in the random forest" or "as predictors for streamflow" or something similar.*

We replaced "as predictors" with "as predictors for streamflow".

*Please state explicitly that this is a limitation of your work.*

We explicitly stated and discussed this limitation under 4.7 Limitations and future research on lines 430-435.

*Please put this in the main text.*

This was added."Given that there is high temporal correlation in daily temperatures, TMIN and TMAX data can provide useful signal to our streamflow forecast."

*What do you mean by "error"? Out-of-bag MAE? Whatever the answer is, please state it in the main text.*

We clarified this in the text, "We tested RF on training data sets of 30 randomly chosen watersheds and observed that the reduction in out-of-bag MAE error is negligible after 2000 trees. We then set `ntree`=2000 for all 86 watersheds.

*Yes, in this case M/3 and ceil(sqrt(M)) are the same, but in general they are not. So my comment remains.*

We're not sure what clarification you're asking. M/3 and ceil(sqrt(M)) are only the same when M is divisible by 3. Since *mtry* cannot be fractional, we have the option to round up or round down. But we ended up tuning this parameter and decided to remove this line of text to avoid potential confusion.

*Please put this explanation in the text.*

The explanation was added to the main text.

*Please put this in the main text, to clarify that the statement "there is less variability in flow behaviors at individual gauges in this group" is backed up by data, rather than being pure conjecture.*

This clarification was added.

*Please clarify this in the main text. Without the clarification, I imagine a lot of readers will have the same question I did.*

The following clairifcaiton was added to the test, "The MAE scores are heavily skewed towards 0 while RMSE scores are more evenly spread among snowmelt-driven watersheds."

*The ROC curve and performance diagram are very different things. The ROC curve plots POD vs. false-alarm rate (sometimes called probability of false detection or POFD), which is b / (b + d) in the contingency table. The performance diagram plots POD vs. false-alarm \*ratio\*, which is b / (a + b) in the contingency table. False-alarm \*rate\* and \*ratio\* tend to be very different for rare*

events, because b + d (the number of actual non-events) tends to be much greater than a + b (the number of forecast events). Thus, models with very good ROC curves often have poor performance diagrams. This is why it is crucial to show both.

We addressed this comment above and why we thought it might be confusing to readers to introduce both false-alarm rate and false-alarm ratio considering that it is not the main focus of our paper. You suggest that models with very good ROC curves often have poor performance diagrams and this makes it crucial to show both. While this may be the case, it's not our intention to argue that partial ROC in our study is "good". In fact, our main observation is that RF becomes "expectedly less skilful in its forecasts with increase in magnitude of the events." In the rest of the discussion in this section, we followed closely with the interpretation of POD and FAR, which we clearly defined.

You cannot detect systematic overestimation from a ROC curve. Systematic overestimation occurs when frequency bias ¿ 1, and frequency bias = (a + b) / (a + c), where a = number of true positives; b = number of false positives; and c = number of false negatives. In other words, frequency bias is (number of forecast events) / (number of actual events). Frequency bias can be contoured in the background of a performance diagram, which is another reason that I've requested you include performance diagrams along with ROC curves.

We addressed this comment above.

**3 Reviewer's comments to the manuscript**

We here address the inline comments. Simple typo fixes were made directly in the revised manuscript.

The fractions are awkward. Please replace with "0.0625" and "0.05"

We believe the current specification is consistent with the current literature on VIC model.

Please replace with "depending on the input variables provided". The current phrasing makes it unclear whether "the selection of input variables" is done by the user or by the random forest.

We replaced "the selection of input variables" with "the input variables provided" as you suggested.

Figure 1 should be referenced throughout Section 2.1, not just at the end. I think referencing Figure 1 throughout could make the explanation much more clear. (I also made this comment in round 1.)

We made the following modification to the text, "Since a single decision tree can produce high variance and is prone to noise [James et al., 2013], RF addresses this limitation by generating multiple trees where each tree is built on a bootstrapped sample of the training data (Fig. 2). Each time a binary split is made in a tree (also known as split node), a random subset of predictors (without replacement) from the full set of predictor variables is considered (Fig. 2)."

Please explain this more clearly. (I also made this comment in round 1.)

This is a bit vague for us to provide clarification. Our current explanation, "One predictor from these candidates is used to make the split where the expected sum variances of the response variable in the two resulting nodes is minimized," is consistent with the principle of regression tree explained in the Elements of Statistical Learning text [Friedman et al., 2001].

Please write that this method was developed by Breiman (2001).

We believe Breiman developed both feature importance measures: Gini-based in an earlier paper on classification and regression trees [Breiman et al., 1984] and permutation-based in the original RF paper [Breiman, 2001]. The following clarification was added, "There are two built-in measures for assessing variable importance in RF : mean decrease in accuracy (MDA) and mean decrease in node impurity (MDI). Both were developed by Breiman [Breiman et al., 1984, Breiman, 2001]."

I don't understand the need for such a convoluted phrase. Could you replace with just "average error reduction"?

Thanks for the suggestion. We replaced "average gain in residual error reduction" with "average error reduction".

Please insert "Pearson" here, since you have been calling it the "Pearson correlation coefficient" elsewhere.

We added "Pearson" to the main text.

Figure 2 should be referenced much earlier (at the beginning of this section and also in the introduction, where you first mention the HUC 17 region). It is difficult to understand all this description without a map. (I also made this comment in round 1.)

As you suggested in round 1, Fig. 1 was first referenced in the Introduction (line 63) when the study area was first mentioned. We referenced it again here at the end of the first sentence of this paragraph in the revised manuscript, "In this study, we focus on watersheds in the Pacific Northwest

Hydrologic Region (Fig. 1)." We think it becomes redundant to keep referencing it after this point because we remain talking about the same region.

*More moderate than what? Does this statement apply to the whole domain, just the west side of the Cascades, or what?*

We provided the clarification to the manuscript, "For this region, proximity to the ocean creates a more moderate climate with a narrower seasonal temperature range compared to the inland areas, particularly in the winter."

*? Similar to what?*

We made the following modification, "Here, we observe a similar trend in $R^2$, KGE, MAE, and RMSE scores compared $r$ values in Fig. 6 where RF performs better in snowmelt-dominated than in rainfall-dominated (higher $R^2$ and KGE, lower MAE and RMSE)."

*You could verify this by computing KGE for the mean-flow benchmark on your own dataset, no?*

We addressed this comment above.

*How did you compute the "no-skill line"? The no-skill line in ROC curves is typically the line where POD = FAR.*

We addressed this comment above.

*According to Figure 8, for snow-melt-dominated watersheds, pentad is either the 3rd- or 4th-most important predictor.*

Thanks for spotting this. We changed the text to the following,"Surprisingly, pentad comes third and fourth in MDI and MDA respectively."

*Do you mean the PDF (probability-density function)?*

Yes, and we modfied the text in the manuscript.

*Why would MDA, but not MDI, underestimate the importance of variables with a non-normal distribution?*

Precipitation data is generally zero inflated (at least 30 percent in our dataset depending on the watershed). As a result, there is a high likelihood that the day with zero precipitation ends up with the same value during the shuffling process used to compute MDA. While we did not perform additional simulation to explore this as it is out of the scope of our paper, we believe it is worth discussing and can be investigated in future research. This is the reason our conclusion is that ,"We suggest RF users to exert cautionwhen interpreting outputs from these two measures." We added this to the discussion.

*What is a "potential" predictor variable?*

We deleted "potential" in the manuscript.

*This sentence is a non-sequitur. In the previous sentence you talked about normal vs.non-normal distributions – the normality of a distribution has nothing to do with the scale of measurement (e.g., if you multiply a [non-]normally distributed variable by 1000, it is still [non-]normally distributed). Also, why would you mention number of categories here? Both temperature and precipitation are continuous, not categorical, variables.*

We agree that both variables, precipitation and temperature, are not categorical variables and removed "their number of categories" from the text. However, among our 8 predictors in our study, pentad is considered an ordinal variable. Also, the scales of measurement of precipitation and temperature variables are slightly different. Precipitation is a flux variable and comprises discrete and continuous components in that if it does not rain the amount of rainfall is discrete whereas if it rains the amount is continuous. Temperature is a state variable and always continuous. Therefore, we believe the findings in Strobl et al. (2007) are relevant for our discussion on variable importance.

*Are you suggesting that the two temperature variables (min and max) have more correlation with other predictors than do the two precip variables (1-day and 3-day)? If so, have you verified this by computing the correlations in your dataset?*

Thanks for the opportunity to clarify this. Yes, temperature variables tend to have more correlation with other predictors than do the two precipitation variables in our dataset. This is likely because temperature controls both the form of precipitation (snowfall vs rainfall) as well as timing of snowmelt. However, due to the blackbox nature of ML models, we don't know for sure if this is directly related to the observed patterns in MDI and MDA.

*What do you mean by the "stability [of a measure] across different datasets"?*

These separate simulation studies used different datasets and concluded that the variable importance ranks based on the two measures (MDA and MDI) can be unreliable.

*The font here is way too small. Please enlarge.*

Thanks for your suggestion. Please see the modified figure below.

Figure 5: Gauge locations with color gradient indicating variations in (a) watershed drainage area, (b)mean watershed elevation, (c) mean watershed annual precipitation, and (d) mean watershed annual temperature.

[Figure]

*The full range (from min to max)?*
That is correct.

**References**

D. G. Altman and J. M. Bland. Statistics notes variables and parameters. *Bmj*, 318(7199):1667, 1999.

S. Bernard, L. Heutte, and S. Adam. Influence of hyperparameters on random forest accuracy. In *International Workshop on Multiple Classifier Systems*, pages 171–180. Springer, 2009.

L. Breiman. Random forests. *Machine learning*, 45(1):5–32, 2001.

L. Breiman, J. Friedman, C. J. Stone, and R. A. Olshen. *Classification and regression trees*. CRC press, 1984.

D. V. Carvalho, E. M. Pereira, and J. S. Cardoso. Machine learning interpretability: A survey on methods and metrics. *Electronics*, 8(8):832, 2019.

D. R. Cayan, K. T. Redmond, and L. G. Riddle. Enso and hydrologic extremes in the western united states. *Journal of Climate*, 12(9):2881–2893, 1999.

X. Chen and H. Ishwaran. Random forests for genomic data analysis. *Genomics*, 99(6):323–329, 2012.

J. Friedman, T. Hastie, and R. Tibshirani. *The elements of statistical learning*, volume 1. Springer series in statistics New York, 2001.

G. James, D. Witten, T. Hastie, and R. Tibshirani. An introduction to statistical learning, volume 103 xiv of, 2013.

M. Kuhn et al. Building predictive models in r using the caret package. *Journal of statistical software*, 28(5):1–26, 2008.

X. Li, J. Sha, and Z.-L. Wang. Comparison of daily streamflow forecasts using extreme learning machines and the random forest method. *Hydrological Sciences Journal*, 64(15):1857–1866, 2019.

A. Mosavi, P. Ozturk, and K.-w. Chau. Flood prediction using machine learning models: Literature review. *Water*, 10(11):1536, 2018.

T. M. Oshiro, P. S. Perez, and J. A. Baranauskas. How many trees in a random forest? In *International workshop on machine learning and data mining in pattern recognition*, pages 154–168. Springer, 2012.

M. Pal. Random forest classifier for remote sensing classification. *International Journal of Remote Sensing*, 26(1):217–222, 2005.

G. A. Papacharalampous and H. Tyralis. Evaluation of random forests and prophet for daily streamflow forecasting. *Advances in Geosciences*, 45:201–208, 2018.

P. Probst, M. N. Wright, and A.-L. Boulesteix. Hyperparameters and tuning strategies for random forest. *Wiley Interdisciplinary Reviews: Data Mining and Knowledge Discovery*, 9(3):e1301, 2019.

K. Rasouli, W. W. Hsieh, and A. J. Cannon. Daily streamflow forecasting by machine learning methods with weather and climate inputs. *Journal of Hydrology*, 414:284–293, 2012.

M. T. Ribeiro, S. Singh, and C. Guestrin. Model-agnostic interpretability of machine learning. *arXiv preprint arXiv:1606.05386*, 2016.

J. E. Shortridge, S. D. Guikema, and B. F. Zaitchik. Machine learning methods for empirical streamflow simulation: a comparison of model accuracy, interpretability, and uncertainty in seasonal watersheds. *Hydrology and Earth System Sciences*, 20(7):2611–2628, 2016.

A. Shrikumar, P. Greenside, and A. Kundaje. Learning important features through propagating activation differences. *arXiv preprint arXiv:1704.02685*, 2017.

H. Tongal and M. J. Booij. Simulation and forecasting of streamflows using machine learning models coupled with base flow separation. *Journal of hydrology*, 564:266–282, 2018.

J. N. Van Rijn and F. Hutter. Hyperparameter importance across datasets. In *Proceedings of the 24th ACM SIGKDD International Conference on Knowledge Discovery & Data Mining*, pages 2367–2376, 2018.

Z. Wang, C. Lai, X. Chen, B. Yang, S. Zhao, and X. Bai. Flood hazard risk assessment model based on random forest. *Journal of Hydrology*, 527:1130–1141, 2015.

F. Wilcoxon, S. Katti, and R. A. Wilcox. Critical values and probability levels for the wilcoxon rank sum test and the wilcoxon signed rank test. *Selected tables in mathematical statistics*, 1:171–259, 1970.

---

## Author Response (AR5)

The authors would like to thank both reviewers and the editor for their valuable comments. We sincerely believe that our manuscript has benefited tremendously from the review process. We have revised the manuscript following the reviewer 2's suggestions by adding our clarifications to the reviewer to the main text.